# Nonlinear Laplacians: Tunable principal component analysis under directional prior information

**Yuxin Ma**
Applied Mathematics & Statistics
Johns Hopkins University
Baltimore, MD 21211
yma93@jhu.edu

**Dmitriy Kunisky**
Applied Mathematics & Statistics
Johns Hopkins University
Baltimore, MD 21211
kunisky@jhu.edu

## Abstract

We introduce a new family of algorithms for detecting and estimating a rank-one signal from a noisy observation under prior information about that signal's direction, focusing on examples where the signal is known to have entries biased to be positive. Given a matrix observation $\boldsymbol{Y}$, our algorithms construct a *nonlinear Laplacian*, another matrix of the form $\boldsymbol{Y} + \mathrm{diag}(\sigma(\boldsymbol{Y}\boldsymbol{1}))$ for a nonlinear $\sigma : \mathbb{R} \to \mathbb{R}$, and examine the top eigenvalue and eigenvector of this matrix. When $\boldsymbol{Y}$ is the (suitably normalized) adjacency matrix of a graph, our approach gives a class of algorithms that search for unusually dense subgraphs by computing a spectrum of the graph "deformed" by the degree profile $\boldsymbol{Y}\boldsymbol{1}$. We study the performance of such algorithms compared to direct spectral algorithms (the case $\sigma = 0$) on models of sparse principal component analysis with biased signals, including the Gaussian planted submatrix problem. For such models, we rigorously characterize the strength of rank-one signal, as a function of the nonlinearity $\sigma$, required for an outlier eigenvalue to appear in the spectrum of a nonlinear Laplacian matrix. While identifying the $\sigma$ that minimizes the required signal strength in closed form seems intractable, we explore three approaches to design $\sigma$ numerically: exhaustively searching over simple classes of $\sigma$, learning $\sigma$ from datasets of problem instances, and tuning $\sigma$ using black-box optimization of the critical signal strength. We find both theoretically and empirically that, if $\sigma$ is chosen appropriately, then nonlinear Laplacian spectral algorithms substantially outperform direct spectral algorithms, while retaining the conceptual simplicity of spectral methods compared to broader classes of computations like approximate message passing or general first order methods.

## 1 Introduction

Principal component analysis (PCA) is one of the most ubiquitous computational tasks in statistics and data science, seeking to extract informative low-rank structures from noisy observations organized into matrices (see, e.g., [AW10, JC16, JP18, GGH+22] for a few of the huge number of available surveys). We will study a family of mathematical models of such problems involving detecting or recovering these low-rank structures. These problems are specified by a family of probability measures $\mathbb{P}_{n,\beta}$ on $n \times n$ symmetric matrices, where $\beta$ is a *signal-to-noise* parameter. The observed matrix is biased in the direction of a low-rank *signal*: there is a latent unobserved unit vector $\boldsymbol{x} \in \mathbb{S}^{n-1} \subset \mathbb{R}^n$ such that, when $\boldsymbol{Y} \sim \mathbb{P}_{n,\beta}$, then

$$\mathbb{E}\left[\boldsymbol{Y} \mid \boldsymbol{x}\right] \approx \beta\sqrt{n} \cdot \boldsymbol{x}\boldsymbol{x}^\top.$$

For this class of problems, we consider two computational tasks:

39th Conference on Neural Information Processing Systems (NeurIPS 2025).

1. **Detection:** Determine whether the observed data is uniformly random ($\beta = 0$) or contains a signal ($\beta > 0$). In particular, we say that a sequence of functions $f_n : \mathbb{R}^{n \times n}_{\mathrm{sym}} \to \{0, 1\}$ (usually encoding a single algorithm allowing for various $n$) achieves *strong detection* if

$$\lim_{n \to \infty} \mathbb{P}_{n,\beta}\left(f_n(\boldsymbol{Y}) = 1\right) = \lim_{n \to \infty} \mathbb{P}_{n,0}\left(f_n(\boldsymbol{Y}) = 0\right) = 1.$$

2. **Recovery:** When $\beta > 0$, estimate the hidden signal $\boldsymbol{x}$.[1] We say that a sequence of functions $\widehat{\boldsymbol{x}} = \widehat{\boldsymbol{x}}_n : \mathbb{R}^{n \times n}_{\mathrm{sym}} \to \mathbb{S}^{n-1}$ achieves *weak recovery* if, for some $\delta > 0$,

$$\liminf_{n \to \infty} \mathbb{P}_{n,\beta}\left(|\langle \widehat{\boldsymbol{x}}_n(\boldsymbol{Y}), \boldsymbol{x}\rangle| \geq \delta\right) > 0,$$

and achieves *strong recovery* if $|\langle \widehat{\boldsymbol{x}}_n(\boldsymbol{Y}), \boldsymbol{x}\rangle| \to 1$ in probability as $n \to \infty$.

A common approach to such problems that we detail in Section 3.1 is to "perform PCA" on $\boldsymbol{Y}$ directly, meaning in this context to look for an unusually large eigenvalue to test whether $\beta = 0$ or $\beta > 0$, and to estimate $\boldsymbol{x}$ by the eigenvector associated to such an eigenvalue. We call this a *direct spectral algorithm*. This approach is effective, but is agnostic to any information we might have in advance about the hidden $\boldsymbol{x}$. In this paper, we propose a new framework for improving direct spectral algorithms when we have some knowledge about $\boldsymbol{x}$. In particular, our approach will be sensible when we have *directional* information about $\boldsymbol{x}$, say that it lies in a given cone, a class of problem proposed by [DMR14]. Our approach is probably *not* well-suited to other kinds of structural prior information that other works have sought to exploit, like sparsity [ZHT06, AW08, JL09, DM14] or the opposite assumption of a perfectly flat signal from the hypercube, $\boldsymbol{x} \in \{\pm 1/\sqrt{n}\}^n$ [DAM15, FMM18].

To be concrete, we will consider the following class of models where $\langle \boldsymbol{x}, \boldsymbol{1}\rangle$ is somewhat large with high probability.

**Definition 1.1** (Sparse Biased PCA). *Let $p = p(n) \in [0, 1]$ satisfy*

$$\omega\left(\frac{\log n}{n}\right) \leq p(n) \leq o(1).$$

*Let $\eta$ be a probability measure with positive mean ($\mathbb{E}_{z \sim \eta}\, z > 0$) and finite third absolute moment ($\mathbb{E}_{z \sim \eta} |z|^3 < \infty$). Let $\boldsymbol{x} = \boldsymbol{y}/\|\boldsymbol{y}\|$ with $\boldsymbol{y} \in \mathbb{R}^n$ having entries $y_i = \varepsilon_i z_i$, where $z_i \overset{\mathrm{i.i.d.}}{\sim} \eta$ and $\varepsilon_i \in \{0, 1\}$ are drawn in one of the following ways:*

- ***Random Subset:*** *Sample $S \subseteq [n]$ uniformly at random among subsets of size $|S| = pn$, and set $\varepsilon_i := \mathbb{1}\{i \in S\}$.*

- ***Independent Entries:*** *Draw $\varepsilon_i \overset{\mathrm{i.i.d.}}{\sim} \mathrm{Ber}(p)$.*

*We call this choice the* sparsity model. *Finally, we draw $\boldsymbol{Y} \sim \mathbb{P}_{n,\beta}$ from this model as*

$$\boldsymbol{Y} = \beta\sqrt{n} \cdot \boldsymbol{x}\boldsymbol{x}^\top + \boldsymbol{W}$$

*for $\boldsymbol{W}$ a symmetric matrix drawn from the* Gaussian orthogonal ensemble (GOE)*, i.e., having $W_{ij} = W_{ji} \sim \mathcal{N}(0, 1 + \mathbb{1}\{i = j\})$ independently.[2] We call $(\eta, p(n))$ the* model parameters *and $\beta$ the* signal strength *of the model.*

In these models, clearly we have arranged to have $\mathbb{E}[\boldsymbol{Y} \mid \boldsymbol{x}] = \beta\sqrt{n} \cdot \boldsymbol{x}\boldsymbol{x}^\top$ exactly. Our models are in the spirit of Non-Negative PCA [MR15], where the stricter entrywise condition $x_i \geq 0$ is imposed. On the other hand, for technical reasons we focus on sparse $\boldsymbol{x}$, though our algorithms seem sensible for dense $\boldsymbol{x}$ as well (see Appendix D.3).

For sparsity $p(n) = o(1)$, it is believed that optimal algorithms for such problems have a tradeoff between runtime and performance, in the sense that one may spend more time computing and in return identify weaker signals (with a smaller value of $\beta$). In contrast, for $p(n)$ a constant, there is a single critical $\beta_*$ such that a polynomial-time algorithm can identify signals of strength $\beta > \beta_*$, while doing so for $\beta < \beta_*$ is believed to require nearly exponential time. In the sparse case see,

---

[1]Since the actual signal is $\boldsymbol{x}\boldsymbol{x}^\top$ which is unchanged by negating $\boldsymbol{x}$, it is only sensible to ask to estimate either this matrix or $\{\boldsymbol{x}, -\boldsymbol{x}\}$, which is why we take the absolute value of the inner product of an estimator with $\boldsymbol{x}$.

[2]The choice of scaling of the diagonal variances has no effect on our results; see our Proposition B.10.

e.g., [AKS98] for the case of the planted clique problem discussed below, or [DKWB23] for Gaussian models as above. Our goal here is to study a particular aspect of the former, more algorithmically flexible situation. Namely, we will ask: how weak of a signal can one detect without straying too far from the direct spectral algorithm?

To give a concrete example, the following is one special case of our model that has been widely studied [MRZ15, MW15, HWX17, CX16, BMV$^+$18, BBH19, LM19, GJS21].

**Example 1.2** (Gaussian Planted Submatrix). *Let $\eta := \delta_1$ be the probability measure concentrated on the constant 1, let $p(n) := \beta/\sqrt{n}$ for the same $\beta$ as the signal strength, and use the Random Subset sparsity model. The result is that $Y$ is the Gaussian random matrix $W$, where a random principal submatrix of dimension $\beta\sqrt{n}$ has had the means of its entries elevated from 0 to 1.*

While we focus on the specific case of $x$ correlated with $\mathbf{1}$, in Appendix F.3 we discuss possible extensions to other forms of directional prior information.

## 1.1 Summary of contributions

In this paper, we first propose a new PCA algorithm that seeks to incorporate our prior information about $x$ by deforming the matrix $Y$ before computing its top eigenpair. Namely, we add to (a normalized version of) $Y$ a diagonal matrix $D$ with entries given by a bounded nonlinear function $\sigma$ of (a normalized version of) the entries of $Y\mathbf{1}$, for $\mathbf{1}$ the all-ones vector. The idea behind these *nonlinear Laplacian* matrices is that, since $xx^\top$ is rank-one and $x$ is positively correlated with $\mathbf{1}$, both the spectrum of $Y$ and the vector $Y\mathbf{1}$ carry information about $x$. Forming a diagonal matrix from the latter and attenuating its largest entries by applying $\sigma$ lets these two sources of information cooperate, leading to a more effective spectral algorithm for detecting and estimating the signal.

We then give a complete description of the appearance of unusually large "outlier" eigenvalues in the spectrum of such matrices built from $Y$ drawn from models as in Definition 1.1. The appearance of such outliers corresponds to when, for instance, an algorithm thresholding the largest eigenvalue can detect the presence of a signal. For a large class of $\sigma$ and $\eta$, we identify the $\beta_* = \beta_*(\sigma)$ such that there is an outlier in the $\sigma$-Laplacian if and only if $\beta > \beta_*(\sigma)$ (Theorem 3.3 and its subsequent discussion). This analysis applies sophisticated tools developed in prior work on random matrix theory and free probability, and gives $\beta_*(\sigma)$ via a sequence of integral equations involving $\sigma$ and $\eta$. As a consequence we learn that, for instance for the concrete example of the Gaussian Planted Submatrix problem, nonlinear Laplacian algorithms considerably outperform direct spectral algorithms (Theorem 3.4 and Figure 1).

Because the description of $\beta_*(\sigma)$ is rather complex and seems unlikely to admit a closed form, we also explore several heuristics for identifying an effective $\sigma$ for a given problem. We find that our algorithms are quite robust to this choice, and a good $\sigma$ can equally well be found by hand, learned from data, or tuned by black-box optimization (Figure 2). Further, nonlinear Laplacian algorithms appear to be robust across the models described in Definition 1.1; a $\sigma$ that we optimize for one such model is quite effective for others (Appendix F.2). Based on these findings, we argue that nonlinear Laplacian algorithms give a simple, robust, and substantial improvement over direct spectral algorithms, using a bare minimum of extra information about the input matrix beyond its spectrum.

## 2 Nonlinear Laplacian spectral algorithms

We will work with the following normalization of $Y$ in the setting of Definition 1.1:

$$\widehat{Y} := Y/\sqrt{n} = \beta xx^\top + \widehat{W}$$

for $\widehat{W} := W/\sqrt{n}$. As we will describe, in our models this ensures that the extreme eigenvalues of $\widehat{Y}$ are of constant order. We will construct algorithms for PCA using the following class of matrix.

**Definition 2.1** ($\sigma$-Laplacian matrix). *Given the observed matrix $Y$ and a scalar function $\sigma : \mathbb{R} \to \mathbb{R}$, we define the $\sigma$-Laplacian as:*

$$L = L_\sigma(\widehat{Y}) := \widehat{Y} + \underbrace{\mathrm{diag}(\sigma(\widehat{Y}\mathbf{1}))}_{=:D_\sigma(\widehat{Y})=D}$$

*where $\sigma$ applies entrywise to the vector $\widehat{Y}\mathbf{1} \in \mathbb{R}^n$.*

**Definition 2.2** ($\sigma$-Laplacian spectral algorithms). *Given* $\sigma : \mathbb{R} \to \mathbb{R}$ *and* $\tau \in \mathbb{R}$, *the associated* $\sigma$-Laplacian spectral algorithm for detection *is* $f : \mathbb{R}^{n \times n}_{\mathrm{sym}} \to \{0, 1\}$ *that outputs*

$$f(\boldsymbol{Y}) := \mathbb{1}\{\lambda_1(\boldsymbol{L}_\sigma(\widehat{\boldsymbol{Y}})) \geq \tau\},$$

*and the associated* $\sigma$-Laplacian spectral algorithm for recovery *is* $\widehat{\boldsymbol{v}} : \mathbb{R}^{n \times n}_{\mathrm{sym}} \to \mathbb{S}^{n-1}$ *that outputs*

$$\widehat{\boldsymbol{v}}(\boldsymbol{Y}) := \boldsymbol{v}_1(\boldsymbol{L}_\sigma(\widehat{\boldsymbol{Y}})).$$

*Here* $(\lambda_1(\cdot), \boldsymbol{v}_1(\cdot))$ *denote the top eigenpair of a matrix.*

As we discuss in Section 3.1, the direct spectral algorithms that previous work has focused on are special cases of the above with $\sigma = 0$.

For technical reasons (see Section 3.4) it is easier to work with the following variant of the $\sigma$-Laplacian.

**Definition 2.3** (Compressed $\sigma$-Laplacian matrix). *For each* $n \geq 1$, *fix* $\boldsymbol{V} \in \mathbb{R}^{n \times (n-1)}$ *with columns an orthonormal basis of the orthogonal complement of the span of the all-ones vector* $\boldsymbol{1}$. *Given* $\widehat{\boldsymbol{Y}} \in \mathbb{R}^{n \times n}_{\mathrm{sym}}$ *and* $\sigma$ *as before, define the* compressed $\sigma$-Laplacian *as* $\widetilde{\boldsymbol{L}}_\sigma(\widehat{\boldsymbol{Y}}) := \boldsymbol{V}^\top \boldsymbol{L}_\sigma(\widehat{\boldsymbol{Y}})\boldsymbol{V} \in \mathbb{R}^{(n-1) \times (n-1)}$. *And, define the* compressed $\sigma$-Laplacian spectral algorithm *for detection as in Definition 2.2, only with* $\boldsymbol{L}$ *replaced by* $\widetilde{\boldsymbol{L}}$. *For recovery, use* $\boldsymbol{V}\boldsymbol{v}_1(\widetilde{\boldsymbol{L}}_\sigma(\widehat{\boldsymbol{Y}}))$.

We expect all results given below for compressed $\sigma$-Laplacians to hold as well for the original $\sigma$-Laplacians; this change is almost certainly merely a theoretical convenience. The simple idea behind these algorithms is that, if $\boldsymbol{x}$ is biased in the $\boldsymbol{1}$ direction, then

$$\widehat{\boldsymbol{Y}}\boldsymbol{1} = \beta\langle\boldsymbol{x}, \boldsymbol{1}\rangle\boldsymbol{x} + \widehat{\boldsymbol{W}}\boldsymbol{1}$$

will be somewhat correlated with $\boldsymbol{x}$. For the models of Definition 1.1, $\widehat{\boldsymbol{W}}\boldsymbol{1}$ will further be a standard Gaussian random vector (up to a negligible adjustment). In particular, if $\sigma$ is monotone, then $\boldsymbol{D}$ will become larger entrywise as $\beta$ increases, and will have larger diagonal entries in the coordinates where $\boldsymbol{x}$ is larger.

While it is tempting to dispense with $\sigma$ entirely, we will see below that we have $\|\widehat{\boldsymbol{Y}}\| = O(1)$, while standard asymptotics about the maximum of independent Gaussian random variables $\max_{i=1}^n |(\widehat{\boldsymbol{W}}\boldsymbol{1})_i| = \Omega(\sqrt{\log n})$. So, we must "tame" the largest entries of $\boldsymbol{D}$ by applying $\sigma$ so that it can "cooperate" with $\widehat{\boldsymbol{Y}}$ in determining the largest eigenvalue of $\boldsymbol{L}$ rather than dominating $\widehat{\boldsymbol{Y}}$.

For further intuition about the $\boldsymbol{D}$ term, note that if $\widehat{\boldsymbol{Y}}$ is a normalized adjacency matrix of a graph, then the vector $\widehat{\boldsymbol{Y}}\boldsymbol{1}$ contains normalized and centered degrees of each vertex in the graph. If a random graph is deformed to have a planted clique (see Appendix D.2) or an unusually dense subgraph, then this degree vector carries some information about which vertices belong to this planted structure. As we discuss in Appendix A, both spectral and degree-based algorithms have been studied before for such problems, and the $\sigma$-Laplacians describe a simple and tunable family of "hybrid" algorithms involving both kinds of information. This example is also why we call $\boldsymbol{L}$ a "Laplacian," since its definition resembles that of the graph Laplacian, the difference of the (unnormalized) diagonal degree and adjacency matrices of a graph.

Our original motivation, which we discuss in greater detail in Appendix F.4 (see also Appendix A for general discussion of related work), was to study a broad class of spectral algorithms where one performs PCA on $M(\boldsymbol{Y})$ for some function $M : \mathbb{R}^{n \times n}_{\mathrm{sym}} \to \mathbb{R}^{n \times n}_{\mathrm{sym}}$. It is reasonable to parametrize such $M(\boldsymbol{Y})$ as a neural network, alternating linear maps and entrywise nonlinearities. Subject to the natural criteria of *equivariance* and *dimension generalization* of $M$ that we explain in the Appendix, such $M(\boldsymbol{Y})$ are closely related to the much-studied *graph neural networks*, and the space of permissible linear maps to use is actually quite small, including the function $\boldsymbol{Y} \mapsto \mathrm{diag}(\boldsymbol{Y}\boldsymbol{1})$ that appears in nonlinear Laplacians. To perform random matrix analysis at the level of detail that we do here, one must be careful to make sure that the spectrum of $M(\boldsymbol{Y})$ remains on a fixed scale as one applies these transformations. This can easily break down if one allows others of the available linear functions in $M(\boldsymbol{Y})$, such as functions with rank-one outputs like $\boldsymbol{Y} \mapsto \boldsymbol{1}(\boldsymbol{Y}\boldsymbol{1})^\top$. We have found this issue to be quite delicate, so, as a first step, we have focused on nonlinear Laplacians as one special case of the above general class of flexible spectral algorithms, which do allow for sufficient control of the spectrum to study limiting empirical spectral distributions and outlier eigenvalues in detail.

# 3 Characterization of outlier eigenvalues

Our main results characterize when a $\sigma$-Laplacian matrix has an outlier eigenvalue. Let us first quickly recall the corresponding results for the case $\sigma = 0$.

## 3.1 Prior work: $\sigma = 0$ and direct spectral algorithms

In this case, when $\beta = 0$, $\widehat{Y} = \widehat{W}$ is just a normalized Wigner matrix,[3] and it is a classical result of random matrix theory that its eigenvalues follow Wigner's *semicircle law* $\mu_{\text{sc}}$, with high probability lying in the interval $[-2 - o(1), 2 + o(1)]$ (Theorem B.12 in Appendix). When $\beta > 0$, $\widehat{Y}$ consists of a rank-one perturbation of a Wigner matrix. Such random matrix distributions are commonly referred to as *spiked matrix models* and have been studied extensively in high-dimensional statistics and random matrix theory [Joh01, BBAP05, Pau07, FP07, CDMF09, PWBM18, EAKJ20, LM19]. The bulk eigenvalues remain stable under this rank-one perturbation (Proposition B.17 in Appendix) and still obey the semicircle law. However, when the signal-to-noise ratio $\beta$ exceeds a certain threshold, the rank-one perturbation induces a single outlier eigenvalue outside of the bulk. This sharp phase transition in the behavior of the largest eigenvalue of $\widehat{Y}$ is known as a *Baik–Ben Arous–Péché (BBP)* transition, named after the work of [BBAP05]. In this setting, it takes the following form:

**Theorem 3.1** ([FP07]). *Consider a symmetric random matrix $Y = W + \beta\sqrt{n} \cdot xx^\top \in \mathbb{R}_{\text{sym}}^{n \times n}$ as above, where $\beta > 0$, $x$ is a unit vector and $W$ is a GOE random matrix independent of $x$. Then the following hold for the largest eigenvalue $\lambda_1(\widehat{Y})$ and the corresponding unit eigenvector*

- *If $\beta \leq 1$, then $\lambda_1(\widehat{Y}) \xrightarrow{(p)} 2$ and $|\langle v_1(\widehat{Y}), x \rangle| \xrightarrow{(p)} 0$ (the arrows denoting convergence in probability).*

- *If $\beta > 1$, then $\lambda_1(\widehat{Y}) \xrightarrow{(p)} \beta + 1/\beta > 2$ and $|\langle v_1(\widehat{Y}), x \rangle|^2 \xrightarrow{(p)} 1 - 1/\beta^2 > 0$.*

For models as in Definition 1.1, this result implies the following analysis of direct spectral algorithms:

**Corollary 3.2.** *In a model of Sparse Biased PCA as in Definition 1.1, a direct spectral algorithm (the algorithm of Definition 2.2 with $\sigma = 0$) achieves strong detection and weak recovery if and only if $\beta > \beta_*(0) := 1$.*

## 3.2 Our contribution: $\sigma \neq 0$ and nonlinear Laplacian spectral algorithms

We now present our results, generalizing part of Theorem 3.1 to (compressed) nonlinear Laplacian matrices. We always make the following assumptions on the nonlinearity $\sigma$ without further mention.

**Assumption 1** (Properties of $\sigma$). *We assume that:*

1. *$\sigma$ is monotonically non-decreasing.*

2. *$\sigma$ is bounded: $|\sigma(x)| \leq K$ for some $K > 0$ and all $x \in \mathbb{R}$.*

3. *$\sigma$ is $\ell$-Lipschitz for some $\ell > 0$.*

We write $\text{edge}^+(\sigma) := \sup_{x \in \mathbb{R}} \sigma(x)$, which is finite by the second assumption, and $\sigma(\mathbb{R})$ for the image of $\sigma$, which is an interval (open or closed on either side) of $\mathbb{R}$ of finite length by the first two assumptions.

For now we give just the final result of our analysis, and describe the idea of the derivation below. Our main result is as follows:

**Theorem 3.3.** *For a model of Sparse Biased PCA as in Definition 1.1, define*

$$m_1 := \mathop{\mathbb{E}}_{x \sim \eta} x > 0, \quad m_2 := \mathop{\mathbb{E}}_{x \sim \eta} x^2.$$

*Given $\sigma$, define $\theta = \theta_\sigma(\beta)$ to solve the equation*

$$\mathop{\mathbb{E}}_{\substack{y \sim \eta \\ g \sim \mathcal{N}(0,1)}} \left[ \frac{y^2}{\theta - \sigma(\frac{m_1}{m_2}\beta y + g)} \right] = \frac{m_2}{\beta}$$

---

[3]That is, a symmetric random matrix with i.i.d. entries above the diagonal; conventions vary for the diagonal entries, but, as we have mentioned, this choice does not affect the results we will discuss.

*if such $\theta > \mathrm{edge}^+(\sigma)$ exists, and $\theta = \mathrm{edge}^+(\sigma)$ otherwise. The following hold almost surely for the sequence of compressed $\sigma$-Laplacians $\widetilde{\boldsymbol{L}} = \widetilde{\boldsymbol{L}}^{(n)}$:*

- *If $\mathbb{E}_{g \sim \mathcal{N}(0,1)} \left[ \frac{1}{(\theta_\sigma(\beta) - \sigma(g))^2} \right] \geq 1$, then*

$$\lambda_1(\widetilde{\boldsymbol{L}}^{(n)}) \to \mathrm{edge}^+(\mu_{\mathrm{sc}} \boxplus \sigma(\mathcal{N}(0,1))),$$

  *the right boundary point of the support of the probability measure $\mu_{\mathrm{sc}} \boxplus \sigma(\mathcal{N}(0,1))$. Here $\mu_{\mathrm{sc}}$ is Wigner's semicircle law, $\boxplus$ is the additive free convolution operation presented in Appendix B.4, and $\sigma(\mathcal{N}(0,1))$ is the pushforward of the standard Gaussian by $\sigma$. Moreover,*

$$|\langle \boldsymbol{x}, \boldsymbol{V}\boldsymbol{v}_1(\widetilde{\boldsymbol{L}}^{(n)})\rangle| \to 0.$$

- *If $\mathbb{E}_{g \sim \mathcal{N}(0,1)} \left[ \frac{1}{(\theta_\sigma(\beta) - \sigma(g))^2} \right] < 1$, then*

$$\lambda_1(\widetilde{\boldsymbol{L}}^{(n)}) \to \theta_\sigma(\beta) + \mathop{\mathbb{E}}_{g \sim \mathcal{N}(0,1)} \left[ \frac{1}{\theta_\sigma(\beta) - \sigma(g)} \right] > \mathrm{edge}^+(\mu_{\mathrm{sc}} \boxplus \sigma(\mathcal{N}(0,1))),$$

$$|\langle \boldsymbol{x}, \boldsymbol{V}\boldsymbol{v}_1(\widetilde{\boldsymbol{L}}^{(n)})\rangle|^2 \to \frac{m_2}{\beta^2} \left( \mathop{\mathbb{E}}_{\substack{y \sim \eta \\ g \sim \mathcal{N}(0,1)}} \left[ \frac{y^2}{\left(\theta_\sigma(\beta) - \sigma\left(\frac{m_1}{m_2}\beta y + g\right)\right)^2} \right] \right)^{-1}$$
$$\left( 1 - \mathop{\mathbb{E}}_{g \sim \mathcal{N}(0,1)} \left[ \frac{1}{\left(\theta_\sigma(\beta) - \sigma(g)\right)^2} \right] \right) > 0.$$

In the special case where the entrywise condition $x_i \geq 0$ holds almost surely (i.e., $\eta$ in Definition 1.1 is a probability measure on $\mathbb{R}_{\geq 0}$), there is a unique $\beta_* = \beta_*(\sigma) > 0$ that solves

$$\mathop{\mathbb{E}}_{g \sim \mathcal{N}(0,1)} \left[ \frac{1}{(\theta_\sigma(\beta_*) - \sigma(g))^2} \right] = 1,$$

and the conditions of the two cases above are equivalent to $\beta \leq \beta_*$ and $\beta > \beta_*$, respectively. In that case, this result precisely identifies the critical signal strength $\beta_*(\sigma)$ mentioned earlier, the threshold beyond which the $\sigma$-Laplacian has an outlier eigenvalue.[4] One may also check that, setting $\sigma = 0$ and using that in this case $\mathrm{edge}^+(\mu_{\mathrm{sc}} \boxplus \sigma(\mathcal{N}(0,1))) = \mathrm{edge}^+(\mu_{\mathrm{sc}}) = 2$, this result is indeed compatible with Theorem 3.1, giving $\beta_*(0) = 1$.

### 3.3 Example: Gaussian Planted Submatrix and Planted Clique models

We demonstrate the concrete consequence of Theorem 3.3 for the model proposed in Example 1.2 above. This is conditional on the accuracy of the numerical evaluation of the Gaussian expectations appearing in the Theorem. These involve only low-dimensional function and integral evaluations and we are confident that our numerical solutions are accurate, but we mark the following result with [(n)] to indicate its mild conditional nature.

**Theorem 3.4** (Gaussian Planted Submatrix [(n)]). *There exist $\sigma : \mathbb{R} \to \mathbb{R}$ and $\tau \in \mathbb{R}$ such that the following holds for the choices of Example 1.2 substituted into the setting of Definition 1.1. If $\beta > 0.76$ and $\boldsymbol{Y} \sim \mathbb{P}_{n,\beta}$, then the compressed $\sigma$-Laplacian $\widetilde{\boldsymbol{L}}_\sigma(\widehat{\boldsymbol{Y}})$ with high probability has a single outlier eigenvalue (see Figure 1 for an illustration and Appendix B.3 for precise definitions), and $|\langle \boldsymbol{x}, \boldsymbol{V}\boldsymbol{v}_1(\widetilde{\boldsymbol{L}}_\sigma(\widehat{\boldsymbol{Y}}))\rangle|$ converges in probability to a strictly positive deterministic number. In particular, if $\beta > 0.76$, then the compressed $\sigma$-Laplacian spectral algorithm with threshold $\tau$ succeeds in strong detection and weak recovery in the Gaussian Planted Submatrix model.*

Underlying this result is a choice of $\sigma$ for which $\beta_*(\sigma) < 0.76$; we discuss below in Section 4 various ways one can find $\sigma$ achieving the above, and illustrate one such $\sigma$ in Figure 1.

---

[4]For general $\eta$, our results do allow for the possibility that, as $\beta$ increases, an outlier eigenvalue first appears, then disappears, then appears again, and so forth. We expect this not to happen for most choices of $\eta$ and $\sigma$, but we leave it to future work to understand when (if ever) this more complicated situation arises. See Appendix C.4.2 for some more discussion of similar issues in prior work.

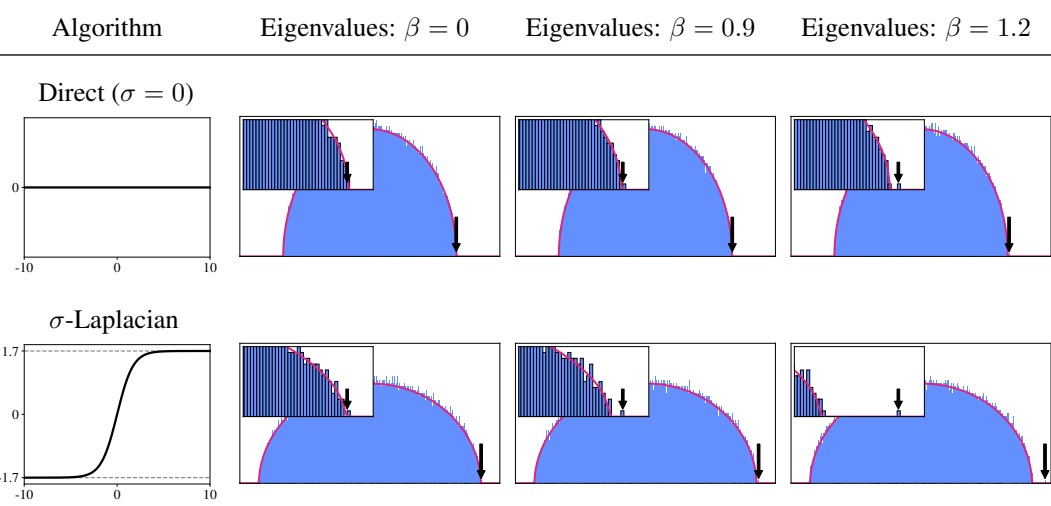

| Algorithm | Eigenvalues: $\beta = 0$ | Eigenvalues: $\beta = 0.9$ | Eigenvalues: $\beta = 1.2$ |
|---|---|---|---|
| Direct ($\sigma = 0$) | | | |
| $\sigma$-Laplacian | | | |

Figure 1: A comparison of the eigenvalues of the matrices used by a direct spectral algorithm and a $\sigma$-Laplacian spectral algorithm for the Gaussian Planted Submatrix problem. The left column shows the $\sigma$ used (which is zero for the direct algorithm). The other columns show the empirical eigenvalue distributions for various signal strengths $\beta$. The maximum eigenvalue is indicated with an arrow to highlight outliers, and the analytic prediction of the limiting eigenvalue density is plotted in red.

There are two reasonable benchmarks with which to compare this performance. On the one hand, $\beta_*(0) = 1$ is the corresponding threshold for the direct spectral algorithm, per Theorem 3.1. In words, our result says that only 76% as strong of a signal is required by a suitable nonlinear Laplacian to achieve strong detection. On the other hand, the work of [HWX17] shows that a belief propagation (BP) algorithm achieves weak recovery provided $\beta > 1/\sqrt{e} \approx 0.61$. In this setting of dense input data, BP is likely also to behave similarly to approximate message passing (AMP), an approximation that is more efficient to compute. Thus the performance of our nonlinear Laplacian algorithm lies between that of the direct spectral algorithm and BP/AMP, while our algorithm enjoys the advantages of being conceptually simpler than BP/AMP and only making a small modification to the direct spectral algorithm. (See Appendix A for a more detailed comparison with BP/AMP.)

In the same vein, we may also better understand the individual power of the two components of any $\sigma$-Laplacian, and show that they *must* be combined in order to achieve the above performance: neither the eigenvalues of $\widehat{Y}$ nor the values of $\widehat{Y}\mathbf{1}$ alone can achieve strong detection for any $\beta < 1$.

**Theorem 3.5.** *The following hold in the Gaussian Planted Submatrix model:*

1. *If $\beta < 1$, then there is no function of the vector $(\lambda_1(\widehat{Y}), \ldots, \lambda_n(\widehat{Y}))$ that achieves strong detection. (This result is due to prior work of [MRZ15].)*

2. *For any $\beta \geq 0$ (not depending on $n$), there is no function of the vector $\widehat{Y}\mathbf{1}$ that achieves strong detection. (This result is our contribution, which we prove in greater generality than just the Gaussian Planted Submatrix model; see Theorem C.10.)*

It is maybe surprising that the information contained in $\widehat{Y}\mathbf{1}$, which by itself is useless for detection in this regime, is enough to "boost" the performance of a spectral algorithm substantially. The question of how effective "purely spectral" algorithms can be for (weak) recovery is raised by [HWX17] (their Section 1.3), asking whether the direct spectral algorithm's $\beta_* = 1$ threshold is optimal in this regard. Our results suggest that only a small step beyond algorithms using only the eigenvalues of $\widehat{Y}$ is enough to improve on this.

Finally, we offer a more speculative extension. As we discuss in Appendix D.2, from the point of view of the random matrix theory of $\sigma$-Laplacians, the much-studied Planted Clique problem looks nearly identical to the Gaussian Planted Submatrix problem. We define this problem formally in Definition D.1, but, in words, it is given by taking $Y$ to be a centered adjacency matrix of an

Erdős-Rényi random graph with each edge present independently with probability $1/2$, with a clique (complete subgraph) inserted on a random subset of $\beta\sqrt{n}$ vertices. Replacing the Gaussian structure with discrete structure creates technical challenges that we have not been able to surmount. We are quite confident, but leave as an open problem to show, that the above results apply directly to the Planted Clique problem.

**Conjecture 3.6.** *The results of Theorem 3.4 hold verbatim if the Gaussian Planted Submatrix problem is replaced by the Planted Clique problem.*

See Appendix D.3 for discussion of further examples and extensions.

### 3.4 Proof techniques

We now sketch the analysis leading to Theorem 3.3. Full proofs are given in Appendix C. Recall that we are interested $\widehat{\boldsymbol{Y}} = \widehat{\boldsymbol{W}} + \beta\boldsymbol{x}\boldsymbol{x}^\top$, where $\widehat{\boldsymbol{W}}$ is a Wigner random matrix with entrywise variance $1/n$ and $\boldsymbol{x}$ is a unit vector. Consequently, the matrix $\boldsymbol{L}$ can be expressed as

$$\boldsymbol{L} = \widehat{\boldsymbol{W}} + \underbrace{\beta\boldsymbol{x}\boldsymbol{x}^\top + \operatorname{diag}(\sigma(\widehat{\boldsymbol{W}}\mathbf{1} + \beta\langle\boldsymbol{x}, \mathbf{1}\rangle\boldsymbol{x}))}_{=:\boldsymbol{X}}, \tag{1}$$

which we interpret as a perturbation of the Wigner noise $\widehat{\boldsymbol{W}}$ by a matrix $\boldsymbol{X}$.

If $\sigma = 0$, the perturbation term is simply $\boldsymbol{X} = \beta\boldsymbol{x}\boldsymbol{x}^\top$, and in particular is low-rank. In that case, the bulk eigenvalue distribution of $\boldsymbol{L}$ is always the same as that of $\widehat{\boldsymbol{W}}$, obeying the semicircle law. The effect of $\boldsymbol{X}$ in such models is limited to creating potential outlier eigenvalues, leaving the bulk spectrum unchanged. Our setting of $\sigma \neq 0$ presents a key difference, stemming from the fact that our $\boldsymbol{X}$ is (usually) full-rank, even when $\beta = 0$. Therefore, even when $\beta = 0$, the spectrum of $\boldsymbol{L}$ undergoes a non-trivial deformation from that of $\widehat{\boldsymbol{W}}$, which is described by free probability theory (specifically, by the operation of additive free convolution appearing in Theorem 3.3). When $\beta > 0$, the bulk eigenvalues will resemble this same deformation, and may have a further outlier eigenvalue generated by a corresponding outlier eigenvalue of $\boldsymbol{X}$. Such results have been obtained by [CDMFF11, Cap17, BG24], which our analysis applies.

Those results, roughly speaking, give a recipe for deducing the behavior of the eigenvalues of $\boldsymbol{L}$ from those of $\boldsymbol{X}$; in particular, outlier eigenvalues in $\boldsymbol{L}$ arise from sufficiently extreme eigenvalues in $\boldsymbol{X}$. So, we proceed by characterizing the eigenvalues of $\boldsymbol{X}$. Notably, $\boldsymbol{X}$ itself resembles a spiked matrix model, although one where the "noise term" is a diagonal matrix, making the analysis different than that for conventional spiked matrix models. The eigenvalues of $\boldsymbol{X}$ are as follows:

**Lemma 3.7.** *In the setting of Theorem 3.3, the following hold almost surely for the sequence of $\boldsymbol{X} = \boldsymbol{X}^{(n)}$:*

1. *The empirical spectral distribution satisfies $\frac{1}{n}\sum_i \delta_{\lambda_i(\boldsymbol{X}^{(n)})} \xrightarrow{\text{(w)}} \sigma(\mathcal{N}(0,1))$, where the arrow denotes weak convergence (Definition B.1 in Appendix).*

2. *The largest eigenvalue of $\boldsymbol{X}^{(n)}$ satisfies $\lambda_1(\boldsymbol{X}^{(n)}) \to \theta_\sigma(\beta)$ for the function $\theta_\sigma$ described in Theorem 3.3.*

3. *All other eigenvalues $\lambda_2(\boldsymbol{X}^{(n)}), \ldots, \lambda_n(\boldsymbol{X}^{(n)})$ lie in $\overline{\sigma(\mathbb{R})}$, where the bar denotes the closure.*

With this understanding, the eigenvalues of $\boldsymbol{L}$ can be effectively described using the above tools, provided that we make the adjustment from Definition 2.3. The reason for this is that $\widehat{\boldsymbol{W}}$ is weakly dependent on $\boldsymbol{X}$, as $\boldsymbol{X}$ depends on $\widehat{\boldsymbol{W}}\mathbf{1}$, while standard analysis from random matrix theory assumes these signal and noise matrices to be independent. When $\boldsymbol{W}$ is drawn from the GOE, we can circumvent this issue by instead analyzing the spectrum of the compressed $\sigma$-Laplacian $\widetilde{\boldsymbol{L}} = \widetilde{\boldsymbol{L}}^{(n)} = \boldsymbol{V}^\top \boldsymbol{L}\boldsymbol{V}$. By the rotational symmetry of the GOE, the noise term of $\widetilde{\boldsymbol{L}}$ remains a $(n-1) \times (n-1)$ GOE matrix, up to a negligible rescaling. And, this noise term has had the $\mathbf{1}$ direction "projected away," whereby, it is now independent of the projected signal term, making the model compatible with existing results.

To analyze the top eigenvector of $\widetilde{\boldsymbol{L}}^{(n)}$, we use a simple trick: if we replace the term $\beta\boldsymbol{x}\boldsymbol{x}^\top$ in the underlying $\boldsymbol{L}$ with $(\beta + t)\boldsymbol{x}\boldsymbol{x}^\top$ for another parameter $t$, then one may show that $\langle\boldsymbol{x}, \boldsymbol{V}\boldsymbol{v}_1(\widetilde{\boldsymbol{L}}^{(n)})\rangle^2$ is

precisely the derivative of $\lambda_1(\widetilde{\boldsymbol{L}}^{(n)})$ with respect to $t$ at $t = 0$. We argue that one may exchange this derivative with the limit $n \to \infty$, and thus $\langle \boldsymbol{x}, \boldsymbol{v}_1(\boldsymbol{L}) \rangle^2$ is obtained as a derivative of a closely related formula to that for $\lim_{n\to\infty} \lambda_1(\widetilde{\boldsymbol{L}}^{(n)})$.

As an aside, in addition to the analysis in Theorem 3.3 of the largest eigenvalue, we obtain the following result on the empirical spectral distribution of the $\sigma$-Laplacian, which is sensible given the meaning of the additive free convolution operation (see Definition B.20 in Appendix) and explains its appearance in Theorem 3.3:

**Lemma 3.8.** *For a model of Sparse Biased PCA as in Definition 1.1, for any $\sigma$ and $\beta \geq 0$, almost surely the empirical spectral distribution satisfies $\frac{1}{n} \sum_i \delta_{\lambda_i(\widetilde{\boldsymbol{L}}^{(n)})} \xrightarrow{\text{(w)}} \mu_{\mathrm{sc}} \boxplus \sigma(\mathcal{N}(0,1))$.*

This statement should be read as describing the bulk eigenvalues of $\widetilde{\boldsymbol{L}}$; recall that weak convergence does not give any guarantees about the behavior of extreme or outlier eigenvalues, so Theorem 3.3 indeed gives additional further information.

# 4 Numerical optimization of nonlinearities

Let us comment briefly on how we actually find the $\sigma$ and the number $0.76$ in Theorem 3.4. First, note that, given $\sigma$, in principle the above results determine $\beta_*(\sigma)$, albeit via an integral equation involving an expectation over $g \sim \mathcal{N}(0,1)$. Further, that equation is only given in terms of the function $\theta_\sigma(\beta)$, which itself is only given in terms of the solution of another integral equation involving an expectation over $g \sim \mathcal{N}(0,1)$ and $y \sim \eta$. This is why we point out above that our results only determine $\beta_*$ assuming the fidelity of the numerical calculations of these integrals (which are, however, only at most two-dimensional and thus not computationally challenging).

The question remains of how to find $\sigma$ that minimizes $\beta_*(\sigma)$. We have not been able to identify even a contrived construction of $\sigma \neq 0$ satisfying Assumption 1 for which we can find $\beta_*(\sigma)$ in closed form. So, we resort to heuristically identifying good $\sigma$ and then estimating $\beta_*(\sigma)$ for such $\sigma$ numerically. We have studied three approaches to this task, which all seem more or less equally effective:

1. Pick a simple class of $\sigma$ given by a small number of parameters, such as $\sigma(x) = a \tanh(bx)$ for $a, b \in \mathbb{R}$ or $\sigma(x) = \min\{c, \max\{d, ax+b\}\}$ for $a, b, c, d \in \mathbb{R}$ and optimize $\beta_*(\sigma)$ over these few parameters by manual inspection of numerical results or exhaustive grid search.

2. Fix a multi-layer perceptron (MLP) structure for $\sigma$ and optimize it by training $\lambda_1(\boldsymbol{L}_\sigma(\boldsymbol{Y}))$ to classify a training dataset of synthetic $\boldsymbol{Y}$ drawn from the null model ($\beta = 0$) and the structured alternative model ($\beta > 0$).

3. Fix a simple structure for $\sigma$ such as a step function[5] over a fixed grid and directly optimize the (complicated) objective function $\beta_*(\sigma)$ via gradient-free black-box optimization methods such as the Nelder-Mead or differential evolution algorithms.

We discuss the implementation details of these choices further in Appendix D.1.2. We conclude from these explorations that $\sigma$-Laplacian algorithms are rather robust to the choice of $\sigma$—just a few degrees of freedom in $\sigma$ appear to suffice to achieve optimal performance. We illustrate this in Figure 2, which gives the $\sigma$ obtained by each of the above methods. On the other hand, mathematically understanding the behavior of the equations determining $\beta_*(\sigma)$ seems quite challenging, and we leave this as an interesting problem for future work.

# 5 Conclusion

Above, we have introduced the new class of nonlinear Laplacian spectral algorithms, given a complete analysis of their performance for the task of strong detection in Sparse Biased PCA models, demonstrated as a consequence that such algorithms substantially outperform direct spectral algorithms for the Gaussian Planted Submatrix problem, and verified those findings empirically (see also Figures 3 and 4 in the Appendix for further experimental results).

---

[5]Strictly speaking a step function does not satisfy the Lipschitz condition of Assumption 1, but it is straightforward to show that an arbitrarily small smoothing $\sigma_\epsilon$ (say by convolution with a Gaussian of small width $\epsilon$) of such $\sigma$ does, and $\lim_{\epsilon \to 0} \beta_*(\sigma_\epsilon)$ recovers the value of $\beta_*(\sigma)$ computed for the step function.

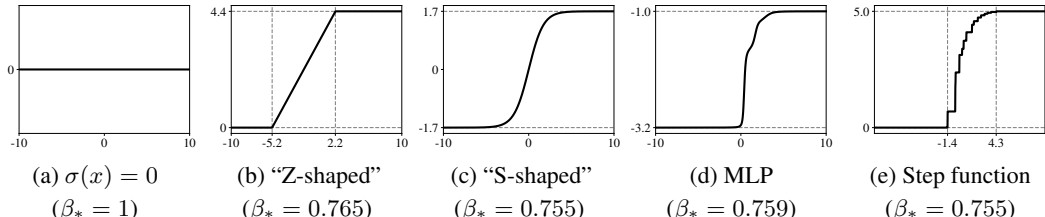

| (a) $\sigma(x) = 0$ | (b) "Z-shaped" | (c) "S-shaped" | (d) MLP | (e) Step function |
|---|---|---|---|---|
| ($\beta_* = 1$) | ($\beta_* = 0.765$) | ($\beta_* = 0.755$) | ($\beta_* = 0.759$) | ($\beta_* = 0.755$) |

Figure 2: Comparison of $\sigma$ obtained by various approaches for the Gaussian Planted Submatrix model: we illustrate $\sigma$ optimized over small function classes (b–c), learned from data using a multi-layer perceptron (MLP) structure (d), and obtained from black-box optimization using a step function structure (e). The corresponding value of $\beta_*(\sigma)$ is given below each.

**Limitations**    We mention three limitations of our work. First, as mentioned, algorithms like BP and AMP can perform better than nonlinear Laplacian algorithms for specific problems, for example as shown for the Gaussian Planted Submatrix problem by [HWX17]. We claim that nonlinear Laplacian algorithms, however, are conceptually simpler than BP/AMP and also easier to tune: our results indicate that one can either tune them mechanically from data or merely "eyeball" a reasonable nonlinearity to use. This is far from the case for AMP, where the iteration rules (including the subtle "Onsager correction") must be chosen carefully to yield a sensible limiting behavior. Second, we have made the assumption that our models involve only additive Gaussian noise (Definition 1.1), but we believe that the same analysis should apply to more general models (see Conjecture 3.6 and Appendix D.2). Finally, we leave open several challenging technical questions, including those of analyzing the algorithm's performance on the Planted Clique problem (Conjecture 3.6) and of understanding analytically the behavior of the threshold value $\beta_*(\sigma)$ and the structure of the optimal nonlinearity $\sigma$ for a given problem.

**Future directions**    It is tempting to consider designing more complex diagonal matrices $D = D(\widehat{Y})$ with which to augment spectral algorithms. One may, for instance, use a graph neural network to map the matrix $\widehat{Y}$ to a vector (to set as the diagonal of $D$) in an equivariant way and optimize this over a dataset (as in our treatment of building $\sigma$ with a multi-layer perceptron). See Section F.4 for more discussion of such an approach; here we only mention a few salient considerations. Firstly, if $D$ can depend on $\widehat{Y}$ more strongly than merely through $\widehat{Y}\mathbf{1}$, then the device of compression that we have used to decouple $D$ from $\widetilde{W}$ may break down and the analysis could become more challenging; for sufficiently complex $D$, the entire apparatus of free probability (which plays a crucial role in the results of [CDMFF11] that we use) might no longer apply, which would leave us with random matrices requiring a fundamentally different toolkit to analyze. Also, allowing very complex $D$ might allow one to build a complex algorithm solving the underlying statistical problem into this diagonal matrix alone, making the first term of the nonlinear Laplacian $L = \widehat{Y} + D$ superfluous. As we have argued, the merit of nonlinear Laplacians is in their balance of simplicity and strong performance, and we believe it would be valuable to understand more precisely the tradeoff between these properties as one blends more and more complex side information into spectral algorithms.

## Acknowledgments and Disclosure of Funding

Thanks to Benjamin McKenna and Cristopher Moore for helpful discussions in the course of this project, and to the anonymous reviewers for several useful suggestions, in particular for the idea behind the analysis of the top eigenvector in Theorem 3.3. YM was partially supported by NSF grant CCF-2430292.

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

## A   Related work

**Spiked matrix models**   Spiked matrix models have a long history in random matrix theory and statistics since their introduction in the work of Johnstone [Joh01]. The BBP transition was first predicted there and proved by [BBAP05], though in a different setting than Theorem 3.1 concerning sample covariance matrices rather than additive noise models. A useful general mathematical reference is the Habilitation à Diriger des Recherches of Capitaine [Cap17], while a statistical survey is given in [JP18]. In addition to the citations in the main text, let us emphasize that the problem of how high-rank signals (rather than finite or low-rank, as the above references considered) interact with additive noise has been studied at length recently and leads to many intriguing interactions with free probability theory with many remaining open problems. See, for example, [DS07, CDMFF11, Cap14, Cap17, McK21, BG24].

**Modified Laplacians**   The idea of modifying the graph Laplacian to solve various inference problems on graphs is not new, but seems mostly to have been explored in the context of community detection problems where the structure planted in a graph does not change its overall density, but rather only the relative density of connections within and between groups, as in the much-studied stochastic block model [Abb17, Moo17]. In this literature, modifications of the Laplacian motivated by the non-backtracking adjacency matrix and the Ihara-Bass formula concerning its eigenvalues have proved useful [KMM+13, BMNZ14, BLM15]. Another approach, more similar to our construction,

considers the *signless Laplacian* or the *sum* rather than difference of the degree and adjacency matrices, used for instance for community detection by [SD11]. However, unlike nonlinear Laplacians, these are all still only *linear* combinations of the adjacency and diagonal degree matrices of a graph. Finally, this general approach has some similarities to the idea of [MTV22] to combine linear and spectral estimators for estimation of generalized linear models; however, that work considers computing two estimators separately and then combining them, while we use the idea of the degree-based algorithm to modify the *input* to the spectral algorithm.

**Nonlinear PCA**    The idea of applying nonlinearities as a preprocessing step to improve the performance of direct spectral algorithms has also appeared before. The work of [LKZ15, PWBM18] considered spiked matrix models with non-Gaussian noise (or, in the former case, more general non-additive noisy observation channels), and showed that in these cases the performance of a direct spectral algorithm can be improved by first applying an entrywise nonlinearity to $\boldsymbol{Y}$. However, our method is again different because we consider applying such a nonlinearity only to the diagonal matrix $\boldsymbol{D}$ (which is not included at all in the above algorithms). It would be natural to combine the two methods by applying some other entrywise nonlinearity to $\boldsymbol{Y}$ in addition to applying $\sigma$ to the diagonal part of a nonlinear Laplacian, but this would complicate the analysis, and, based on the observations of the above works that this entrywise nonlinearity is *not* helpful under Gaussian noise, it seems unlikely that this would be useful for the Gaussian problems we describe in Definition 1.1. The line of work [LAL19, LL20, MKLZ22] explored using spectral algorithms with a general nonlinearity $\sigma$ for the phase retrieval problem; the settings are somewhat similar, but in the case of phase retrieval the description of how well a given $\sigma$ performs is simpler and in fact the optimal $\sigma$ can be identified in closed form.

**Degree-based algorithms**    In the Planted Clique model, it has been observed before that computing the degree of each node (and in particular the maximum degree over all nodes) is sufficient to detect a clique of size $O(\sqrt{n \log n})$ [Kuč95]. It is straightforward to show a similar phenomenon for the Gaussian Planted Submatrix model, which in that case takes the form of thresholding the largest entry of $\boldsymbol{Y}\mathbf{1}$ succeeding at strong detection once $\beta = \beta(n) \geq C\sqrt{\log n}$ for sufficiently large $n$.

**Power of restricted algorithms**    As we have mentioned, [MRZ15] showed that, in the Gaussian Planted Submatrix model, a direct spectral algorithm is optimal for strong detection (achieving the signal strength threshold $\beta_* = 1$) among algorithms that only examine the eigenvalues of $\boldsymbol{Y}$. This kind of claim was pursued in greater generality and detail by [BMV+18, PWBM18]. We show in our Theorem 3.5 that, not surprisingly, algorithms based only on the (analog of the) degree vector $\boldsymbol{Y}\mathbf{1}$ fare even worse, failing at strong detection for any constant $\beta$. Our Theorem 3.4 viewed in this light then is rather surprising: it shows that one may improve considerably on the direct spectral algorithm using only the information of the degree vector $\boldsymbol{Y}\mathbf{1}$, which on its own is useless for detection in this regime.

**Equivariant algorithms and graph neural networks (GNN)**    GNNs are a family of neural network architectures that are sensible to apply to graph data, which may be viewed as an adjacency matrix $\boldsymbol{Y}$ (possibly centered and/or normalized). Mathematically, they have the properties of *invariance* or *equivariance*: if $f : \mathbb{R}^{n \times n}_{\mathrm{sym}} \to \mathbb{R}$ as for a classification or detection problem then $f(\boldsymbol{PYP}^\top) = f(\boldsymbol{Y})$, while if $f : \mathbb{R}^{n \times n}_{\mathrm{sym}} \to \mathbb{R}^n$ as for an estimation or recovery problem then $f(\boldsymbol{PYP}^\top) = \boldsymbol{P}f(\boldsymbol{Y})$, for all permutation matrices $\boldsymbol{P}$. The algorithms we consider, which output the top eigenvalue or eigenvector of a $\sigma$-Laplacian, have these properties, respectively. Following the standard principles of neural network design, it seems reasonable to construct more general learnable spectral algorithms by combining linear equivariant layers with entrywise nonlinearities. The possibilities of such architectures were characterized and studied by [MBHSL18] (see also results in a similar spirit of [MFSL19, KP19, PLK+23]). In Section F.4, we show how the much simpler family of $\sigma$-Laplacian spectral algorithms arises from trying to construct specifically *spectral* algorithms based on such an architecture, if one seeks to constrain these layers so as not to "blow up" the spectrum of the matrix being repeatedly transformed.

**Approximate message passing (AMP)**    AMP algorithms take the general form of a "nonlinear power method", similar to the power method one can use to estimate the leading eigenpair of a matrix (see, e.g., [BPW18, FVRS21] for this general perspective, as well as [CMW20] for the broader

family of "general first order methods"). In particular, if one expands the power method applied to the $\sigma$-Laplacian, the result somewhat resembles such an algorithm. AMP for Non-Negative PCA with dense signals was studied by [MR15], for the Gaussian Planted Submatrix problem a "dense BP" similar to (but less efficient than) AMP was studied by [HWX17], and the same for the Planted Clique problem by [DM15].

While they are statistically powerful and conjectured in many situations to perform optimally, AMP algorithms are more complex to specify, calibrate, and study than the simple kind of spectral algorithm we consider, and are known to be fragile to mismatches between the assumptions (on the distribution of $\boldsymbol{Y}$ and $\boldsymbol{x}$, in our setting) used to design the specific AMP algorithm and the actual distributions from which inputs are drawn. See, e.g., the discussion in [RSF19] as well as [CZK14] for further practical subtleties, and the results of [IS24, IS25], who demonstrate that AMP must be modified in order to obtain certain robustness guarantees. In contrast, spectral algorithms enjoy straightforward such guarantees by applying standard eigenvalue and eigenvector perturbation inequalities. We also remark that much research has focused on faster computation and approximation of the spectral decomposition and top eigenpair, and any such improvements immediately translate to speedups for spectral algorithms (modulo matching any special structural assumptions on the matrices involved). For example, the recent work [BBC+25] uses sketching to approximate top eigenvector computations for matrices that are too large to fit in memory. To the best of our knowledge, such large-scale execution of AMP or BP has not yet been explored.

Lastly, we note that nonlinear Laplacian spectral algorithms could also be used together with AMP: often for PCA problems the output of a direct spectral algorithm is used to initialize AMP, and the "warm start" this gives to AMP is important to its iterations converging to an informative fixed point. (This is not necessary for the theoretical analysis of the algorithms of [DM15, HWX17] that apply to our specific setting, but could still be used with them to improve the quantitative performance and rate of convergence.) This could be substituted with the output of a nonlinear Laplacian spectral algorithm, which, by having higher correlation with the signal and giving a "warmer start," should reduce the number of AMP iterations required to achieve a given quality of estimate.

**General-purpose non-negative PCA**    As we have discussed, our approach begins from a spectral algorithm for PCA, which may be viewed as solving

$$
\begin{aligned}
\text{maximize} \quad & \widehat{\boldsymbol{x}}^\top \boldsymbol{Y} \widehat{\boldsymbol{x}} \\
\text{subject to} \quad & \|\widehat{\boldsymbol{x}}\| = 1.
\end{aligned}
\tag{2}
$$

When $\boldsymbol{x} \geq \boldsymbol{0}$, i.e., $x_i \geq 0$ for all $i$, which is one way in which $\boldsymbol{x}$ can be biased toward the $\boldsymbol{1}$ direction (more restrictive than the general models we propose in Definition 1.1, which are only biased in this direction on average rather than entrywise), a natural choice is to instead solve

$$
\begin{aligned}
\text{maximize} \quad & \widehat{\boldsymbol{x}}^\top \boldsymbol{Y} \widehat{\boldsymbol{x}} \\
\text{subject to} \quad & \|\widehat{\boldsymbol{x}}\| = 1 \\
& \widehat{\boldsymbol{x}} \geq \boldsymbol{0}
\end{aligned}
\tag{3}
$$

Unfortunately, while (2) can be solved efficiently, (3) is NP-hard to solve in general [DKP02]. Various algorithmic approaches to approximating the above problem (not necessarily attached to a statistical setting or assumption) have been proposed, for instance including semidefinite programming relaxations [MR15, BKW22]. Generally, the above problem is an instance of optimization over the convex cone of *completely positive matrices*, and various convex optimization approaches have been studied in the optimization literature and are discussed by [DKP02] and further citations given there. We have not explored the performance of nonlinear Laplacian spectral algorithms outside of a statistical setting, but it seems plausible that they could also give a faster alternative to such convex optimization methods for optimization problems like (3).

# B    Technical preliminaries

## B.1    Notation

**Linear algebra**    For matrices, we use $\|\cdot\|$ to denote the spectral norm. For vectors, we use $\|\cdot\|$, $\|\cdot\|_1$, and $\|\cdot\|_\infty$ to denote the Euclidean ($\ell^2$) norm, $\ell_1$ norm, and $\ell_\infty$ norm, respectively. For a symmetric matrix $\boldsymbol{X} \in \mathbb{R}^{n \times n}_{\mathrm{sym}}$, we write $\lambda_1(\boldsymbol{X}) \geq \cdots \geq \lambda_n(\boldsymbol{X})$ for its ordered eigenvalues.

We use $\mathbf{1}$ to denote the all-ones vector. For an index set $S \subset [n]$, we write $\mathbf{1}_S$ for the indicator vector of $S$, i.e., $(\mathbf{1}_S)_i = \mathbb{1}\{i \in S\}$. For a vector $\boldsymbol{x}$, we use $\mathrm{diag}(\boldsymbol{x})$ to denote the diagonal matrix with diagonal entries given by $\boldsymbol{x}$.

Throughout the paper, we use $\widehat{\boldsymbol{X}}$ and $\widehat{\boldsymbol{x}}$ to denote the suitably normalized matrix and vector, respectively, where

$$\widehat{\boldsymbol{x}} := \frac{\boldsymbol{x}}{\|\boldsymbol{x}\|}, \quad \widehat{\boldsymbol{X}} := \frac{\boldsymbol{X}}{\sqrt{n}}.$$

**Analysis**  The asymptotic notations $O(\cdot), o(\cdot), \Omega(\cdot), \omega(\cdot), \Theta(\cdot)$ will have their usual meaning, always referring to the limit $n \to \infty$. We write $\delta_x$ for the Dirac delta measure at $x \in \mathbb{R}$, and use the linear combination of measures notation $\sum a_i \mu_i$ to denote a mixture of measures $\mu_i$ with weights $a_i$. For a vector $\boldsymbol{x} \in \mathbb{R}^n$, we use $\mathrm{ed}(\boldsymbol{x})$ to denote its empirical distribution: $\mathrm{ed}(\boldsymbol{x}) := \frac{1}{n} \sum_{i=1}^n \delta_{x_i}$. For a symmetric matrix $\boldsymbol{X} \in \mathbb{R}^{n \times n}_{\mathrm{sym}}$, we use $\mathrm{esd}(\boldsymbol{X})$ to denote its empirical spectral distribution: $\mathrm{esd}(\boldsymbol{X}) := \frac{1}{n} \sum_{i=1}^n \delta_{\lambda_i(\boldsymbol{X})}$. We use $\mu_n \xrightarrow{(\mathrm{w})} \mu$ to denote weak convergence of (probability) measures.

**Probability**  $X_n \xrightarrow{(\mathrm{p})} X$ and $X_n \xrightarrow{(\mathrm{a.s.})} X$ to denote convergence in probability and almost sure convergence of random variables, respectively. For an event $A$, we use $A^c$ to denote its complement. For a sequence of events $A(n)$, we define the following:

$$\{A(n) \text{ happens eventually always}\} := \liminf_{n\to\infty} A(n) = \bigcup_{N=1}^\infty \bigcap_{n \geq N} A(n),$$

$$\{A(n) \text{ happens infinitely often}\} := \limsup_{n\to\infty} A(n) = \bigcap_{N=1}^\infty \bigcup_{n \geq N} A(n).$$

Note that $\{A(n) \text{ happens eventually always}\}^c = \{A(n)^c \text{ happens infinitely often}\}$.

**Random matrices**  We use $\mathrm{GOE}(n, \sigma^2)$ to denote the Gaussian Orthogonal Ensemble (GOE) on $\mathbb{R}^{n \times n}_{\mathrm{sym}}$ with entrywise variance $\sigma^2$, the law of $\boldsymbol{X}$ having $X_{ij} = X_{ji} \sim \mathcal{N}(0, \sigma^2 \mathbb{1}\{i = j\})$ independently for all $i \leq j$. We use $\mathrm{Wig}(n, \nu)$ to denote the Wigner matrix distribution on $\mathbb{R}^{n \times n}_{\mathrm{sym}}$, with entries (up to symmetry) i.i.d. from the distribution $\nu$ on $\mathbb{R}$. We denote the Wigner semicircle law by $\mu_{\mathrm{sc}}$, which is the probability measure on $\mathbb{R}$ supported on $[-2, 2]$ with density $\frac{1}{2\pi}\sqrt{4 - x^2}$. We use $G(n, p)$ to denote the Erdős–Rényi random graph with $n$ vertices and edge probability $p$. We write $\mathrm{edge}^+(\sigma)$ for the right endpoint of the (closed) image $\overline{\sigma(\mathbb{R})}$. For a probability measure $\mu$ on $\mathbb{R}$, we write $\mathrm{supp}(\mu)$ for the support of $\mu$, and $\mathrm{edge}^+(\mu)$ for the rightmost point in $\mathrm{supp}(\mu)$. We denote by $G_\nu$ the Stieltjes transform of $\nu$, and by $R_\nu$ the $R$-transform of $\nu$. We use $\omega_{\mu_{\mathrm{sc}}, \nu}$ to denote the subordination function, and $H_\mu$ for its functional inverse. (See Section B.4 for these notions from free probability.)

## B.2 Probability tools

**Weak convergence**  Many of our results are phrased in terms of the weak convergence of probability measures, whose definition and basic properties we recall below.

**Definition B.1** (Weak convergence). *A sequence of probability measures $(\mu_n)_{n \geq 1}$ on $\mathbb{R}$ converges weakly to another probability measure $\mu$ on $\mathbb{R}$ if, for every bounded continuous function $f : \mathbb{R} \to \mathbb{R}$,*

$$\lim_{n\to\infty} \int_\mathbb{R} f \, d\mu_n = \int_\mathbb{R} f \, d\mu.$$

*In this case, we write $\mu_n \xrightarrow{(\mathrm{w})} \mu$.*

**Proposition B.2** (Weak convergence of empirical distribution). *Let $\mu$ be a probability measure on $\mathbb{R}$ and define a sequence of random vectors $\boldsymbol{x} = \boldsymbol{x}^{(n)}$ with $x_i^{(n)} \overset{\mathrm{i.i.d.}}{\sim} \mu$ for each $1 \leq i \leq n$. Then, almost surely, $\mathrm{ed}(\boldsymbol{x}^{(n)}) \xrightarrow{(\mathrm{w})} \mu$.*

*Proof.* Let $\mu_n := \mathrm{ed}(\boldsymbol{x}^{(n)})$. By the Portmanteau theorem, weak convergence is equivalent to

$$\mathbb{P}\left(\mu_n \xrightarrow{\text{(w)}} \mu\right) = \mathbb{P}\left(\left|\mu_n((-\infty, x]) - \mu((-\infty, x])\right| \to 0 \text{ for all } x \in \mathbb{R}\right)$$

$$\geq \mathbb{P}\left(\sup_{x \in \mathbb{R}} \left|\mu_n((-\infty, x]) - \mu((-\infty, x])\right| \to 0\right),$$

and the latter probability is 1 by the Glivenko-Cantelli theorem. $\qquad\square$

**Definition B.3** (Wasserstein distance)**.** *Define*
$$\mathcal{P}_1(\mathbb{R}) := \{\mu \text{ a probability measure on } \mathbb{R} \text{ with } \mathbb{E}_{X \sim \mu}\left[|X|\right] < \infty\}.$$

*Let $\Gamma(\mu, \nu)$ be the set of probability measures on $\mathbb{R}^2$ whose marginal on the first coordinate is $\mu$ and whose marginal on the second coordinate is $\nu$. The* Wasserstein distance *on the space $\mathcal{P}_1(\mathbb{R})$ is*
$$W_1(\mu, \nu) := \inf_{\gamma \in \Gamma(\mu, \nu)} \mathbb{E}_{(x,y) \sim \gamma} |x - y|.$$

**Lemma B.4** (Weak convergence and Wasserstein distance, Theorem 6.9 of [Vil09])**.** *Let $\mu_n, \mu \in \mathcal{P}_1(\mathbb{R})$. Then, $\mu_n \xrightarrow{\text{(w)}} \mu$ if and only if $W_1(\mu_n, \mu) \to 0$.*

**Proposition B.5** (Stability of weak convergence under perturbation)**.** *Consider sequences of vectors $\boldsymbol{x}^{(n)}, \boldsymbol{y}^{(n)} \in \mathbb{R}^n$ such that $\|\boldsymbol{x}^{(n)} - \boldsymbol{y}^{(n)}\|_\infty \to 0$ as $n \to \infty$. For $\mu \in \mathcal{P}_1(\mathbb{R})$, $\mathrm{ed}(\boldsymbol{x}^{(n)}) \xrightarrow{\text{(w)}} \mu$ if and only if $\mathrm{ed}(\boldsymbol{y}^{(n)}) \xrightarrow{\text{(w)}} \mu$.*

*Proof.* We will apply Lemma B.4. Since they are discrete probability measures, $\mathrm{ed}\left(\boldsymbol{x}^{(n)}\right), \mathrm{ed}\left(\boldsymbol{y}^{(n)}\right) \in \mathcal{P}_1(\mathbb{R})$ for all $n$. The Wasserstein distance between $\mathrm{ed}\left(\boldsymbol{x}^{(n)}\right)$ and $\mathrm{ed}\left(\boldsymbol{y}^{(n)}\right)$ is given by

$$W_1\left(\mathrm{ed}(\boldsymbol{x}^{(n)}), \mathrm{ed}(\boldsymbol{y}^{(n)})\right) = \inf_{\pi \in S_n} \frac{1}{n} \sum_{i=1}^n \left|x_i^{(n)} - y_{\pi(i)}^{(n)}\right|$$

$$\leq \frac{1}{n} \sum_{i=1}^n |x_i^{(n)} - y_i^{(n)}|$$

$$\leq \|\boldsymbol{x}^{(n)} - \boldsymbol{y}^{(n)}\|_\infty$$

$$\to 0.$$

Suppose $\mathrm{ed}\left(\boldsymbol{x}^{(n)}\right) \xrightarrow{\text{(w)}} \mu$. Since the Wasserstein distance is a metric satisfying the triangle inequality,

$$W_1\left(\mathrm{ed}(\boldsymbol{y}^{(n)}), \mu\right) \leq W_1\left(\mathrm{ed}(\boldsymbol{x}^{(n)}), \mu\right) + W_1\left(\mathrm{ed}(\boldsymbol{x}^{(n)}), \mathrm{ed}(\boldsymbol{y}^{(n)})\right) \to 0,$$

so $\mathrm{ed}\left(\boldsymbol{y}^{(n)}\right) \xrightarrow{\text{(w)}} \mu$. The converse follows in the same way. $\qquad\square$

**Almost sure convergence**  We also will use the following familiar results about almost sure convergence.

**Lemma B.6.** *For $s \sim \mathrm{Bin}(n, p)$, where $p = p(n) = \omega\left(\frac{\log n}{n}\right)$, we have*
$$\frac{s}{np} \xrightarrow{\text{(a.s.)}} 1.$$

*Proof.* Let $0 < \epsilon < 1$. Using the Chernoff inequality [Ver09, Corollary 2.3.4],
$$\mathbb{P}\left(\left|\frac{s}{np} - 1\right| > \epsilon\right) = \mathbb{P}\left(|s - np| > np\epsilon\right) \leq 2\exp\left(-\frac{\epsilon^2 np}{3}\right).$$

Since $p = \omega\left(\frac{\log n}{n}\right)$, we have $\exp\left(-\frac{\epsilon^2 np}{3}\right) < n^{-2}$ for all sufficiently large $n$. Hence, $\sum_{n=1}^\infty \exp\left(-\frac{\epsilon^2 np}{3}\right) < \infty$. By the Borel-Cantelli lemma, this shows that
$$\mathbb{P}\left(\left\{\left|\frac{s}{np} - 1\right| > \epsilon\right\} \text{ occurs infinitely often}\right) = 0. \qquad\square$$

**Lemma B.7.** *Let $\varepsilon_i \overset{\text{i.i.d.}}{\sim} \text{Ber}(p)$, where $p = p(n) = \omega\left(\frac{\log n}{n}\right)$. Let $x_i$ be i.i.d. random variables with mean $m = \mathbb{E}[x_i]$, and suppose $\mathbb{E}[\|x_i\|] < \infty$. Then*

$$\frac{1}{np} \sum_{i=1}^{n} \varepsilon_i x_i \xrightarrow{\text{(a.s.)}} m.$$

*Proof.* Using the triangle inequality, we have

$$\left| \frac{1}{np} \sum_{i=1}^{n} \varepsilon_i x_i - m \right| \leq \left| \frac{\sum_{i=1}^{n} \varepsilon_i x_i}{\sum_{i=1}^{n} \varepsilon_i} - m \right| + \left| \frac{\sum_{i=1}^{n} \varepsilon_i x_i}{\sum_{i=1}^{n} \varepsilon_i} \right| \cdot \left| \frac{\sum_{i=1}^{n} \varepsilon_i}{pn} - 1 \right|.$$

By Lemma B.6, we have $\frac{1}{np} \sum_{i=1}^{n} \varepsilon_i \xrightarrow{\text{(a.s.)}} 1$. Since $p = \omega(n^{-1})$, it follows that $\sum_{i=1}^{n} \varepsilon_i \xrightarrow{\text{(a.s.)}} \infty$. Conditional on the $\varepsilon_i$, $\sum_{i=1}^{n} \varepsilon_i x_i$ is a sum of $\sum_{i=1}^{n} \varepsilon_i$ many i.i.d. random variables. By our above observation, the number of terms in this sum almost surely diverges as $n \to \infty$. So, by the Strong Law of Large Numbers applied after conditioning on the $\varepsilon_i$, we get

$$\frac{\sum_{i=1}^{n} \varepsilon_i x_i}{\sum_{i=1}^{n} \varepsilon_i} \xrightarrow{\text{(a.s.)}} m.$$

Combining the results above, we conclude that $\left| \frac{1}{np} \sum_{i=1}^{n} \varepsilon_i x_i - m \right| \xrightarrow{\text{(a.s.)}} 0$. □

## B.3 Random matrix theory

We review the models of random matrices that will be relevant to us and some of their main properties.

**Definition B.8** (Gaussian orthogonal ensemble). *For $n \in \mathbb{N}$, we define the Gaussian orthogonal ensemble (GOE) distribution with variance $\sigma^2$, denoted as $\text{GOE}(n, \sigma^2)$ to be the law of a $n \times n$ symmetric matrix $\mathbf{W}$ with entries $W_{ij} = W_{ji} \sim \mathcal{N}\left(0, \sigma^2\right)$ for $i < j$ and $W_{ii} \sim \mathcal{N}\left(0, 2\sigma^2\right)$ for all $i$, with all entries with $i \leq j$ distributed independently.*

**Definition B.9** (Wigner matrix). *For $n \in \mathbb{N}$, we define the Wigner matrix distribution $\text{Wig}(n, \nu)$ to be the law of a $n \times n$ symmetric matrix $\mathbf{W}$ with entries $W_{ij} = W_{ji} \overset{\text{i.i.d.}}{\sim} \nu$ for $i < j$ and $W_{ii} := 0$ for all $i$. We say $\mathbf{W}$ is Wigner matrix with variance $\sigma^2$ if $\mathbf{W} \sim \text{Wig}(n, \nu)$ for some $\nu$ with zero expectation and variance $\sigma^2$.*

The following shows that the specific choice of distribution of the diagonal entries is not important.

**Proposition B.10.** *Consider random symmetric matrices $\widehat{\mathbf{W}}_0 = \widehat{\mathbf{W}}_0^{(n)}$ where $(\widehat{\mathbf{W}}_0)_{ij} = (\widehat{\mathbf{W}}_0)_{ji} \overset{\text{i.i.d.}}{\sim} \mathcal{N}\left(0, 1/n\right)$ for all $i \leq j$, and $\widehat{\mathbf{W}}_1 = \widehat{\mathbf{W}}_1^{(n)} \sim \text{GOE}(n, 1/n)$. Then, $\text{Law}(\widehat{\mathbf{W}}_1) = \text{Law}(\widehat{\mathbf{W}}_0 + \mathbf{\Delta})$ for a sequence of random matrices $\mathbf{\Delta} = \mathbf{\Delta}^{(n)}$ satisfying $\|\mathbf{\Delta}^{(n)}\| \xrightarrow{\text{(a.s.)}} 0$.*

*Proof.* Take $\mathbf{\Delta} = \text{diag}(\mathbf{g})$ where $\mathbf{g} \sim \mathcal{N}(0, \mathbf{I}_n/n)$ is independent of $\widehat{\mathbf{W}}_0$. Then, checking means and covariances shows that these $\mathbf{\Delta}$ satisfy the stated distributional equality, and $\|\mathbf{\Delta}\| = \|\mathbf{g}\|_\infty = O(\sqrt{(\log n)/n})$ almost surely by standard concentration results. □

In the introduction we have repeatedly mentioned the notion of "bulk" eigenvalues; let us be more specific about the meaning of this. For a deterministic sequence of symmetric matrices $\mathbf{X}^{(n)} \in \mathbb{R}^{n \times n}_{\text{sym}}$ with eigenvalues $\{\lambda_i(\mathbf{X}^{(n)})\}_{i=1}^{n}$, we say its bulk eigenvalues have distribution $\mu$ if its empirical spectral distribution

$$\text{esd}\left(\mathbf{X}^{(n)}\right) := \frac{1}{n} \sum_{i=1}^{n} \delta_{\lambda_i(\mathbf{X}^{(n)})} \xrightarrow{\text{(w)}} \mu.$$

Thus the bulk indicates where the vast majority of eigenvalues are located asymptotically, but does not describe the behavior of any $o(n)$ eigenvalues, and in particular of small numbers of outliers outside the support of $\mu$ (which in all cases we consider is compactly supported).

We will be interested in such notions for random matrices, in which case the empirical spectral distribution $\mathrm{esd}\left(\boldsymbol{X}^{(n)}\right)$ is itself a random measure. Therefore, it is necessary to specify a precise mode of weak convergence. In this work, we focus on the *almost sure weak convergence*, which is commonly studied in random matrix theory.

**Definition B.11** (Almost sure weak convergence). *Let $\boldsymbol{X}^{(n)} \in \mathbb{R}_{\mathrm{sym}}^{n \times n}$ be a sequence of random symmetric matrices. We say that the empirical spectral distributions $\mathrm{esd}(\boldsymbol{X}^{(n)})$ converge weakly almost surely to a deterministic probability measure $\mu$, denoted by $\mathrm{esd}(\boldsymbol{X}^{(n)}) \xrightarrow[\mathrm{a.s.}]{(\mathrm{w})} \mu$ if for all continuous functions $f : \mathbb{R} \to \mathbb{R}$, $\frac{1}{n} \sum_{i=1}^{n} f(\lambda_i(\boldsymbol{X}^{(n)})) \xrightarrow{(\mathrm{a.s.})} \int f(x) \, d\mu(x)$.*

The celebrated Wigner semicircle law establishes such almost sure weak convergence for the Wigner matrix model:

**Theorem B.12** (Wigner's semicircle limit theorem). *Let $\boldsymbol{W} \sim \mathrm{Wig}(n, \nu)$, where $\nu$ is a probability measure with zero mean and unit variance. Then $\mathrm{esd}(\boldsymbol{W}/\sqrt{n}) \xrightarrow[\mathrm{a.s.}]{(\mathrm{w})} \mu_{\mathrm{sc}}$ where $\mu_{\mathrm{sc}}$ is the semicircle measure on $\mathbb{R}$, supported on $[-2, 2]$ with density $\frac{1}{2\pi}\sqrt{4 - x^2}$ on that interval.*

In our work, for technical convenience, we will mostly consider a different notion: we say $\mathrm{esd}\left(\boldsymbol{X}^{(n)}\right) \xrightarrow{(\mathrm{w})} \mu$ almost surely if $\mathbb{P}(\mathrm{esd}\left(\boldsymbol{X}^{(n)}\right) \xrightarrow{(\mathrm{w})} \mu) = 1$. But in the situation of our concern, it is usually equivalent to the aforementioned notion.

**Proposition B.13.** *When $\mu$ is a continuous probability measure on $\mathbb{R}$, $\mathrm{esd}(\boldsymbol{X}^{(n)}) \xrightarrow{(\mathrm{w})} \mu$ almost surely if and only if $\mathrm{esd}(\boldsymbol{X}^{(n)}) \xrightarrow[\mathrm{a.s.}]{(\mathrm{w})} \mu$.*

*Proof.* Let $\mu^{(n)} := \mathrm{esd}(\boldsymbol{X}^{(n)})$. By [Tao12, Exercise 2.4.1], it is equivalent to prove the equivalence:

$$\mathbb{P}\left(\mu^{(n)}(-\infty, x] \to \mu(-\infty, x] \text{ for all } x \in \mathbb{R}\right) = 1$$

$$\Leftrightarrow \quad \text{for all } x \in \mathbb{R}, \ \mathbb{P}\left(\mu^{(n)}(-\infty, x] \to \mu(-\infty, x]\right) = 1.$$

The "$\Rightarrow$" direction is immediate. We will prove "$\Leftarrow$".

By the union bound and countable subadditivity,

$$\mathbb{P}\left(\exists x \in \mathbb{Q}, \ \mu^{(n)}(-\infty, x] \not\to \mu(-\infty, x]\right) \leq \sum_{x \in \mathbb{Q}} \mathbb{P}\left(\mu^{(n)}(-\infty, x]) \not\to \mu(-\infty, x]\right) = 0.$$

Hence, with probability 1, we have $\mu^{(n)}((-\infty, x]) \to \mu((-\infty, x])$ for all $x \in \mathbb{Q}$. It remains to show that this implies convergence for all $x \in \mathbb{R}$.

Fix $\epsilon > 0$ and let $x \in \mathbb{R}$. Choose rationals $q_1, q_2 \in \mathbb{Q}$ such that $q_1 < x < q_2$ and $\mu((q_1, q_2)) < \epsilon$. For sufficiently large $n$, we have:

$$\left|\mu^{(n)}(-\infty, q_1] - \mu(-\infty, q_1]\right| < \epsilon, \quad \text{and} \quad \left|\mu^{(n)}(-\infty, q_2] - \mu(-\infty, q_2]\right| < \epsilon.$$

Then by the monotonicity of $\mu^{(n)}$, we have:

$$\mu^{(n)}(-\infty, x] - \mu(-\infty, x] \leq \mu^{(n)}(-\infty, q_2] - \mu(-\infty, q_2] + \mu(x, q_2) < 2\epsilon,$$

$$\mu^{(n)}(-\infty, x] - \mu(-\infty, x] \geq \mu^{(n)}(-\infty, q_1] - \mu(-\infty, q_1] - \mu(q_1, x) > -2\epsilon.$$

Since $\epsilon > 0$ is arbitrary, it follows that

$$\mu^{(n)}(-\infty, x] \to \mu(-\infty, x] \quad \text{for all } x \in \mathbb{R}.$$

This concludes the proof. $\qquad\qquad\square$

The complementary notion to bulk eigenvalues is that of outlier eigenvalues, which we also mentioned earlier. While a statement like $\mathrm{esd}\left(\boldsymbol{X}^{(n)}\right) \xrightarrow[\mathrm{a.s.}]{(\mathrm{w})} \mu$ captures the asympotic distribution of the

eigenvalue bulk, it does not give any information on the extreme eigenvalues. We call the top eigenvalue $\lambda_1(\boldsymbol{X}^{(n)})$ an outlier eigenvalue if it asymptotically resides outside the bulk, i.e., if almost surely there exists $\epsilon > 0$ such that for sufficiently large $n$,

$$\lambda_1(\boldsymbol{X}^{(n)}) > \text{edge}^+(\mu) + \epsilon,$$

where $\text{edge}^+(\mu)$ denotes the right boundary point of the support of $\mu$.

The proposition below states that there is no outlier eigenvalue for Wigner random matrices (and thus for GOE matrices).

**Proposition B.14** (Largest eigenvalue of Wigner matrix [FK81, BY88]). *Let $\boldsymbol{W} \sim \text{Wig}(n, \nu)$, where $\nu$ is a probability measure with zero mean, unit variance, and finite fourth moment. Then,*
$\lambda_1(\boldsymbol{W}/\sqrt{n}) \xrightarrow{\text{(a.s.)}} 2.$

Finally, the following results about the stability of eigenvalues and empirical spectral distributions will be useful to handle small perturbations to the matrices we work with.

**Proposition B.15** (Weyl's inequality). *For $\boldsymbol{X}, \boldsymbol{Y} \in \mathbb{R}_{\text{sym}}^{n \times n}$, for any $i, j \in [n]$ where $i + j - 1 \leq n$,*

$$\lambda_{i+j-1}(\boldsymbol{X} + \boldsymbol{Y}) \leq \lambda_i(\boldsymbol{X}) + \lambda_j(\boldsymbol{Y}).$$

**Corollary B.16** (Weyl's interlacing inequality). *For $\boldsymbol{X}, \boldsymbol{\Delta} \in \mathbb{R}_{\text{sym}}^{n \times n}$,*

$$\lambda_{i+\sigma^-}(\boldsymbol{X}) \leq \lambda_i(\boldsymbol{X} + \boldsymbol{\Delta}) \leq \lambda_{i-\sigma^+}(\boldsymbol{X})$$

*where $\sigma^+, \sigma^-$ are the number of positive and negative eigenvalues of perturbation $\boldsymbol{\Delta}$.*

*Proof.* Applying Proposition B.15, get

$$\lambda_i(\boldsymbol{X} + \boldsymbol{\Delta}) \leq \lambda_{i-\sigma^+}(\boldsymbol{X}) + \underbrace{\lambda_{\sigma^++1}(\boldsymbol{\Delta})}_{\leq 0}$$

$$\lambda_{i+\sigma^-}(\boldsymbol{X}) \leq \lambda_i(\boldsymbol{X} + \boldsymbol{\Delta}) + \underbrace{\lambda_{\sigma^-+1}(-\boldsymbol{\Delta})}_{\leq 0},$$

which gives the result. $\qquad\square$

**Corollary B.17** (Stability of esd under low-rank perturbation). *Consider sequences of matrices $\boldsymbol{X} = \boldsymbol{X}^{(n)}, \boldsymbol{\Delta} = \boldsymbol{\Delta}^{(n)} \in \mathbb{R}_{\text{sym}}^{n \times n}$ where $\frac{\text{rank}(\boldsymbol{\Delta}^{(n)})}{n} \to 0$. For $\mu \in \mathcal{P}_1(\mathbb{R})$, i.e. $\mu$ is a probability measure with finite first moment, $\text{esd}(\boldsymbol{X}) \xrightarrow{(w)} \mu$ if and only if $\text{esd}(\boldsymbol{X} + \boldsymbol{\Delta}) \xrightarrow{(w)} \mu$.*

*Proof.* Let $k = k(n) := \text{rank}(\boldsymbol{\Delta}^{(n)})$. By Corollary B.16, $\lambda_{i+k}(\boldsymbol{X}) \leq \lambda_i(\boldsymbol{X} + \boldsymbol{\Delta}) \leq \lambda_{i-k}(\boldsymbol{X})$.

We would like to apply Lemma B.4. It is easy to check that $\text{esd}(\boldsymbol{X}), \text{esd}(\boldsymbol{X} + \boldsymbol{\Delta}) \in \mathcal{P}_1$.

The Wasserstein-1 distance between $\text{esd}(\boldsymbol{X})$ and $\text{esd}(\boldsymbol{X} + \boldsymbol{\Delta})$ is

$$W_1(\text{esd}(\boldsymbol{X}), \text{esd}(\boldsymbol{X} + \boldsymbol{\Delta})) = \inf_{\pi \in S_n} \frac{1}{n} \sum_{i=1}^n \left| \lambda_i(\boldsymbol{X} + \boldsymbol{\Delta}) - \lambda_{\pi(i)}(\boldsymbol{X}) \right|$$

$$\leq \frac{1}{n} \sum_{i=k+1}^{n-k} (\lambda_i(\boldsymbol{X} + \boldsymbol{\Delta}) - \lambda_{i+k}(\boldsymbol{X})) + O(kn^{-1})$$

$$\leq \frac{1}{n} \sum_{i=k+1}^{n-k} (\lambda_{i-k}(\boldsymbol{X}) - \lambda_{i+k}(\boldsymbol{X})) + O(kn^{-1})$$

$$= \frac{1}{n} \sum_{i=1}^{2k} \lambda_i(\boldsymbol{X}) - \frac{1}{n} \sum_{i=n-2k+1}^{n} \lambda_i(\boldsymbol{X}) + O(kn^{-1})$$

$$= O(kn^{-1})$$

Suppose $\text{esd}(\boldsymbol{X}) \xrightarrow{(w)} \mu$. By the triangle inequality,
$$W_1(\mu, \text{esd}(\boldsymbol{X} + \boldsymbol{\Delta})) \leq W_1(\mu, \text{esd}(\boldsymbol{X})) + W_1(\text{esd}(\boldsymbol{X}), \text{esd}(\boldsymbol{X} + \boldsymbol{\Delta})) \to 0$$

Hence $\text{esd}(\boldsymbol{X} + \boldsymbol{\Delta}) \xrightarrow{(w)} \mu$. The converse follows in the same way. $\qquad\square$

### B.4 Free probability

We now introduce some of the main notions of free probability theory. One of the main concerns of this field is the behavior of the eigenvalues of sums $\boldsymbol{X} + \boldsymbol{Y}$ of matrices whose eigenvectors are "generically positioned" in a suitable sense with respect to one another. More precisely, if $\boldsymbol{X}^{(n)}$ and $\boldsymbol{Y}^{(n)}$ are growing sequences of matrices with esd's converging to probability measures $\mu$ and $\nu$ respectively, then free probability explores situations where $\mathrm{esd}\left(\boldsymbol{X}^{(n)} + \boldsymbol{Y}^{(n)}\right)$ has a limit expressible in terms of $\mu$ and $\nu$.

One of the main definitions of free probability, albeit one which we will not need to use explicitly and thus do not introduce here, is that of *asymptotic freeness*, a condition under which the above can be achieved and the corresponding limit is the *additive free convolution* $\mu \boxplus \nu$. Asymptotic freeness is a particular notion of the generic position condition mentioned above, and for instance holds if $\boldsymbol{Y}^{(n)}$ is conjugated by a Haar-distributed orthogonal matrix independent of $\boldsymbol{X}^{(n)}$. Below we describe the generating function tools that can be used to *compute* the additive free convolution, but we mention the above motivation to explain its appearance in our results. Namely, in our setting the summands of $\boldsymbol{L} = \widehat{\boldsymbol{W}} + \boldsymbol{X}$ are asymptotically free, the former having eigenvalues with limiting distribution $\mu_{\mathrm{sc}}$ and the latter with distribution $\sigma(\mathcal{N}(0, 1))$. This explains the repeated appearance of the probability measure $\mu_{\mathrm{sc}} \boxplus \sigma(\mathcal{N}(0, 1))$ in our results. For further details on these notions, the interested reader may consult [VDN92, NS06, MS17].

In all definitions and statements below, we assume $\mu$ and $\nu$ are compactly supported probability measures on $\mathbb{R}$.

**Definition B.18** (Stieltjes transform). *The* Stieltjes transform *of $\mu$ is the function*

$$G_\mu(z) := \underset{X \sim \mu}{\mathbb{E}} \left[ \frac{1}{z - X} \right] \text{ for } z \in \mathbb{C} \setminus \mathrm{supp}(\mu).$$

**Definition B.19** ($R$-transform). *The $R$-transform of $\mu$ is the function*

$$R_\mu(z) := G_\mu^{-1}(z) - \frac{1}{z}$$

*where $G_\mu^{-1}$ is the functional inverse of $G_\mu$, on a domain where this is well-defined.*

**Definition B.20** (Additive free convolution). *Given two compactly supported probability measures $\mu$ and $\nu$, their additive free convolution $\mu \boxplus \nu$ is the unique probability measure such that $R_{\mu \boxplus \nu}(z) = R_\mu(z) + R_\nu(z)$.*

The following further special function plays an important role in the description of the additive free convolution of $\mu_{\mathrm{sc}}$ with some other compactly supported probability measure $\nu$ (see [Bia97]).

**Definition B.21** (Inverse subordination function). *For $\nu$ a compactly supported probability measure, define*

$$H_\nu(z) := z + G_\nu(z).$$

The importance of this function is that it is the inverse (on a suitable domain) of the *subordination function* associated to $\mu_{\mathrm{sc}}$ and $\rho$, which is the function $\omega_{\mu_{\mathrm{sc}}, \nu}$ satisfying

$$G_{\mu_{\mathrm{sc}} \boxplus \nu}(z) = G_{\mu_{\mathrm{sc}}}(\omega_{\mu_{\mathrm{sc}}, \nu}(z)) \text{ for all } z \in \mathbb{C}^+.$$

More details may be found in the older references [Voi93, Bia97, Bia98] or the more recent work of [CDMFF11, BG24] that we will use in our proofs.

## C  General random matrix analysis

### C.1  Largest eigenvalue of a rank-one perturbation of a diagonal matrix

We first present the main tool used to prove the second (and most complicated) part of Lemma 3.7 on the largest eigenvalue of the signal part $\boldsymbol{X}$ of a $\sigma$-Laplacian. Recall the form of this matrix: given a choice of $\sigma$, the matrix $\boldsymbol{X}$ from (1) takes the form

$$\boldsymbol{X} = \boldsymbol{X}(\boldsymbol{x}, \widehat{\boldsymbol{W}}; \beta) = \beta \boldsymbol{x} \boldsymbol{x}^\top + \mathrm{diag}(\sigma(\widehat{\boldsymbol{W}} \mathbf{1} + \beta \langle \boldsymbol{x}, \mathbf{1} \rangle \boldsymbol{x})). \tag{4}$$

This resembles a spiked matrix model, except where the noise component is a diagonal matrix instead of a "more random" matrix like a Wigner matrix.

We will understand such matrices through a more general result characterizing the largest eigenvalue of a sequence of matrices of the form

$$\boldsymbol{M} = \boldsymbol{M}^{(n)} = \boldsymbol{y}\boldsymbol{y}^\top + \mathrm{diag}(\boldsymbol{d}),$$

where $\boldsymbol{y} = \boldsymbol{y}^{(n)}, \boldsymbol{d} = \boldsymbol{d}^{(n)} \in \mathbb{R}^n$ are deterministic sequences of vectors satisfying

$$d_i^{(n)} \in [A, B] \text{ for all } i \in [n], \quad \text{and} \quad \lim_{n\to\infty} \max_{1 \le i \le n} d_i^{(n)} = B.$$

Our proof strategy follows that for the classical spiked matrix model using resolvents: we first express the outlier eigenvalues $\lambda$ of $\boldsymbol{M}$ as solutions to an equation of the form $G_n(\lambda) = 1$ (Lemma C.1). We then consider $\lim_{n\to\infty} G_n(z)$ and show that, with high probability, this limit exists pointwise and is given by some $G(z)$. Then the solutions of $G(z) = 1$ characterize the limiting behavior of the outlier eigenvalues of $\boldsymbol{M}$, if such outliers exist.

**Lemma C.1.** *$M$ has an eigenvalue $\lambda \notin [A, B]$ if and only if $\sum_{i=1}^n \frac{y_i^2}{\lambda - d_i} = 1$.*

*Proof.* $\lambda \notin [A, B]$ is an eigenvalue of $\boldsymbol{M}$ if and only if

$$0 = \det(\lambda \boldsymbol{I} - \mathrm{diag}(\boldsymbol{d}) - \boldsymbol{y}\boldsymbol{y}^\top) = \det(\lambda \boldsymbol{I} - \mathrm{diag}(\boldsymbol{d})) \det(\boldsymbol{I} - (\lambda \boldsymbol{I} - \mathrm{diag}(\boldsymbol{d}))^{-1}\boldsymbol{y}\boldsymbol{y}^\top).$$

Using the identity $\det(\boldsymbol{I} - \boldsymbol{A}\boldsymbol{B}) = \det(\boldsymbol{I} - \boldsymbol{B}\boldsymbol{A})$, we see that this in turn holds if and only if

$$0 = \det(\boldsymbol{I} - (\lambda \boldsymbol{I} - \mathrm{diag}(\boldsymbol{d}))^{-1}\boldsymbol{y}\boldsymbol{y}^\top) = 1 - \boldsymbol{y}^\top(\lambda \boldsymbol{I} - \mathrm{diag}(\boldsymbol{d}))^{-1}\boldsymbol{y} = 1 - \sum_{i=1}^n \frac{y_i^2}{\lambda - d_i},$$

completing the proof. $\qquad\square$

**Proposition C.2.** *Define functions $G_n : \mathbb{R} \setminus [A, B] \to \mathbb{R}$ by*

$$G_n(z) := \sum_{i=1}^n \frac{y_i^{(n)2}}{z - d_i^{(n)}}.$$

*Suppose there exists a function $G : \mathbb{R} \setminus [A, B] \to \mathbb{R}$ satisfying:*

1. *The function $G$ is continuous and strictly decreasing on $(B, \infty)$. Also, there exists $C > B$ for which $G(C) < 1$.*

2. *We have $G_n(z) \to G(z)$ for each $z \in (B, C]$.*

*If $G(z) = 1$ has a unique solution $\theta \in (B, \infty)$, then $\lambda_1(\boldsymbol{M}^{(n)}) \to \theta$ as $n \to \infty$. Otherwise, $\lambda_1(\boldsymbol{M}^{(n)}) \to B$ as $n \to \infty$.*

*Proof.* By Condition 1, the equation $G(z) = 1$ has at most one solution $z$ in $(B, C)$, and no solutions in $[C, \infty)$. We will analyze these two cases separately.

*Case 1: $G(z) = 1$ has a unique solution $\theta \in (B, C)$.* Fix a constant $0 < \epsilon < \max(\theta - B, C - \theta)$. Since the function $G$ is continuous and strictly decreasing, we have $G(\theta - \epsilon) > 1$ and $G(\theta + \epsilon) < 1$. By Condition 2, for sufficiently large $n$, we have $G_n(\theta - \epsilon) > 1$ and $G_n(\theta + \epsilon) < 1$. Since $G_n$ is continuous and strictly decreasing on $(B, \infty)$, we conclude that $G_n(z) = 1$ has a unique solution in $(B, \infty)$, and this solution lies in the interval $(\theta - \epsilon, \theta + \epsilon)$. Applying Lemma C.1, this result implies that $|\lambda_1(\boldsymbol{M}^{(n)}) - \theta| < \epsilon$. Hence, we have $\lambda_1(\boldsymbol{M}^{(n)}) \to \theta$ as $n \to \infty$.

*Case 2: $G(z) = 1$ has no solutions in $(B, C]$.* Fix a constant $0 < \epsilon < C - B$. In this case, we must have $G(B + \epsilon) < 1$. By Condition 2, for sufficiently large $n$, it follows that $G_n(B + \epsilon) < 1$. Since $G_n$ is continuous and strictly decreasing on $(B, \infty)$, we have $G_n(z) < 1$ for all $z \ge B + \epsilon$. Thus, by Lemma C.1, we deduce that $\lambda_1(\boldsymbol{M}^{(n)}) < B + \epsilon$. On the other hand, by Weyl's interlacing inequality (Corollary B.16), we know $\lambda_1(\boldsymbol{M}^{(n)}) \ge \lambda_1(\mathrm{diag}(\boldsymbol{d}^{(n)})) = \max_{i=1}^n d_i^{(n)} \to B$. Combining these results, we conclude that $\lambda_1(\boldsymbol{M}^{(n)}) \to B$. $\qquad\square$

## C.2 Analysis of signal eigenvalues: Proof of Lemma 3.7

Before we proceed, note that, in the Sparse Biased PCA model, $\boldsymbol{X}$ from (1) takes the form

$$\boldsymbol{X} = \beta \boldsymbol{x}\boldsymbol{x}^\top + \mathrm{diag}\bigg( \underbrace{\sigma\bigg(\widehat{\boldsymbol{W}}\boldsymbol{1} + \beta\frac{\|\boldsymbol{y}\|_1}{\|\boldsymbol{y}\|^2}\boldsymbol{y}\bigg)}_{=:\boldsymbol{d}} \bigg).$$

Since $\widehat{\boldsymbol{W}}\boldsymbol{1} \sim \mathcal{N}\left(0, \boldsymbol{I} + \frac{\boldsymbol{1}\boldsymbol{1}^\top}{n}\right)$, we may instead model $\boldsymbol{X}$ as

$$\boldsymbol{X} = \beta \boldsymbol{x}\boldsymbol{x}^\top + \mathrm{diag}\bigg( \underbrace{\sigma\bigg(\beta\frac{\|\boldsymbol{y}\|_1}{\|\boldsymbol{y}\|^2}\boldsymbol{y} + \boldsymbol{g} + \frac{t\boldsymbol{1}}{\sqrt{n}}\bigg)}_{=:\boldsymbol{d}} \bigg), \tag{5}$$

where $\boldsymbol{g} = \boldsymbol{g}^{(n)} \sim \mathcal{N}(0, \boldsymbol{I}_n)$ and $t = t^{(n)} \sim \mathcal{N}(0,1)$ are independent.

### C.2.1 Claim 1: Weak convergence

Recall that the first claim of the Lemma states that, almost surely,

$$\mathrm{esd}\left(\boldsymbol{X}^{(n)}\right) \xrightarrow{\text{(w)}} \nu = \nu_\sigma := \mathrm{Law}(\sigma(g)) \text{ where } g \sim \mathcal{N}(0,1).$$

*Proof of Lemma 3.7 (1).* By Corollary B.17, it is sufficient to prove that almost surely

$$\mathrm{ed}\,(\boldsymbol{d}) = \mathrm{ed}\bigg(\sigma\bigg(\beta\frac{\|\boldsymbol{y}\|_1}{\|\boldsymbol{y}\|^2}\boldsymbol{y} + \boldsymbol{g} + \frac{t\boldsymbol{1}}{\sqrt{n}}\bigg)\bigg) \xrightarrow{\text{(w)}} \nu.$$

Note that the matrix

$$\mathrm{diag}\bigg(\sigma\bigg(\beta\frac{\|\boldsymbol{y}\|_1}{\|\boldsymbol{y}\|^2}\boldsymbol{y} + \boldsymbol{g} + \frac{t\boldsymbol{1}}{\sqrt{n}}\bigg)\bigg) - \mathrm{diag}\bigg(\sigma\bigg(\boldsymbol{g} + \frac{t\boldsymbol{1}}{\sqrt{n}}\bigg)\bigg)$$

is of rank at most $\sum_{i=1}^n \varepsilon_i$. Under the random subset sparsity model, we have $\sum_{i=1}^n \varepsilon_i = np = o(n)$ by assumption. Under the independent entries sparsity model, consider the event $\left\{\frac{1}{np}\sum_{i=1}^n \varepsilon_i \to 1\right\}$, which happens almost surely by Lemma B.6. Under this event, $\sum_{i=1}^n \varepsilon_i = o(n)$, so again the rank is $o(n)$.

Therefore, by Corollary B.17, in both sampling models, it is sufficient to show that almost surely

$$\mathrm{ed}\bigg(\sigma\bigg(\boldsymbol{g} + \frac{t\boldsymbol{1}}{\sqrt{n}}\bigg)\bigg) \xrightarrow{\text{(w)}} \nu.$$

Conditional on $t$, the Lipschitz continuity of $\sigma$ (Assumption 1) implies

$$\bigg\|\sigma\bigg(\boldsymbol{g} + \frac{t\boldsymbol{1}}{\sqrt{n}}\bigg) - \sigma(\boldsymbol{g})\bigg\|_\infty \le \frac{\ell t}{\sqrt{n}} \to 0,$$

where $\ell$ is the Lipschitz constant of $\sigma$. On the event $\mathcal{E} := \{\mathrm{ed}\,(\sigma(\boldsymbol{g})) \xrightarrow{\text{(w)}} \nu\}$, which holds almost surely by Proposition B.2, $\mathrm{ed}\left(\sigma\left(\boldsymbol{g} + \frac{t\boldsymbol{1}}{\sqrt{n}}\right)\right) \xrightarrow{\text{(w)}} \nu$ by Proposition B.5. Therefore,

$$\begin{aligned}
\mathbb{P}\bigg(\mathrm{ed}\bigg(\sigma\bigg(\boldsymbol{g} + \frac{t\boldsymbol{1}}{\sqrt{n}}\bigg)\bigg) \xrightarrow{\text{(w)}} \nu\bigg) &= \mathbb{E}_t\bigg[\mathbb{P}_{\boldsymbol{g}^{(n)}}\bigg(\bigg\{\mathrm{ed}\bigg(\sigma\bigg(\boldsymbol{g} + \frac{t\boldsymbol{1}}{\sqrt{n}}\bigg)\bigg) \xrightarrow{\text{(w)}} \nu\bigg\} \cap \mathcal{E}\,\bigg|\,t\bigg)\bigg] \\
&= \mathbb{E}_t[1] \\
&= 1,
\end{aligned}$$

completing the proof. $\qquad\square$

### C.2.2 Claim 2: Convergence of largest eigenvalue

Recall that the second claim of the Lemma states that, almost surely, $\lambda_1(\boldsymbol{X}^{(n)}) \to \theta$ where $\theta$ solves the equation

$$\mathbb{E}_{y \sim \eta,\, g \sim \mathcal{N}(0,1)} \left[ \frac{y^2}{\theta - \sigma\left( \frac{\beta m_1}{m_2} y + g \right)} \right] = \frac{m_2}{\beta}$$

if such $\theta > \mathrm{edge}^+(\sigma)$ exists, and $\theta = \mathrm{edge}^+(\sigma)$ otherwise. Here we set

$$m_1 := \mathbb{E}_{x \sim \eta}\; x,$$

$$m_2 := \mathbb{E}_{x \sim \eta}\; x^2.$$

*Proof of Lemma 3.7 (2).* We prove this by specializing Proposition C.2 to the particular form of $\boldsymbol{X}$ given in (5).

Take $A = \mathrm{edge}^-(\sigma)$, $B = \mathrm{edge}^+(\sigma)$, then $d_i = \sigma\left( \beta \frac{\sum_j y_j}{\|\boldsymbol{y}\|^2} y_i + g_i + \frac{t}{\sqrt{n}} \right) \in [A, B]$ for all $i$, and $\max_{i=1}^n d_i \to B$ almost surely. The function $G_n \colon \mathbb{R} \setminus [A, B] \to \mathbb{R}$ is defined by

$$G_n(z) := \frac{\beta}{\|\boldsymbol{y}\|^2} \sum_{i=1}^n \frac{y_i^2}{z - \sigma\left( \beta \frac{\sum_j y_j}{\|\boldsymbol{y}\|^2} y_i + g_i + \frac{t}{\sqrt{n}} \right)}.$$

We further define $G \colon \mathbb{R} \setminus [A, B] \to \mathbb{R}$ by

$$G(z) := \frac{\beta}{m_2} \mathbb{E}_{\substack{y \sim \eta \\ g \sim \mathcal{N}(0,1)}} \left[ \frac{y^2}{z - \sigma\left( \frac{\beta m_1}{m_2} y + g \right)} \right]. \tag{6}$$

Take $C = B + \beta$, then $G(C) < 1$. It is easy to check that Conditions 1 of Proposition C.2 is satisfied.

To prove Condition 2, which concerns the closeness of $G_n$ and $G$, we first introduce an intermediate function $H_n$. In Lemma C.3, we establish pointwise closeness between $G_n$ and $H_n$, as well as between $H_n$ and $G$. Building on this, we then prove the desired closeness in Lemma C.4. The statements and proofs of both Lemmas will be given below.

Finally, since all the conditions in Proposition C.2 are satisfied almost surely by the defined functions $G_n$ and $G$, the proof of the claim is completed by using the Proposition. $\qquad \square$

**Lemma C.3.** *Define* $H_n : \mathbb{R} \setminus [A, B] \to \mathbb{R}$ *by*

$$H_n(z) := \frac{\beta}{m_2 np} \sum_{i=1}^n \frac{y_i^2}{z - \sigma\left( \frac{\beta m_1}{m_2} y_i + g_i \right)}.$$

*Then, the following hold.*

1. *Almost surely, for all $z \in (B, \infty)$, we have $|G_n(z) - H_n(z)| \to 0$.*

2. *For each $z \in (B, \infty)$, we have $H_n(z) \xrightarrow{\text{(a.s.)}} G(z)$.*

We note that the orders of quantifiers are different in the two results: the first says that almost surely a convergence happens for all $z \in (B, \infty)$, while the second says that it happens almost surely for any particular $z$.

*Proof.* Let $\ell$ be the Lipschitz constant of $\sigma$ (which is finite by our Assumption 1).

For the first claim, define event

$$\mathcal{E} := \left\{ \frac{\sum_{i=1}^n y_i}{np} \to m_1, \;\; \frac{\|\boldsymbol{y}\|^2}{np} \to m_2, \;\; \frac{\sum_{i=1}^n |y_i|^3}{np} \to \mathbb{E}_{x \sim \eta} |x|^3, \;\; \frac{t}{\sqrt{n}} \to 0 \right\}.$$

Under the random subset sparsity model, $\mathcal{E}$ happens almost surely by the Strong Law of Large Numbers; under the independent entries sparsity model, $\mathcal{E}$ also happens almost surely by Lemma B.7. For each $z \in (B, \infty)$, fix a constant $0 < \epsilon < z - B$, then on the event $\mathcal{E}$,

$$
|G_n(z) - H_n(z)| \leq \left| \|\boldsymbol{y}\|^{-2} - \frac{1}{m_2 np} \right| \left| \sum_{i=1}^{n} \frac{\beta y_i^2}{z - \sigma \left( \beta \frac{\sum_j y_j}{\|\boldsymbol{y}\|^2} y_i + g_i + \frac{t}{\sqrt{n}} \right)} \right|
$$

$$
+ \frac{\beta}{m_2 np} \sum_{i=1}^{n} y_i^2 \left| \frac{1}{z - \sigma \left( \beta \frac{\sum_j y_j}{\|\boldsymbol{y}\|^2} y_i + g_i + \frac{t}{\sqrt{n}} \right)} - \frac{1}{z - \sigma \left( \frac{\beta m_1}{m_2} y_i + g_i \right)} \right|
$$

$$
\leq \beta \epsilon^{-1} \left| 1 - \frac{\|\boldsymbol{y}\|^2}{m_2 np} \right| + m_2^{-1} \epsilon^{-2} \beta^2 \ell \frac{\sum_i |y_i|^3}{np} \left| \frac{\sum_i y_i}{\|y\|^2} - \frac{m_1}{m_2} \right|
$$

$$
+ m_2^{-1} \epsilon^{-2} \beta \ell \frac{\|\boldsymbol{y}\|^2}{np} \left| \frac{t}{\sqrt{n}} \right|
$$

$$
\to 0
$$

giving the result.

For the second claim, recall that $y_i = \varepsilon_i z_i$, where $\varepsilon_i \in \{0, 1\}$. Hence,

$$
H_n(z) = \frac{\beta}{m_2 np} \sum_{i=1}^{n} \frac{\varepsilon_i z_i^2}{z - \sigma \left( \frac{\beta m_1}{m_2} z_i + g_i \right)}.
$$

For each $z \in (B, \infty)$, we have $\mathbb{E} \left| \frac{z_i^2}{z - \sigma \left( \frac{\beta m_1}{m_2} z_i + g_i \right)} \right| \leq m_2 (z - B)^{-1} < \infty$, hence $H_n(z) \xrightarrow{\text{(a.s.)}} G(z)$ by the Strong Law of Large Numbers under the random subset sparsity model, and by Lemma B.7 under the independent entries sparsity model. $\qquad \square$

**Lemma C.4.** *Almost surely, for all $z \in (B, C]$, we have $G_n(z) \to G(z)$.*

*Proof.* Let $\mathcal{E}$ be the event that both $H_n(z) \to G(z)$ for all rational numbers $z \in \mathbb{Q} \cap (B, C]$ and $\frac{\|\boldsymbol{y}\|^2}{np} \to m_2$. By Lemma C.3(2), Lemma B.7, and the countability of $\mathbb{Q}$, the event $\mathcal{E}$ occurs almost surely. We first show that, on the event $\mathcal{E}$, we have $H_n(z) \to G(z)$ for all $z \in (B, C]$.

Consider an arbitrary $z \in (B, C]$, and let $\epsilon > 0$. Choose a constant $0 < \delta < z - B$. We note that $H_n$ and $G$ are both Lipschitz on $(B + \delta, C]$, and observe that

$$
|H_n'(z)| \leq \frac{\beta}{\delta^2 m_2} \frac{\|\boldsymbol{y}\|^2}{np}, \quad |G'(z)| \leq \frac{\beta}{\delta^2}.
$$

Choose a rational number $w \in \mathbb{Q} \cap (B, C]$ such that $|z - w| < \min \left( \frac{\delta^2 \epsilon}{2\beta}, z - B - \delta \right)$. Then we have

$$
|H_n(z) - G(z)| \leq |H_n(w) - G(w)| + |H_n(z) - H_n(w)| + |G(w) - G(z)|
$$

$$
\leq |H_n(w) - G(w)| + \left( \frac{\beta}{\delta^2 m_2} \frac{\|\boldsymbol{y}\|^2}{np} + \frac{\beta}{\delta^2} \right) \frac{\delta^2 \epsilon}{2\beta}.
$$

Taking the limit $n \to \infty$, we get $\lim_{n \to \infty} |H_n(z) - G(z)| \leq \epsilon$. Since $\epsilon$ is arbitrary, we conclude that $H_n(z) \to G(z)$. Therefore, almost surely, for all $z \in (B, C]$, we have $H_n(z) \to G(z)$.

Finally, combining this result with Lemma C.3(1), we complete the proof. $\qquad \square$

### C.2.3 Claim 3: Control of other eigenvalues

*Proof of Lemma 3.7 (3).* Since $\boldsymbol{X} = \beta \boldsymbol{x} \boldsymbol{x}^\top + \text{diag}(\boldsymbol{d})$, by Weyl's interlacing inequality (Corollary B.16),

$$
\text{edge}(\sigma)^+ \geq \lambda_1(\text{diag}(\boldsymbol{d})) \geq \lambda_2(\boldsymbol{X}) \geq \lambda_2(\text{diag}(\boldsymbol{d})) \geq \cdots
$$

$$
\cdots \geq \lambda_{n-1}(\boldsymbol{X}) \geq \lambda_{n-1}(\text{diag}(\boldsymbol{d})) \geq \text{edge}(\sigma)^-.
$$

Thus, $\lambda_2(\boldsymbol{X}), \ldots, \lambda_n(\boldsymbol{X})$ lie in $\sigma(\mathbb{R})$, as claimed. $\qquad \square$

## C.3 Compression to obtain independent spiked matrix model

Recall that the $\sigma$-Laplacian $\boldsymbol{L} = \boldsymbol{L}_\sigma$ from (1) takes the form

$$\boldsymbol{L} = \widehat{\boldsymbol{W}} + \boldsymbol{X},$$

$$\boldsymbol{X} = \boldsymbol{X}(\boldsymbol{x}, \widehat{\boldsymbol{W}}; \beta) = \beta \boldsymbol{x}\boldsymbol{x}^\top + \operatorname{diag}(\underbrace{\sigma(\widehat{\boldsymbol{W}}\boldsymbol{1} + \beta\langle \boldsymbol{x}, \boldsymbol{1}\rangle \boldsymbol{x})}_{=:\boldsymbol{d}}).$$

where $\widehat{\boldsymbol{W}}$ is a Wigner matrix with entrywise variance $1/n$. This resembles the spiked matrix model studied in [CDMFF11], with the key complication that $\widehat{\boldsymbol{W}}$ and $\boldsymbol{X}$ are weakly dependent, since $\boldsymbol{X}$ depends on $\widehat{\boldsymbol{W}}\boldsymbol{1}$. We do not expect this dependence to make a difference in the behavior of these random matrices in general: it should be possible to merely pretend that $\widehat{\boldsymbol{W}}\boldsymbol{1}$ is a Gaussian random vector with suitable distribution sampled independently of $\widehat{\boldsymbol{W}}$. However, to rigorously treat the actual $\sigma$-Laplacian matrices under consideration, we now introduce a trick to circumvent this complication and construct from this weakly dependent spiked matrix model an exactly independent one.

The idea is to "project away" the $\boldsymbol{1}$ direction from $\boldsymbol{L}$, which will make $\widehat{\boldsymbol{W}}$ independent of the (now slightly deformed) signal matrix. Specifically, we now present the conditions under which "compressing" $\boldsymbol{L}$ in this way via

$$\boldsymbol{L} \mapsto \widetilde{\boldsymbol{L}} := \boldsymbol{V}^\top \boldsymbol{L} \boldsymbol{V},$$

where $\boldsymbol{V} \in \mathbb{R}^{n \times (n-1)}$ has columns forming an orthonormal basis for the orthogonal complement of the all-ones vector $\boldsymbol{1}$, produces the desired independence of signal and noise components.

The main technical point is that $\widehat{\boldsymbol{W}}$ drawn from the GOE is an orthogonally invariant random matrix; that is, $\boldsymbol{Q}^\top \widehat{\boldsymbol{W}} \boldsymbol{Q}$ has the same law as $\widehat{\boldsymbol{W}}$ for any orthogonal $\boldsymbol{Q}$. So, the compressed noise term $\widetilde{\boldsymbol{W}} = \boldsymbol{V}^\top \widehat{\boldsymbol{W}} \boldsymbol{V}$ remains a GOE matrix, now of dimension $(n-1) \times (n-1)$. Moreover, in our formulation, $\boldsymbol{X}$ depends on $\widehat{\boldsymbol{W}}$ only through the vector $\widehat{\boldsymbol{W}}\boldsymbol{1}$. By projecting $\widehat{\boldsymbol{W}}$ onto the orthogonal complement of $\widehat{\boldsymbol{1}}$, we remove the component that correlates with $\boldsymbol{X}$, and thus the resulting noise term becomes independent of $\boldsymbol{X}$ after compression. This is formalized in the Proposition below.

**Proposition C.5.** *Suppose* $\boldsymbol{L} = \boldsymbol{L}^{(n)}$ *of the form in (1) satisfies the following:*

1. $\widehat{\boldsymbol{W}} = \widehat{\boldsymbol{W}}^{(n)} \sim \operatorname{GOE}(n, 1/n)$.

2. *Almost surely,* $\sum_{i=1}^n \mathbb{1}\{x_i = 0\} \to \infty$, $\|\boldsymbol{x}^{(n)}\|_1 = o(\sqrt{n})$, $\operatorname{esd}\left(\boldsymbol{X}^{(n)}\right) \xrightarrow{(\mathrm{w})} \nu$ *where* $\nu$ *has finite first moment, and* $\lambda_1(\boldsymbol{X}^{(n)}) \to \theta \in \mathbb{R}$.

*Then, for any sequence of matrices* $\boldsymbol{V} = \boldsymbol{V}^{(n)} \in \mathbb{R}^{n \times (n-1)}$ *satisfying* $\boldsymbol{V}\boldsymbol{V}^\top = \boldsymbol{I}_n - \widehat{\boldsymbol{1}}\widehat{\boldsymbol{1}}^\top$ *and* $\boldsymbol{V}^\top \boldsymbol{V} = \boldsymbol{I}_{n-1}$, *the compression* $\widetilde{\boldsymbol{L}} = \widetilde{\boldsymbol{L}}^{(n)} := \boldsymbol{V}^\top \boldsymbol{L} \boldsymbol{V}$ *can be decomposed as*

$$\widetilde{\boldsymbol{L}} = \widetilde{\boldsymbol{W}} + \widetilde{\boldsymbol{X}} + \boldsymbol{\Delta} \in \mathbb{R}_{\mathrm{sym}}^{(n-1) \times (n-1)},$$

*where:*

1. $\widetilde{\boldsymbol{W}} = \widetilde{\boldsymbol{W}}^{(n)} \sim \operatorname{GOE}(n-1, \frac{1}{n-1})$,

2. *Almost surely,* $\operatorname{esd}(\widetilde{\boldsymbol{X}}^{(n)}) \xrightarrow{(\mathrm{w})} \nu$, $\|\boldsymbol{\Delta}^{(n)}\| \to 0$, $\lambda_1(\widetilde{\boldsymbol{X}}^{(n)}) \to \theta$, *and all other eigenvalues* $\lambda_2(\widetilde{\boldsymbol{X}}^{(n)}), \ldots, \lambda_{n-1}(\widetilde{\boldsymbol{X}}^{(n)})$ *lie in* $[\operatorname{edge}(\sigma)^-, \operatorname{edge}(\sigma)^+]$.

3. $\widetilde{\boldsymbol{W}}^{(n)}$ *and* $\widetilde{\boldsymbol{X}}^{(n)}$ *are independent.*

**Remark C.6.** *The matrix* $\boldsymbol{L}$ *from the Sparse Biased PCA model satisfies the assumptions, since almost surely*

$$\sum_{i=1}^n \mathbb{1}\{x_i = 0\} \geq \sum_{i=1}^n \mathbb{1}\{\varepsilon_i = 0\} \to \infty,$$

$$\|\boldsymbol{x}^{(n)}\|_1 = \frac{\|\boldsymbol{y}^{(n)}\|_1}{\|\boldsymbol{y}^{(n)}\|^2} \|\boldsymbol{y}^{(n)}\| = O(\sqrt{np}) = o(\sqrt{n}),$$

*and* $\nu = \sigma(\mathcal{N}(0,1))$ *has finite first moment due to the boundedness of* $\sigma$.

*Proof.* Define

$$\widetilde{\boldsymbol{W}} = \widetilde{\boldsymbol{W}}^{(n)} = \sqrt{\frac{n}{n-1}} \boldsymbol{V}^{(n)^\top} \widehat{\boldsymbol{W}}^{(n)} \boldsymbol{V}^{(n)},$$

$$\widetilde{\boldsymbol{X}} = \widetilde{\boldsymbol{X}}^{(n)} = \boldsymbol{V}^{(n)^\top} \boldsymbol{X}^{(n)} \boldsymbol{V}^{(n)},$$

$$\boldsymbol{\Delta} = \boldsymbol{\Delta}^{(n)} = \left(1 - \sqrt{\frac{n}{n-1}}\right) \boldsymbol{V}^{(n)^\top} \widehat{\boldsymbol{W}}^{(n)} \boldsymbol{V}^{(n)}.$$

Consider the matrix $\boldsymbol{Q} := [\,\widehat{\boldsymbol{1}}\ \ \boldsymbol{V}\,]$, which is orthogonal. Since the GOE ensemble is invariant under orthogonal conjugation, we have $\boldsymbol{Q}^\top \widehat{\boldsymbol{W}} \boldsymbol{Q} \sim \mathrm{GOE}(n, n^{-1})$. Its right bottom $(n-1) \times (n-1)$ submatrix is $\boldsymbol{V}^\top \widehat{\boldsymbol{W}} \boldsymbol{V}$, and hence $\widetilde{\boldsymbol{W}} \sim \sqrt{\frac{n}{n-1}} \cdot \mathrm{GOE}(n-1, n^{-1}) = \mathrm{GOE}(n-1, (n-1)^{-1})$.

By Proposition B.14, $\|\widetilde{\boldsymbol{W}}\| \xrightarrow{\text{(a.s.)}} 2$, hence

$$\|\boldsymbol{\Delta}\| = \left(1 - \sqrt{\frac{n-1}{n}}\right) \|\widetilde{\boldsymbol{W}}\| \xrightarrow{\text{(a.s.)}} 0.$$

Moreover, the first column (and row) of $\boldsymbol{Q}^\top \widehat{\boldsymbol{W}} \boldsymbol{Q}$ is independent of $\widetilde{\boldsymbol{W}}$, and corresponds to $\widehat{\boldsymbol{W}}\widehat{\boldsymbol{1}}$ expressed in the basis formed by the columns of $\boldsymbol{Q}$. Since $\widetilde{\boldsymbol{X}}$ is a function of $\widehat{\boldsymbol{W}}\widehat{\boldsymbol{1}}$, we conclude that $\widetilde{\boldsymbol{W}}$ and $\widetilde{\boldsymbol{X}}$ are independent.

Let $\boldsymbol{Q}_0 := [\,\boldsymbol{0}\ \ \boldsymbol{V}\,]$, so that

$$\boldsymbol{Q}_0^\top \boldsymbol{X} \boldsymbol{Q}_0 = \begin{bmatrix} 0 & \boldsymbol{0}^\top \\ \boldsymbol{0} & \boldsymbol{V}^\top \boldsymbol{X} \boldsymbol{V} \end{bmatrix}.$$

Then, $\boldsymbol{Q}^\top \boldsymbol{X} \boldsymbol{Q} - \boldsymbol{Q}_0^\top \boldsymbol{X} \boldsymbol{Q}_0$ has rank at most 2. By the stability of the esd under low rank perturbation (Corollary B.17), this does not affect the weak convergence of the esd:

$$\mathbb{P}\left(\mathrm{esd}\left(\widetilde{\boldsymbol{X}}\right) \xrightarrow{\text{(w)}} \nu\right) = \mathbb{P}\left(\mathrm{esd}\left(\boldsymbol{Q}_0^\top \boldsymbol{X} \boldsymbol{Q}_0\right) \xrightarrow{\text{(w)}} \nu\right) \geq \mathbb{P}\left(\mathrm{esd}\left(\boldsymbol{Q}^\top \boldsymbol{X} \boldsymbol{Q}\right) \xrightarrow{\text{(w)}} \nu\right) = 1$$

To control the eigenvalues aside from the top one, note that, for any unit vector $\boldsymbol{y} \in \mathbb{R}^{n-1}$, $\boldsymbol{y}^\top \boldsymbol{V}^\top \mathrm{diag}(\boldsymbol{d}) \boldsymbol{V} \boldsymbol{y} = \sum_i \boldsymbol{d}_i (\boldsymbol{V}\boldsymbol{y})_i^2 \in [\mathrm{edge}(\sigma)^-, \mathrm{edge}(\sigma^+)]$. So, all eigenvalues of $\boldsymbol{V}^\top \mathrm{diag}(\boldsymbol{d}) \boldsymbol{V}$ are in $[\mathrm{edge}(\sigma)^-, \mathrm{edge}(\sigma)^+]$. Since $\widetilde{\boldsymbol{X}} = \beta (\boldsymbol{V}^\top \boldsymbol{x})(\boldsymbol{V}^\top \boldsymbol{x})^\top + \boldsymbol{V}^\top \mathrm{diag}(\boldsymbol{d}) \boldsymbol{V}$, by Weyl's interlacing inequality (Corollary B.16),

$$\mathrm{edge}(\sigma)^+ \geq \lambda_1(\boldsymbol{V}^\top \mathrm{diag}(\boldsymbol{d}) \boldsymbol{V}) \geq \lambda_2(\widetilde{\boldsymbol{X}}) \geq \lambda_2(\boldsymbol{V}^\top \mathrm{diag}(\boldsymbol{d}) \boldsymbol{V}) \geq \cdots$$

$$\cdots \geq \lambda_{n-1}(\widetilde{\boldsymbol{X}}) \geq \lambda_{n-1}(\boldsymbol{V}^\top \mathrm{diag}(\boldsymbol{d}) \boldsymbol{V}) \geq \mathrm{edge}(\sigma)^-,$$

as claimed.

It is left to prove $\lambda_1(\widetilde{\boldsymbol{X}}) \to \theta$ almost surely. First, by Weyl's interlacing inequality (Corollary B.16), and on the event $\{\sum_{i=1}^n \mathbb{1}\{x_i = 0\} \to \infty\}$ which happens almost surely by assumption, we have

$$\lambda_1(\boldsymbol{X}) \geq \lambda_1(\mathrm{diag}(\boldsymbol{d})) = \max_i d_i \geq \max_{i:x_i=0} \sigma((\widehat{\boldsymbol{W}}\boldsymbol{1})_i) \to \mathrm{edge}(\sigma)^+.$$

Hence, $\theta \geq \mathrm{edge}(\sigma^+)$. We now consider two cases, depending on whether equality holds here.

*Case 1:* $\theta > \mathrm{edge}(\sigma)^+$. On the one hand, since $\|\boldsymbol{V}\boldsymbol{x}\|^2 = \boldsymbol{x}^\top \boldsymbol{V}^\top \boldsymbol{V} \boldsymbol{x} = \|\boldsymbol{x}\|^2$ for all $\boldsymbol{x} \in \mathbb{R}^{n-1}$,

$$\lambda_1(\widetilde{\boldsymbol{X}}) = \max_{\|\boldsymbol{x}\|=1} \boldsymbol{x}^\top \boldsymbol{V}^\top \boldsymbol{X} \boldsymbol{V} \boldsymbol{x} \leq \max_{\|\boldsymbol{y}\|=1} \boldsymbol{y}^\top \boldsymbol{X} \boldsymbol{y} = \lambda_1(\boldsymbol{X})$$

On the other hand, let $\boldsymbol{v} = \boldsymbol{v}_1(\boldsymbol{X})$. We have

$$\lambda_1(\widetilde{\boldsymbol{X}}) \geq \frac{(\boldsymbol{V}^\top \boldsymbol{v})^\top (\boldsymbol{V}^\top \boldsymbol{X} \boldsymbol{V})(\boldsymbol{V}^\top \boldsymbol{v})}{\|\boldsymbol{V}^\top \boldsymbol{v}\|^2}$$

$$= \frac{\boldsymbol{v}^\top \boldsymbol{P} \boldsymbol{X} \boldsymbol{P} \boldsymbol{v}}{\|\boldsymbol{P}\boldsymbol{v}\|^2}$$

where $\boldsymbol{P} := \boldsymbol{I}_n - \widehat{\boldsymbol{1}}\widehat{\boldsymbol{1}}^\top$ is the projection matrix to the orthogonal complement of the $\widehat{\boldsymbol{1}}$ direction. Continuing,

$$
\begin{aligned}
&\geq \boldsymbol{v}^\top \boldsymbol{P}\boldsymbol{X}\boldsymbol{P}\boldsymbol{v} \\
&= \boldsymbol{v}^\top \boldsymbol{X}\boldsymbol{v} - \langle \boldsymbol{v}, \widehat{\boldsymbol{1}}\rangle(\widehat{\boldsymbol{1}}^\top \boldsymbol{X}\boldsymbol{v} + \boldsymbol{v}^\top \boldsymbol{X}\widehat{\boldsymbol{1}}) + \langle \boldsymbol{v}, \widehat{\boldsymbol{1}}\rangle^2 \\
&\geq \lambda_1(\boldsymbol{X}) - 2\|\boldsymbol{X}\| \cdot |\langle \boldsymbol{v}, \widehat{\boldsymbol{1}}\rangle|.
\end{aligned}
$$

It suffices to show that the last term is $o(1)$ as $n \to \infty$.

Suppose $|\sigma(x)| < K$ for all $x \in \mathbb{R}$. Then, $\|\boldsymbol{X}\| \leq \beta + K$, so $\|\boldsymbol{X}\| = \|\boldsymbol{X}^{(n)}\| = O(1)$. So, it further suffices to show that $|\langle \boldsymbol{v}, \widehat{\boldsymbol{1}}\rangle| = o(1)$.

Moreover, note that $\boldsymbol{v}$ satisfies

$$
\lambda_1(\boldsymbol{X})\boldsymbol{v} = \boldsymbol{X}\boldsymbol{v} = (\beta\boldsymbol{x}\boldsymbol{x}^\top + \mathrm{diag}(\boldsymbol{d}_i))\boldsymbol{v}
$$

Rearranging,

$$
\boldsymbol{v} = \beta\langle \boldsymbol{x}, \boldsymbol{v}\rangle(\lambda_1(\boldsymbol{X})\boldsymbol{I} - \mathrm{diag}(\boldsymbol{d}))^{-1}\boldsymbol{x}
$$

Hence,

$$
|\langle \boldsymbol{v}, \widehat{\boldsymbol{1}}\rangle| \leq \beta \frac{1}{\sqrt{n}} \sum_{i=1}^n \frac{|x_i|}{|\lambda_1(\boldsymbol{X}) - d_i|}
$$

Let $\epsilon = (\theta - \mathrm{edge}(\sigma)^+)/2 > 0$. Then, on the event $\{\|\boldsymbol{x}\|_1 = o(\sqrt{n}), \lambda_1(\boldsymbol{X}^{(n)}) \to \theta\}$ which happens almost surely, $\lambda_1(\boldsymbol{X}^{(n)}) > \theta - \epsilon$ for all sufficiently large $n$, so

$$
|\langle \boldsymbol{v}, \widehat{\boldsymbol{1}}\rangle| \leq \beta\epsilon^{-1}n^{-1/2}\|\boldsymbol{x}\|_1 = o(1)
$$

and therefore

$$
\lambda_1(\widetilde{\boldsymbol{X}}) - \lambda_1(\boldsymbol{X}) \to 0
$$

We conclude that $\lambda_1(\widetilde{\boldsymbol{X}}) \to \theta$ almost surely.

*Case 2:* $\theta = \mathrm{edge}(\sigma)^+$. Just like in Case 1, $\lambda_1(\widetilde{\boldsymbol{X}}) \leq \lambda_1(\boldsymbol{X})$. On the other hand, as $\widetilde{\boldsymbol{X}} = \boldsymbol{V}^\top \mathrm{diag}(\boldsymbol{d})\boldsymbol{V} + \beta(\boldsymbol{V}^\top \boldsymbol{x})(\boldsymbol{V}^\top \boldsymbol{x})^\top$, by Weyl's interlacing inequality (Corollary B.16)

$$
\lambda_1(\widetilde{\boldsymbol{X}}) \geq \lambda_1(\boldsymbol{V}^\top \mathrm{diag}(\boldsymbol{d})\boldsymbol{V}).
$$

Let $i := \arg\max_j d_j$, and set $\boldsymbol{y} := \boldsymbol{V}^\top \boldsymbol{e}_i = \boldsymbol{V}^\top \boldsymbol{e}_i$. Then $\|\boldsymbol{y}\|^2 = \|\boldsymbol{P}\boldsymbol{e}_i\|^2 = 1 - \frac{1}{n}$, and we have

$$
\begin{aligned}
&\geq \frac{\boldsymbol{y}^\top \boldsymbol{V}^\top \mathrm{diag}(\boldsymbol{d})\boldsymbol{V}\boldsymbol{y}}{\|\boldsymbol{y}\|^2} \\
&= \frac{\boldsymbol{e}_i^\top \boldsymbol{P}\mathrm{diag}(\boldsymbol{d})\boldsymbol{P}\boldsymbol{e}_i}{1 - \frac{1}{n}} \\
&= \frac{d_i - \frac{2}{n}d_i + \frac{1}{n^2}}{1 - \frac{1}{n}} \\
&\geq \frac{n-2}{n-1}d_i \\
&\to \mathrm{edge}(\sigma)^+
\end{aligned}
$$

We conclude that $\lambda_1(\widetilde{\boldsymbol{X}}) \to \theta = \mathrm{edge}(\sigma)^+$ almost surely, completing the proof. $\qquad\square$

## C.4 Analysis of compressed $\sigma$-Laplacian eigenvalues

### C.4.1 Weak convergence: Proof of Lemma 3.8

Recall from Proposition C.5 that we may decompose

$$
\widetilde{\boldsymbol{L}}^{(n)} = \boldsymbol{V}^{(n)\top}\boldsymbol{L}^{(n)}\boldsymbol{V}^{(n)} = \widetilde{\boldsymbol{W}}^{(n)} + \widetilde{\boldsymbol{X}}^{(n)} + \boldsymbol{\Delta}^{(n)},
$$

where $\widetilde{\boldsymbol{W}}^{(n)} \sim \mathrm{GOE}(n-1, \frac{1}{n-1})$, $\widetilde{\boldsymbol{W}}^{(n)}$ and $\boldsymbol{\Delta}^{(n)}$ are independent of $\widetilde{\boldsymbol{X}}^{(n)}$ (in fact $\boldsymbol{\Delta}^{(n)}$ is a multiple of $\widetilde{\boldsymbol{W}}^{(n)}$) and $\|\boldsymbol{\Delta}^{(n)}\| \to 0$ almost surely as $n \to \infty$. For the weak convergence result, we then first note that, by our Lemma 3.7, the esd of $\widetilde{\boldsymbol{X}}^{(n)}$ almost surely converges weakly to $\sigma(\mathcal{N}(0,1))$, while that of $\widetilde{\boldsymbol{W}}^{(n)}$ almost surely converges weakly to $\mu_{\mathrm{sc}}$ by Theorem B.12. Thus, by Proposition 4.3.9 of [HP00], the esd of $\widetilde{\boldsymbol{W}}^{(n)} + \widetilde{\boldsymbol{X}}^{(n)}$ almost surely converges weakly to $\mu_{\mathrm{sc}} \boxplus \sigma(\mathcal{N}(0,1))$, since the law of $\widetilde{\boldsymbol{W}}^{(n)}$ is orthogonally invariant (the Proposition in the reference is stated for unitary invariance, but the same result holds by the same proof under orthogonal invariance, as also mentioned in the discussion following this result in the reference).

### C.4.2  Largest eigenvalue: Proof of Theorem 3.3, eigenvalue part

Before proceeding to the proof, let us note the relationship between the thresholds we state and the ones stated in related results [CDMFF11, BG24] in terms of the inverse subordination function $H_\nu$ (Definition B.21). These are actually equivalent. Recall that, for a given $\nu$, this is defined as

$$H_\nu(z) = z + G_\nu(z)$$
$$= z + \mathop{\mathbb{E}}_{X \sim \nu}\left[\frac{1}{z - X}\right]$$

In our case, the $\nu$ that will arise is $\nu = \sigma(\mathcal{N}(0,1))$, which gives

$$= z + \mathop{\mathbb{E}}_{g \sim \mathcal{N}(0,1)}\left[\frac{1}{z - \sigma(g)}\right],$$

defined for all $z \in \mathbb{C} \setminus \overline{\sigma(\mathbb{R})}$.

In particular, we also have

$$H_\nu'(z) = 1 - \mathop{\mathbb{E}}_{g \sim \mathcal{N}(0,1)}\left[\frac{1}{(z - \sigma(g))^2}\right].$$

Note that this function converges to 1 as $z \to \infty$ along the real axis, and is continuous and strictly increasing on $(\mathrm{edge}^+(\sigma), +\infty)$. Moreover, by [CDMFF11, Lemma 2.1], $\mathrm{edge}^+(\sigma) \in \mathrm{supp}(\nu) \subseteq \overline{\{u \in \mathbb{R} \setminus \mathrm{supp}(\nu) : H_\nu'(u) < 0\}}$. Particularly, this implies that there exists some $u > \mathrm{edge}^+(\sigma)$ where $H_\nu'(u) < 0$. Recall also that $\theta_{\sigma,\beta} \geq \mathrm{edge}^+(\sigma)$ by definition.

Theorem 8.1 of [CDMFF11] implies that there is an outlier eigenvalue if and only if

$$\theta_{\sigma,\beta} \in \mathbb{R} \setminus \overline{\{u \in \mathbb{R} \setminus \mathrm{supp}(\nu) : H_\nu'(u) < 0\}}.$$

This is equivalent to our condition in Theorem 3.3,

$$H_\nu'(\theta_\sigma(\beta)) > 0,$$

since, by our assumption that $\sigma$ is Lipschitz-continuous and bounded, we know that $\mathrm{supp}(\nu)$ consists of a single closed interval of real numbers.[6]

Similarly, when $H_\nu'(\theta_\sigma(\beta)) > 0$, we state in Theorem 3.3 that

$$\lambda_1(\widetilde{\boldsymbol{L}}^{(n)}) \xrightarrow{\text{(a.s.)}} \theta_\sigma(\beta) + \mathop{\mathbb{E}}_{g \sim \mathcal{N}(0,1)}\left[\frac{1}{\theta_\sigma(\beta) - \sigma(g)}\right]$$

which may be rewritten

$$= H_\nu(\theta_\sigma(\beta)),$$

giving the form stated in the other references.

In the special case where the entrywise condition $x_i \geq 0$ holds almost surely (i.e., $\eta$ in Definition 1.1 is a probability measure on $\mathbb{R}_{\geq 0}$), $\theta_\sigma$ is an increasing function of $\beta$. Thus, $H_\nu'(\theta_\sigma(\beta)) > 0$ if and only if $\beta > \beta_*$, where $\beta_* = \beta_*(\sigma)$ solves

$$H_\nu'(\theta_\sigma(\beta_*)) = 0.$$

---

[6] Note that the matter of outlier eigenvalues is more complicated when considering deformations of matrices whose limiting esd's support has several connected components, since outliers can appear between these components (thus rather being "inliers" between parts of the undeformed limiting spectrum).

*Proof of Theorem 3.3, eigenvalue part.* Let us define a function $f_\sigma : \mathbb{R}_{\geq 0} \to \mathbb{R}$ by

$$f_\sigma(\beta) = \begin{cases} H_{\nu_{\sigma,\beta}}(\theta_\sigma(\beta)) & \text{if } H'_{\nu_{\sigma,\beta}}(\beta) > 0 \\ \text{edge}^+(\mu_{\text{sc}} \boxplus \sigma(\mathcal{N}(0,1))) & \text{otherwise.} \end{cases}$$

We then want to show that $\lambda_1(\widetilde{\boldsymbol{L}}^{(n)}) \to f_\sigma(\beta)$ almost surely.

Let $\mathcal{X}$ be the set of sequences $(\widetilde{\boldsymbol{X}}^{(n)})_{n \geq 1} \in \mathbb{R}^{1 \times 1}_{\text{sym}} \times \mathbb{R}^{2 \times 2}_{\text{sym}} \times \cdots$ that satisfy the conclusions of Lemma 3.7. The Lemma states that

$$\mathbb{P}\left((\widetilde{\boldsymbol{X}}^{(n)})_{n \geq 1} \in \mathcal{X}\right) = 1.$$

Conditioning the probability we are interested in on the value of $\widetilde{\boldsymbol{X}}^{(n)}$ for all $n$ and using the above independence property,

$$\mathbb{P}\left(\lambda_1(\widetilde{\boldsymbol{L}}^{(n)}) \to f_\sigma(\beta)\right)$$

$$= \mathbb{P}\left(\lambda_1(\widetilde{\boldsymbol{W}}^{(n)} + \widetilde{\boldsymbol{X}}^{(n)} + \boldsymbol{\Delta}^{(n)}) \to f_\sigma(\beta)\right)$$

$$= \mathop{\mathbb{E}}_{(\widetilde{\boldsymbol{X}}^{(n)})_{n \geq 1}} \left[ \mathop{\mathbb{P}}_{(\widetilde{\boldsymbol{W}}^{(n)})_{n \geq 1}} \left(\lambda_1(\widetilde{\boldsymbol{W}}^{(n)} + \widetilde{\boldsymbol{X}}^{(n)} + \boldsymbol{\Delta}^{(n)}) \to f_\sigma(\beta)\right) \right]$$

and using that $\boldsymbol{\Delta}^{(n)}$ is a function of $\widetilde{\boldsymbol{W}}^{(n)}$ and $\|\boldsymbol{\Delta}^{(n)}\| \to 0$ almost surely, we may rewrite

$$= \mathop{\mathbb{E}}_{(\widetilde{\boldsymbol{X}}^{(n)})_{n \geq 1}} \left[ \mathop{\mathbb{P}}_{(\widetilde{\boldsymbol{W}}^{(n)})_{n \geq 1}} \left(\lambda_1(\widetilde{\boldsymbol{W}}^{(n)} + \widetilde{\boldsymbol{X}}^{(n)} + \boldsymbol{\Delta}^{(n)}) \to f_\sigma(\beta), \|\boldsymbol{\Delta}^{(n)}\| \to 0\right) \right]$$

and now, by Weyl's inequality (Proposition B.15), the convergence of the largest eigenvalue is not affected by $\boldsymbol{\Delta}^{(n)}$, so

$$= \mathop{\mathbb{E}}_{(\widetilde{\boldsymbol{X}}^{(n)})_{n \geq 1}} \left[ \mathop{\mathbb{P}}_{(\widetilde{\boldsymbol{W}}^{(n)})_{n \geq 1}} \left(\lambda_1(\widetilde{\boldsymbol{W}}^{(n)} + \widetilde{\boldsymbol{X}}^{(n)}) \to f_\sigma(\beta), \|\boldsymbol{\Delta}^{(n)}\| \to 0\right) \right]$$

$$= \mathop{\mathbb{E}}_{(\widetilde{\boldsymbol{X}}^{(n)})_{n \geq 1}} \left[ \mathop{\mathbb{P}}_{(\widetilde{\boldsymbol{W}}^{(n)})_{n \geq 1}} \left(\lambda_1(\widetilde{\boldsymbol{W}}^{(n)} + \widetilde{\boldsymbol{X}}^{(n)}) \to f_\sigma(\beta)\right) \right]$$

and similarly, since $(\widetilde{\boldsymbol{X}}^{(n)})_{n \geq 1} \in \mathcal{X}$ almost surely, we may introduce the indicator of this event in the expectation

$$= \mathop{\mathbb{E}}_{(\widetilde{\boldsymbol{X}}^{(n)})_{n \geq 1}} \left[ \mathop{\mathbb{P}}_{(\widetilde{\boldsymbol{W}}^{(n)})_{n \geq 1}} \left(\lambda_1(\widetilde{\boldsymbol{W}}^{(n)} + \widetilde{\boldsymbol{X}}^{(n)}) \to f_\sigma(\beta)\right) \mathbb{1}\{(\widetilde{\boldsymbol{X}}^{(n)})_{n \geq 1} \in \mathcal{X}\} \right]$$

Finally, the condition $(\widetilde{\boldsymbol{X}}^{(n)})_{n \geq 1} \in \mathcal{X}$ precisely verifies the conditions of Theorem 1.1 of [BG24], whose conclusion is that the inner probability equals 1, whereby

$$= \mathop{\mathbb{E}}_{(\widetilde{\boldsymbol{X}}^{(n)})_{n \geq 1}} \left[ \mathbb{1}\{(\widetilde{\boldsymbol{X}}^{(n)})_{n \geq 1} \in \mathcal{X}\} \right]$$

$$= \mathbb{P}\left((\widetilde{\boldsymbol{X}}^{(n)})_{n \geq 1} \in \mathcal{X}\right)$$

$$= 1,$$

completing the proof. $\qquad\square$

As an aside, Theorem 1.1 presented in [BG24] is attributed there to [CDMFF11] by the authors, though the latter work only considers complex-valued Wigner matrices where the real and complex parts of the entries are i.i.d., while [BG24] and our application concern real-valued Wigner matrices. In any case, the real-valued version follows from the much more detailed large deviations principle proved by [BG24] (their Theorem 1.2).

## C.5  Eigenvector analysis: Proof of Theorem 3.3, eigenvector part

*Proof of Theorem 3.3, eigenvector part.*  To analyze the eigenvector, we introduce an ancillary matrix function

$$\boldsymbol{L}(t) = \boldsymbol{L}^{(n)}(t) := \widehat{\boldsymbol{W}} + \underbrace{(\beta + t)\boldsymbol{x}\boldsymbol{x}^\top + \operatorname{diag}(\sigma(\widehat{\boldsymbol{W}}\mathbf{1} + \beta\langle\boldsymbol{x},\mathbf{1}\rangle\boldsymbol{x}))}_{=:\boldsymbol{X}(t)}. \tag{7}$$

This choice ensures that evaluating at $t = 0$ recovers the $\sigma$-Laplacian matrix $\boldsymbol{L}$ defined in (1). We likewise define the compressed matrix

$$\widetilde{\boldsymbol{L}}(t) = \widetilde{\boldsymbol{L}}^{(n)}(t) := \boldsymbol{V}^\top \boldsymbol{L}(t)\,\boldsymbol{V}.$$

We use Lemma C.7, which shows that almost surely, for any $\beta, t$ such that $\beta > 0$, and $\beta + t > 0$, we have $\lambda_1^{(n)}(t) = \lambda_1(\widetilde{\boldsymbol{L}}^{(n)}(t)) \to f_\sigma(t)$, where $f_\sigma$ is a differentiable function. The statement and proof will follow below. Moreover, almost surely $\widetilde{\boldsymbol{L}}^{(n)}(0)$ has no repeated eigenvalues for all $n$ and $\beta > 0$, since matrices with repeated eigenvalues form an algebraic set of codimension 2 in the space of symmetric matrices (see, e.g., [BKL18]). On this event, the maps $t \mapsto \lambda_1^{(n)}(t)$ and $t \mapsto v_1^{(n)}(t)$ are differentiable at $t = 0$, by [Mag85, Theorem 1]. Define $\mathcal{E}$ be the almost sure event where these two conditions hold.

We write $\dot{\widetilde{\boldsymbol{L}}}$, $\dot{\boldsymbol{v}}_1$, and $\dot{\lambda}_1$ for derivatives with respect to $t$ whenever they are well-defined. Differentiating the eigenpair relation $\widetilde{\boldsymbol{L}}\,\boldsymbol{v}_1 = \lambda_1\,\boldsymbol{v}_1$ with respect to $t$ and left-multiplying by $\boldsymbol{v}_1^\top$ yields

$$\boldsymbol{v}_1^\top \dot{\widetilde{\boldsymbol{L}}}\,\boldsymbol{v}_1 + \boldsymbol{v}_1^\top \widetilde{\boldsymbol{L}}\,\dot{\boldsymbol{v}}_1 = \dot{\lambda}_1\,\boldsymbol{v}_1^\top \boldsymbol{v}_1 + \lambda_1\,\boldsymbol{v}_1^\top \dot{\boldsymbol{v}}_1,$$

which simplifies to

$$\dot{\lambda}_1 = \boldsymbol{v}_1^\top \dot{\widetilde{\boldsymbol{L}}}\,\boldsymbol{v}_1.$$

In our setup, $\dot{\widetilde{\boldsymbol{L}}} = \boldsymbol{V}^\top \boldsymbol{x}\boldsymbol{x}^\top \boldsymbol{V}$, so

$$\dot{\lambda}_1 = \langle\boldsymbol{x}, \boldsymbol{V}\boldsymbol{v}_1\rangle^2.$$

We observe that the map $t \mapsto \lambda_1^{(n)}(t)$ is convex for any $n$ and $\beta$. Indeed,

$$\lambda_1^{(n)}(t) = \max_{\|\boldsymbol{x}\|=1} \boldsymbol{x}^\top \widetilde{\boldsymbol{L}}^{(n)}(t)\,\boldsymbol{x},$$

and, for each fixed unit vector $\boldsymbol{x}$, the objective $\boldsymbol{x}^\top \widetilde{\boldsymbol{L}}^{(n)}(t)\,\boldsymbol{x}$ is affine in $t$, so taking the maximum over $\boldsymbol{x}$ yields a convex function of $t$. Hence, on the event $\mathcal{E}$, we can apply Lemma C.8 to obtain

$$\langle\boldsymbol{x}, \boldsymbol{V}\boldsymbol{v}_1^{(n)}(0)\rangle^2 = \dot{\lambda}_1^{(n)}(0) \longrightarrow f_\sigma'(0).$$

The result follows. □

**Lemma C.7.** *For each $\sigma$, $\beta$ and $t$, define $\theta = \theta_\sigma(\beta, t)$ to solve the equation*

$$\mathop{\mathbb{E}}_{\substack{y\sim\eta \\ g\sim\mathcal{N}(0,1)}} \left[\frac{y^2}{\theta - \sigma(\frac{m_1}{m_2}\beta y + g)}\right] = \frac{m_2}{\beta + t}$$

*if such $\theta > \operatorname{edge}^+(\sigma)$ exists, and let $\theta_\sigma = \operatorname{edge}^+(\sigma)$ otherwise. Furthermore, recall the inverse subordination function*

$$H_\nu(z) = z + \mathop{\mathbb{E}}_{g\sim\mathcal{N}(0,1)}\left[\frac{1}{z - \sigma(g)}\right].$$

*Almost surely, for every $\beta$ and $t$ so that $\beta > 0, \beta + t > 0$, the sequence $\widetilde{\boldsymbol{L}}(t) = \widetilde{\boldsymbol{L}}^{(n)}(t)$ satisfies $\lambda_1^{(n)}(t) \to f_\sigma(t)$, where*

$$f_\sigma(t) := \begin{cases} H_\nu(\theta_\sigma(\beta, t)) & \text{if } H_\nu'(\theta_\sigma(\beta, t)) > 0 \\ \operatorname{edge}^+(\mu_{\mathrm{sc}} \boxplus \sigma(\mathcal{N}(0,1))) & \text{otherwise.} \end{cases}$$

*Moreover, $f_\sigma$ is differentiable with*

$$f_\sigma'(0) = \begin{cases} \dfrac{m_2}{\beta^2}\left(\mathbb{E}_{y\sim\eta,\, g\sim\mathcal{N}(0,1)}\left[\dfrac{y^2}{\left(\theta_\sigma(\beta,0)-\sigma\left(\frac{m_1}{m_2}\beta y+g\right)\right)^2}\right]\right)^{-1} H_\nu'(\theta_\sigma(\beta, 0)) & \text{if } H_\nu'(\theta_\sigma(\beta, 0)) > 0 \\ 0 & \text{otherwise.} \end{cases}$$

*Proof.* The proof for the first part, which characterizes the limit of the largest eigenvalue, follows exactly the same argument as that for $\widetilde{L}$. Below, we show that $f_\sigma$ is differentiable and compute its limit.

Notice that $H'_\nu$ is strictly increasing, and $\theta_\sigma(\beta, t)$ also strictly increases with $t$. Consequently, there exists a unique $t_* = t_*(\beta)$ that solves $H'_\nu\big(\theta_\sigma(\beta, t_*(\beta))\big) = 0$, and $H'_\nu(\theta_\sigma(\beta, t)) > 0$ if and only if $t > t_*$.

We first consider the case where $H'_\nu(\theta_\sigma(\beta, t)) > 0$. By [CDMFF11, Lemma 2.1], we have $\mathrm{edge}^+(\sigma) \in \mathrm{supp}(\nu) \subseteq \overline{\{u \in \mathbb{R} \setminus \mathrm{supp}(\nu) : H'_\nu(u) < 0\}}$. Hence, $\theta_\sigma(\beta, t) \neq \mathrm{edge}^+(\sigma)$. This implies that $\theta = \theta_\sigma(\beta, t)$ solves $G(\theta, t) = 1$, where

$$G(\theta, t) := \frac{\beta + t}{m_2} \mathop{\mathbb{E}}_{\substack{y \sim \eta \\ g \sim \mathcal{N}(0,1)}} \left[ \frac{y^2}{\theta - \sigma\left(\frac{m_1}{m_2}\beta y + g\right)} \right].$$

By the implicit function theorem, $\theta_\sigma(\beta, t)$ is differentiable with respect to $t$ for all $t > t_*$. Using the chain rule, we can compute

$$\frac{d\theta_\sigma(\beta, t)}{dt} = \frac{m_2}{(\beta + t)^2} \left( \mathop{\mathbb{E}}_{\substack{y \sim \eta \\ g \sim \mathcal{N}(0,1)}} \left[ \frac{y^2}{\left(\theta_\sigma(\beta, t) - \sigma\left(\frac{m_1}{m_2}\beta y + g\right)\right)^2} \right] \right)^{-1}.$$

Consequently, $H_\nu\big(\theta_\sigma(\beta, t)\big)$ is differentiable, with

$$\frac{dH_\nu(\theta_\sigma(\beta, t))}{dt} = \frac{m_2}{(\beta + t)^2} \left( \mathop{\mathbb{E}}_{\substack{y \sim \eta \\ g \sim \mathcal{N}(0,1)}} \left[ \frac{y^2}{\left(\theta_\sigma(\beta, t) - \sigma\left(\frac{m_1}{m_2}\beta y + g\right)\right)^2} \right] \right)^{-1} H'_\nu(\theta_\sigma(\beta, t)).$$

Finally, since

$$\lim_{t \downarrow t_*(\beta)} \frac{dH_\nu(\theta_\sigma(\beta, t))}{dt} = 0,$$

$f_\sigma$ is also differentiable at $t = 0$. $\qquad\square$

**Lemma C.8.** *Let $(f_n)$ be a sequence of convex functions defined on an open interval $I$ containing $0$. Assume that $f_n(x) \to f(x)$ for every $x \in I$. Suppose further that each $f_n$ and $f$ is differentiable at $0$. Then,*

$$\lim_{n \to \infty} f'_n(0) = f'(0).$$

*Proof.* By convexity of $f_n$, for any $0 < h < \epsilon$, we have

$$\frac{f_n(0) - f_n(-h)}{h} \leq f'_n(0) \leq \frac{f_n(h) - f_n(0)}{h}.$$

Taking the limit as $n \to \infty$ yields

$$\frac{f(0) - f(-h)}{h} \leq \liminf_{n \to \infty} f'_n(0) \leq \limsup_{n \to \infty} f'_n(0) \leq \frac{f(h) - f(0)}{h}.$$

Since this holds for any $0 < h < \epsilon$, letting $h \to 0^+$ gives

$$f'(0) \leq \liminf_{n \to \infty} f'_n(0) \leq \limsup_{n \to \infty} f'_n(0) \leq f'(0).$$

Hence, $\lim_{n \to \infty} f'_n(0) = f'(0)$. This completes the proof. $\qquad\square$

## C.6 Limitation of degree methods: Proof of Theorem 3.5

We follow the approach developed in [MRZ15, BMV$^+$18, PWBM18, PWB20] for proving statistical indistinguishability.

**Lemma C.9** (Second moment method for contiguity, Lemma 1.13 of [KWB19])**.** *Consider two probability distributions $\mathbb{P} = (\mathbb{P}_n), \mathbb{Q} = (\mathbb{Q}_n)$ over a common sequence of measurable spaces $S_n$. Write $L_n(t)$ for the likelihood ratio $\frac{d\mathbb{P}_n}{d\mathbb{Q}_n}(t)$.*

*If $\limsup_{n\to\infty} \mathbb{E}_{t\sim\mathbb{Q}_n}\left[L_n(t)^2\right] < \infty$, then $\mathbb{P}$ is* contiguous *to $\mathbb{Q}$, meaning that whenever $\mathbb{Q}_n(A_n) \to 0$ for a sequence of events $(A_n)$, then $\mathbb{P}_n(A_n) \to 0$ as well.*

**Theorem C.10.** *Define the sequence of probability distributions $\mathbb{Q}_{n,\beta} := \mathrm{Law}(\boldsymbol{Y}\boldsymbol{1}/\sqrt{n})$, where $\boldsymbol{Y} \sim \mathbb{P}_{n,\beta}$ is drawn from the Sparse Biased PCA model defined in Definition 1.1, with the additional assumption that $p = p(n) = O(n^{-1/2})$ and that $\eta$ has bounded support. Then, for any $\beta > 0$, the sequence of distributions $\mathbb{Q}_{n,\beta}$ is contiguous to the sequence $\mathbb{Q}_{n,0}$. In particular, no function of $\boldsymbol{Y}\boldsymbol{1}$ can achieve strong detection in such a model.*

**Remark C.11.** *Since the Gaussian Planted Submatrix model of Example 1.2 satisfies the additional assumptions above, this proposition establishes a result that is more general than Theorem 3.5 from the main text.*

*Proof.* Let $\rho$ denote the distribution of $\boldsymbol{y}$ from Definition 1.1. Recall that $\frac{\boldsymbol{Y}\boldsymbol{1}}{\sqrt{n}} = \widehat{\boldsymbol{W}}\boldsymbol{1} + \beta\frac{\sum_i y_i}{\|\boldsymbol{y}\|^2}\boldsymbol{y}$, so

$$\mathbb{Q}_{n,\beta} = \mathrm{Law}\left(\boldsymbol{t} + \beta\frac{\sum_i y_i}{\|\boldsymbol{y}\|^2}\boldsymbol{y}\right), \quad \boldsymbol{t} \sim \mathcal{N}\left(\boldsymbol{0}, \boldsymbol{I} + \frac{\boldsymbol{1}\boldsymbol{1}^\top}{n}\right), \quad \boldsymbol{y} \sim \rho.$$

Let $\delta > 0$, and define another sequence of distributions

$$\widetilde{\mathbb{Q}}_{n,\beta} = \mathrm{Law}\left(\boldsymbol{t} + \beta\mathbb{1}\{\mathcal{E}_n\}\frac{\sum_i y_i}{\|\boldsymbol{y}\|^2}\boldsymbol{y}\right),$$

where the event sequence

$$\mathcal{E}_n := \left\{\left|\frac{\sum_{i=1}^n y_i}{\|\boldsymbol{y}\|^2} - \frac{m_1}{m_2}\right| < \delta\right\}.$$

Notice that $\frac{\sum_{i=1}^n y_i}{np} \xrightarrow{\text{(a.s.)}} m_1$ and $\frac{\|\boldsymbol{y}\|^2}{np} \xrightarrow{\text{(a.s.)}} m_2$ by Lemma B.7 under the independent entries sparsity model, or by the Strong Law of Large Numbers under the random subset sparsity model. Therefore,

$$\delta_{\mathrm{TV}}(\mathbb{Q}_{n,\beta}, \widetilde{\mathbb{Q}}_{n,\beta}) \le \mathbb{P}\left(\mathcal{E}_n^c\right) \to 0.$$

Let $\widetilde{L}_n(\boldsymbol{t})$ denote the likelihood ratio $\frac{d\widetilde{\mathbb{Q}}_{n,\beta}}{d\mathbb{Q}_{n,0}}(\boldsymbol{t})$. It suffices to show

$$\limsup_{n\to\infty} \mathbb{E}_{\boldsymbol{t}\sim\mathbb{Q}_{n,0}}\left[\widetilde{L}_n(\boldsymbol{t})^2\right] < \infty. \tag{8}$$

Indeed, if (8) holds, then by Lemma C.9, whenever $\mathbb{Q}_{n,0}(A_n) \to 0$ for a sequence of events $(A_n)$, it follows that $\widetilde{\mathbb{Q}}_{n,\beta}(A_n) \to 0$. Hence,

$$\mathbb{Q}_{n,\beta}(A_n) \le \widetilde{\mathbb{Q}}_{n,\beta}(A_n) + \delta_{\mathrm{TV}}\left(\mathbb{Q}_{n,\beta}, \widetilde{\mathbb{Q}}_{n,\beta}\right) \to 0,$$

establishing contiguity of $\mathbb{Q}_{n,\beta}$ to $\mathbb{Q}_{n,0}$, and thus statistical indistinguishability by [KWB19, Proposition 1.12].

To prove (8), write $\Sigma := \left(\boldsymbol{I} + \frac{\boldsymbol{1}\boldsymbol{1}^\top}{n}\right)^{-1} = \boldsymbol{I} - \frac{\boldsymbol{1}\boldsymbol{1}^\top}{2n}$. Let $\langle\cdot,\cdot\rangle_\Sigma$ denote the inner product defined by $\langle\boldsymbol{x},\boldsymbol{y}\rangle_\Sigma = \boldsymbol{x}^\top\Sigma\boldsymbol{y}$, and $\|\cdot\|_\Sigma$ the corresponding norm. Define $\widetilde{\boldsymbol{y}} := \mathbb{1}\{\mathcal{E}\}\frac{\sum_i y_i}{\|\boldsymbol{y}\|^2}\boldsymbol{y}$. Then,

$$\widetilde{L}_n(\boldsymbol{t}) = \frac{\mathbb{E}_{\widetilde{\boldsymbol{y}}}\left[\exp\left(-\frac{1}{2}\|\boldsymbol{t} - \beta\widetilde{\boldsymbol{y}}\|_\Sigma^2\right)\right]}{\exp\left(-\frac{1}{2}\|\boldsymbol{t}\|_\Sigma^2\right)}$$

$$= \mathbb{E}_{\widetilde{\boldsymbol{y}}}\left[\exp\left(-\frac{\beta^2}{2}\|\widetilde{\boldsymbol{y}}\|_\Sigma^2 + \beta\langle\boldsymbol{t},\widetilde{\boldsymbol{y}}\rangle_\Sigma\right)\right].$$

Therefore,

$$\mathbb{E}_{t\sim\mathcal{N}(0,\Sigma^{-1})}\left[\widetilde{L}_n(t)^2\right] = \mathbb{E}_{\widetilde{y},\widetilde{y}'}\left[\exp\left(-\frac{\beta^2}{2}\left(\|\widetilde{y}\|_\Sigma^2 + \|\widetilde{y}'\|_\Sigma^2\right)\right)\mathbb{E}_t\left[\exp\left(\beta\langle t,\widetilde{y}+\widetilde{y}'\rangle_\Sigma\right)\right]\right]$$

where $\widetilde{y},\widetilde{y}'$ are two independent copies of $y$ (sometimes called "replicas" in this context)

$$= \mathbb{E}_{\widetilde{y},\widetilde{y}'}\left[\exp\left(-\frac{\beta^2}{2}\left(\|\widetilde{y}\|_\Sigma^2 + \|\widetilde{y}'\|_\Sigma^2 - \|\widetilde{y}+\widetilde{y}'\|_\Sigma^2\right)\right)\right]$$

$$= \mathbb{E}_{\widetilde{y},\widetilde{y}'}\left[\exp\left(\beta^2\langle\widetilde{y},\widetilde{y}'\rangle_\Sigma\right)\right]$$

$$= \mathbb{E}_{y,y'}\left[\exp\left(\beta^2\mathbb{1}\{\mathcal{E}_n\cap\mathcal{E}_n'\}\frac{\sum_i y_i}{\|y\|^2}\frac{\sum_i y_i'}{\|y'\|^2}\left(y^\top y' - \frac{(\sum_i y_i)(\sum_i y_i')}{2n}\right)\right)\right]$$

where $y,y'$ are two independent replicas of $y$, and $\mathcal{E}_n,\mathcal{E}_n'$ are the events corresponding to independent sequences $y^{(n)},y'^{(n)}$ respectively.

$$\leq \mathbb{E}_{y,y'}\left[\exp\left(\beta^2\left(\frac{m_1}{m_2}+\delta\right)^2 y^\top y'\right)\right]$$

$$= \left(1 + \frac{p^2 n}{n}\left(\mathbb{E}_{z_1,z_1'\sim\eta}\left[\exp\left(\beta^2\left(\frac{m_1}{m_2}+\delta\right)^2 z_1 z_1'\right)\right]-1\right)\right)^n$$

Suppose $0 \leq z \leq M$ almost surely if $z \sim \eta$. Then for $n$ sufficiently large such that $p^2 n \leq C$ for some constant $C > 0$,

$$\leq \left(1 + \frac{C}{n}\left(\exp\left(4\beta^2\left(\frac{m_1}{m_2}+\delta\right)^2 M^2\right)-1\right)\right)^n$$

$$\to \exp\left(C\exp\left(4\beta^2\left(\frac{m_1}{m_2}+\delta\right)^2 M^2\right)-1\right)$$

which completes the proof for (8). $\qquad\square$

## D  Specific models

### D.1  Gaussian Planted Submatrix model

We give some additional discussion of the special case discussed in Example 1.2. As a reminder, this special case is obtained by taking $\eta = \delta_1$ and sparsity level $p(n) = \beta/\sqrt{n}$ under the Random Subset sparsity model.

#### D.1.1  Specialization of main results

In this special case, our main general results take the following form, obtained by substituting $\eta = \delta_1, m_1 = m_2 = 1$ into Theorem 3.3.

First, for the eigenvalues of the signal part $X^{(n)}$, we have that almost surely

$$\mathrm{esd}\left(X^{(n)}\right) \xrightarrow{\text{(w)}} \sigma(\mathcal{N}(0,1)).$$

For the largest eigenvalue, we have almost surely $\lambda_1(X^{(n)}) \to \theta$ where $\theta$ solves the equation

$$\mathbb{E}_{g\sim\mathcal{N}(0,1)}\left[\frac{1}{\theta - \sigma(\beta+g)}\right] = \frac{1}{\beta} \tag{9}$$

if such $\theta > \mathrm{edge}^+(\sigma)$ exists, and $\theta = \mathrm{edge}^+(\sigma)$ otherwise. Finally, we find that the associated $\sigma$-Laplacian has an outlier eigenvalue if and only if $\beta > \beta_* = \beta_*(\sigma)$, where this $\beta_*$ solves

$$\mathbb{E}_{g\sim\mathcal{N}(0,1)}\left[\frac{1}{(\theta_\sigma(\beta_*) - \sigma(g))^2}\right] = 1. \tag{10}$$

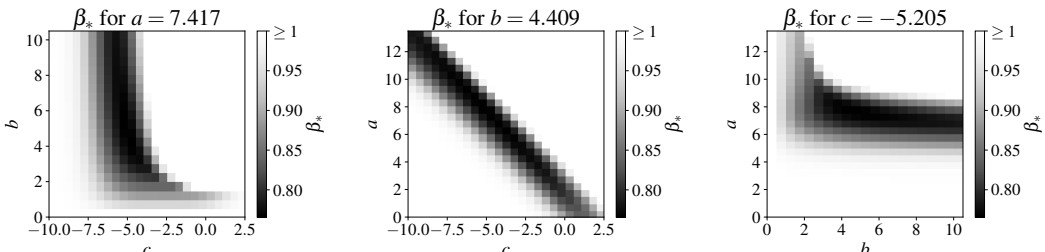

Figure 3: $\beta_*(\sigma)$ for the family of "Z"-shaped piecewise linear $\sigma$ given in (11), with each parameter fixed at its optimal value and the other two varying.

### D.1.2 Discussion of numerically optimized nonlinearities

We now describe how we derive the numerical result of Theorem 3.4 from this theoretical characterization. This amounts to finding a sufficiently "good" $\sigma$, i.e., one that has a small value of $\beta_*(\sigma)$. We take it for granted here that $\beta_*(\sigma)$ can be estimated numerically; further details about this are given in Appendix E. We focus instead on the search methods that we use to look for a good $\sigma$.

As we have briefly described in Section 4, we consider three different approaches to this problem.

**Explore simple class of** $\sigma$  We first consider a parametric family of piecewise linear functions characterized by a "Z" shape:

$$\sigma(x; a, b, c) := \begin{cases} 0 & \text{if } x < c, \\ b \cdot \frac{x-c}{a} & \text{if } c \leq x \leq a + c, \\ b & \text{if } x > a + c. \end{cases} \tag{11}$$

Given that a vertical shift of $\sigma$ results in an essentially equivalent algorithm (since it merely shifts the $\sigma$-Laplacian by a multiple of the identity, we fix the minimum value of $\sigma$ to 0. We start with optimizing the objective function $\beta_*(\sigma)$ with respect to the parameters $(a, b, c)$ using the Nelder-Mead black-box optimization method. The optimized result is presented in Figure 2(b). Subsequently, we investigate the individual effects of varying each parameter $(a, b, c)$ on $\beta_*(\sigma)$. In Figure 3, we illustrate the result of fixing any one parameter at its optimal value and plotting, as a heatmap, the value of $\beta_*$ with respect to the other two parameters.

The resulting heatmaps demonstrate that $\beta_*$ is relatively robust with respect to all parameters. As the middle plot shows, perhaps the most clearly important quantity is $a + c$, the right endpoint of the sloped part of the "Z". On the other hand, the first and third plots show that the value of $b$, the height of the "Z", is relatively unimportant provided $a$ and $c$ are chosen well.[7]

Next, we examine another simple but smoother parametric family of functions,

$$\sigma(x; a, b) := a \tanh(bx).$$

Similar to the previous case, we optimize $\beta_*(\sigma)$ with respect to the parameters $(a, b)$ using the Nelder-Mead method. The resulting optimized parameters and the corresponding objective function value are presented in Figure 2(c). We note that optimizing within this family yields a slightly improved $\beta_*$ compared to the "Z"-shaped choice of $\sigma$. For this $\sigma$, we plot the phase transitions of the top eigenvalue and eigenvectors on random samples from the planted submatrix model in Figure 4. The results show that our theoretical predictions for the critical signal strength $\beta_*$ and the outlier eigenvalue are accurate, and that the eigenvector overlap exhibits a phase transition at the same $\beta_*$.

**Nelder-Mead optimization over step functions**  Next, we explore parameterizing $\sigma$ using more complex function classes and optimizing using black-box optimization. We first consider running Nelder-Mead optimization over $2n$ parameters controlling a step function. Let $\boldsymbol{a} = [a_1, \ldots, a_n]$ and $\boldsymbol{b} = [b_1, \ldots, b_n]$, where we impose the constraint that all parameters, except $a_1$, are non-negative.

---

[7]We do not illustrate it for the sake of the legibility of the plot, but the performance of the algorithm is not entirely indifferent to $b$: for very large values, $\beta_*(\sigma)$ does increase, with the optimal $\beta_*$ occurring around 4.4.

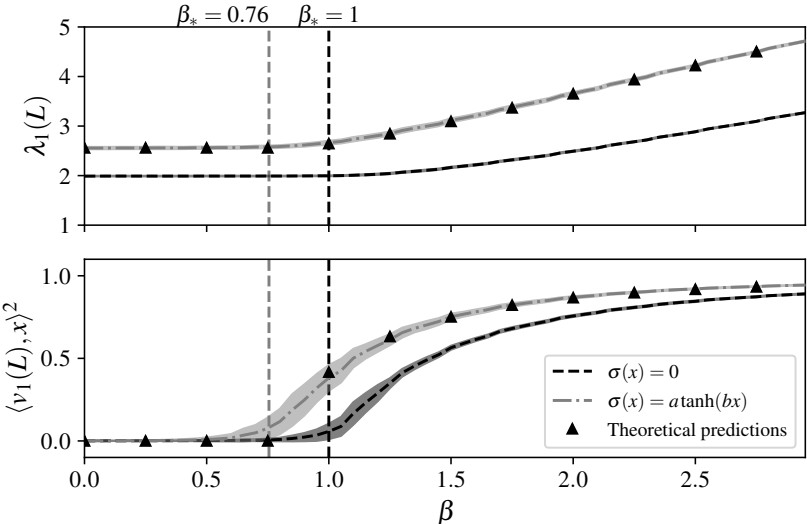

Figure 4: An illustration of the phase transition of top eigenvalue and eigenvector. Each point of the plot shows the average over 500 random inputs of the largest eigenvalue and the squared correlation of the top eigenvector with the signal $\boldsymbol{x}$ for a nonlinear Laplacian $\boldsymbol{L}_\sigma(\boldsymbol{Y})$ with $\boldsymbol{Y}$ drawn from the Gaussian Planted Submatrix model, for $\sigma = 0$ (dark line) and $\sigma$ the best "S-shaped" nonlinearity found by black-box optimization (light line). Error bars have width of plus/minus one standard deviation over these trials. We also include the theoretical prediction from Theorem 3.3.

We define discontinuity points $x_k = \sum_{j=1}^k a_j$ for $1 \le k \le n$. The step function $\sigma(x; \boldsymbol{a}, \boldsymbol{b})$ is then defined as:

$$\sigma(x; \boldsymbol{a}, \boldsymbol{b}) := \begin{cases} 0 & \text{if } x < x_1 \\ \sum_{j=1}^k b_j & \text{if } x_k \le x < x_{k+1}, \quad 1 \le k \le n-1 \\ \sum_{j=1}^n b_j & \text{if } x \ge x_n \end{cases}$$

The primary motivation for considering step functions is that numerical integration simplifies to summation in this case, which significantly accelerates computation. For example, in computing $\theta_\sigma(\beta)$, if $\sigma$ is as above, we may rewrite

$$\mathop{\mathbb{E}}_{g \sim \mathcal{N}(0,1)} \left[ \frac{1}{\theta - \sigma(\beta + g)} \right] = \sum_{k=0}^n \mathop{\mathbb{P}}_{g \sim \mathcal{N}(0,1)} [g \in (x_k, x_{k+1})] \cdot \frac{1}{\theta - \sigma(\beta + \sum_{j=1}^k b_j)}$$

where we set $x_0 = -\infty$ and $x_{n+1} = +\infty$. The probabilities on the right-hand side can be computed in terms of the Gaussian error function, for which highly optimized implementations are provided in many numerical computing libraries, making one calculation of this integral far faster than numerically integrating the left-hand side. Recall also that to compute $\theta_\sigma(\beta)$ we must perform a root-finding calculation on this function of $\theta$, requiring many evaluations; further, to compute $\beta_*(\sigma)$ we must perform *that* root-finding operation many times as the "inner loop" of another root-finding calculation, making the total number of times that we need to compute the above kind of integral quite large. The computational efficiency of $\sigma$ a step function makes the application of the Nelder-Mead method feasible for optimizing over a large number of parameters. The optimized result for this step function parameterization with $n = 16$ is presented in the Figure 2(e).

**Multi-Layer Perceptron learned from data** We also consider parameterizing $\sigma$ using a Multi-Layer Perceptron (MLP) with $L$ layers and $h$ hidden channels. Specifically, the output $\sigma(x) = \boldsymbol{z}^{(L)}$ (viewed as a $1 \times 1$ matrix) is defined by the following recursive relations:

$$\boldsymbol{z}^{(0)} = x$$

$$\boldsymbol{z}^{(\ell)} = \rho \left( \boldsymbol{W}^{(\ell)} \boldsymbol{z}^{(\ell-1)} + \boldsymbol{b}^{(\ell)} \right) \quad \text{for } \ell = 1, \dots, L$$

where $\boldsymbol{W}^{(1)} \in \mathbb{R}^{h \times 1}, \boldsymbol{b}^{(1)} \in \mathbb{R}^h, \boldsymbol{W}^{(\ell)} \in \mathbb{R}^{h \times h}, \boldsymbol{b}^{(\ell)} \in \mathbb{R}^h$ for $\ell = 2, \ldots, L-1$, and $\boldsymbol{W}^{(L)} \in \mathbb{R}^{1 \times h}, \boldsymbol{b}^{(L)} \in \mathbb{R}$ are learnable weight matrices and bias vectors, respectively. The function $\rho$ denotes a fixed, element-wise nonlinearity, for which we use either the tanh or ReLU activation function. In our experiment, we use $L = 8, h = 20$, and the tanh activation function.

To find the optimized $\sigma$, we generate a dataset of i.i.d. pairs $(\boldsymbol{Y}_i, y_i)$ according to the following procedure: We fix parameters $n_0$ and $\beta_0$. For each data point $i$, we first draw a binary label $y_i \overset{\text{i.i.d.}}{\sim} \text{Bernoulli}(1/2)$, where $y_i = 1$ indicates the presence of a planted signal and $y_i = 0$ indicates its absence. Subsequently, we generate the observed symmetric matrix $\boldsymbol{Y}_i \in \mathbb{R}^{n \times n}_{\text{sym}}$ from the Planted Submatrix model, $\boldsymbol{Y}_i \sim \mathbb{P}_{n_0, \beta}$ in our previous notation. If $y_i = 1$, the matrix is generated with a planted signal strength of $\beta = \beta_0$; if $y_i = 0$, the matrix is generated with $\beta = 0$. In our experiments, we use $n_0 = 100, \beta_0 = 1.3$.

Our neural network model processes the input $\boldsymbol{Y}_i$ by first passing each element through the MLP parameterized by $\sigma$. Then, it computes the $\sigma$-Laplacian matrix, $\boldsymbol{L}_\sigma(\boldsymbol{Y}_i) = \boldsymbol{Y}_i + \text{diag}(\sigma(\boldsymbol{Y}_i \boldsymbol{1}))$. We then utilize the built-in eigenvalue computation functionality in PyTorch, which supports gradient flow, to obtain the top eigenvalue of the $\sigma$-Laplacian. Finally, this top eigenvalue is passed through a learnable linear map $f(x) = \alpha x + \gamma$ to produce a scalar output. Thus in summary we compute

$$\alpha \lambda_1(\boldsymbol{Y}_i + \text{diag}(\sigma(\boldsymbol{Y}_i \boldsymbol{1}))) + \gamma \in \mathbb{R}.$$

We train this to solve the task of classifying $\boldsymbol{Y}_i$ according to the (binary) value of $y_i$ by minimizing the standard binary cross-entropy loss function using stochastic gradient descent.

The key distinction between this approach and the previous ones lies in the fact that **we do not directly optimize the theoretical $\beta_*(\sigma)$ derived earlier**. Instead, our optimization focuses on the classification performance on datasets with a fixed, small matrix size $n$. We then validate the asymptotic performance of the learned $\sigma$ by computing the corresponding theoretical $\beta_*$ using our numerical procedure outlined above.

We train the above neural network with 10 different random initializations, post-process the $\sigma$ learned it to make it monotone, and compute the corresponding $\beta_*$. We report the $\sigma$ achieving the smallest $\beta_*$ in Figure 2(d). Encouragingly, despite being trained only on finite $n$, the learned $\sigma$ performs comparably to the other methods. We take this as an indication that $\sigma$ can effectively be learned from relatively small datasets without depending on the theoretical analysis of $\beta_*(\sigma)$, making nonlinear Laplacians a promising method for more realistic applications than these models of synthetic data.

### D.2 Planted Clique model

The Planted Clique problem is similar in spirit to the Gaussian Planted Submatrix problem, but is defined over random graphs rather than matrices. We view graphs on vertext set $[n]$ as being encoded in matrix form by a graph $G$ giving rise to the so-called Seidel adjacency matrix $\boldsymbol{Y} \in \mathbb{R}^{n \times n}_{\text{sym}}$,

$$Y_{ij} = \begin{cases} +1 & \text{if nodes } i, j \text{ are connected,} \\ -1 & \text{if nodes } i, j \text{ are not connected,} \\ 0 & \text{if } i = j. \end{cases}$$

Thus, when we describe a random graph below, we may equivalently view it as a random such matrix.

**Definition D.1** (Planted Clique). *Let $n \geq 1$ and $\beta \geq 0$. We observe a random simple graph $G = (V = [n], E)$ constructed as follows: first, draw $G$ from the Erdős-Rényi random graph distribution, i.e., each pair of distinct $u, v \in V$, take $(u, v) \in E$ independently with probability $1/2$. Then, choose a subset $S \subseteq [n]$ of size $|S| = \beta\sqrt{n}$ uniformly at random, and add an edge to $G$ between each $u, v \in S$ distinct if that edge is not in $G$ already. Thus in the resulting $G$, the vertices of $S$ form a* clique *or complete subgraph.*

Taking $\boldsymbol{x} = \widehat{\boldsymbol{1}}_S$ as for Planted Submatrix, a simple calculation shows that we still have

$$\mathbb{E}\left[\boldsymbol{Y} \mid \boldsymbol{x}\right] \approx \boldsymbol{1}_S \boldsymbol{1}_S^\top = \beta\sqrt{n} \cdot \boldsymbol{x}\boldsymbol{x}^\top.$$

While the entries of $\boldsymbol{Y} - \mathbb{E}\left[\boldsymbol{Y} \mid \boldsymbol{x}\right]$ are not exactly i.i.d. anymore, it turns out that this matrix is sufficiently "Wigner-like" that many of the phenomena discussed for direct spectral algorithms in Section 3.1 still hold for the Planted Clique model.

In particular, it is well-known as folklore that a version of Theorem 3.1 on the BBP transition holds in this case. The idea of showing this is to note that the matrix $\boldsymbol{W} := \boldsymbol{Y} - \mathbb{E}\left[\boldsymbol{Y} \mid \boldsymbol{x}\right]$ is a Wigner matrix with i.i.d. Rademacher entries drawn uniformly from $\{\pm 1\}$ (above the diagonal), except that the entries indexed by $S$ are identically zero. We may write this as $\boldsymbol{W} = \boldsymbol{W}^{(0)} - \boldsymbol{W}_{S,S}^{(0)}$ for $\boldsymbol{W}^{(0)}$ truly a Wigner matrix. But, we have with high probability $\boldsymbol{W}^{(0)} = \Theta(n^{1/2})$, while $\|\boldsymbol{W}_{S,S}^{(0)}\| = \Theta(n^{1/4})$, since the latter is just a Wigner matrix of dimension $\beta\sqrt{n}$ padded by zeroes (see Proposition B.14). Thus this modification is of relatively negligible spectral norm, and routine perturbation inequalities (in particular Weyl's inequality, our Proposition B.15, for the eigenvalues) thus show that the result of Theorem 3.1 holds for $\boldsymbol{Y}$ drawn from the Planted Clique model.

As an aside, the Planted Clique problem does have some non-trivial distinctions from the Gaussian Planted Submatrix problem. For example, because of the special discrete structure of the Planted Clique problem, for $\beta$ sufficiently large it is also possible to introduce a post-processing step that improves weak recovery to exact recovery (i.e., recovering the exact value of the set $S$), as detailed in [AKS98].

Unfortunately, in this non-Gaussian setting the device of compression used to make the signal and noise parts of $\boldsymbol{Y}$ exactly independent (Section C.3) does not apply to the Planted Clique model, and we have not been able to find a substitute for that part of our argument. On the other hand, we do still expect $(\widehat{\boldsymbol{Y}} - \boldsymbol{1}_S \boldsymbol{1}_S^\top)\boldsymbol{1}$ to be approximately a Gaussian random vector in the Planted Clique model by the central limit theorem, since each entry is approximately a normalized sum of i.i.d. random variables drawn uniformly at random from $\{\pm 1\}$. Thus, while we are confident that the analogy between Planted Clique and Gaussian Planted Submatrix holds sufficiently precisely for Conjecture 3.6 to hold (as also supported by numerical experiments analogous to those we present for the Gaussian Planted Submatrix model), we have not been able to prove this and leave the Conjecture for future work.

Meanwhile, we numerically validate Conjecture 3.6 using the same setup as in Figure 4, as shown in Figure 5 below. The Conjecture is merely the statement that these two Figures should look identical as $n \to \infty$, which we observe even for these experiments with finite $n$.

### D.3 Heuristic discussion of dense signals

Our choice of a model of Biased PCA in Definition 1.1 prescribes that the signal vector $\boldsymbol{x}$ must be sparse; in particular, we forbid the situation where $\|\boldsymbol{x}\|_0 = \Theta(n)$. This regime is different from that treated by, for instance, the previous work [MR15], which instead allows only such signals of constant sparsity, though many of their results concern the limit of this sparsity decreasing to zero (and thus in a sense approaching our regime of $\|\boldsymbol{x}\|_0 = o(n)$).

In fact, it remains unclear to us to what extent our results should transfer to the case of denser signals. Recall that our intuition is that the matrix $\boldsymbol{D} = \text{diag}(\sigma(\widehat{\boldsymbol{Y}}\boldsymbol{1}))$ may be treated as approximately independent of $\widehat{\boldsymbol{Y}}$ for the purposes of studying the spectrum of their sum. This is in some sense encoded in our use of the compression operation (Section C.3)—when we project away the $\boldsymbol{1}$ direction from $\widehat{\boldsymbol{Y}}$, we intuitively believe that the important structure of $\widehat{\boldsymbol{Y}}$ remains unchanged.

At a technical level, this is simply false once $\boldsymbol{x}$ is a dense ($\Theta(n)$ non-zero entries) non-negative vector: $\boldsymbol{x}$ then has positive correlation with $\boldsymbol{1}$, so applying compression will reduce the signal strength in $\widehat{\boldsymbol{Y}}$. Thus we should not expect a spectral algorithm using the compressed $\sigma$-Laplacian to be as powerful as one using the uncompressed $\sigma$-Laplacian in this dense regime. Between these two algorithms, it seems that using the uncompressed $\sigma$-Laplacian will be superior, but we have not found a way to carry out analogous analysis without using compression.

Let us sketch how our results would look for an example of such a model, supposing that our technique based on the random matrix analysis of [CDMFF11] applied. Consider the case where $\eta = |\mathcal{N}(0,1)|$, the law of the absolute value of a standard Gaussian random variable, and $p(n) = 1$, so that the signal is not sparsified at all. Thus we have that $y_i \overset{\text{i.i.d.}}{\sim} \eta$ and $\boldsymbol{x} = \boldsymbol{y}/\|\boldsymbol{y}\|$. We expect to have

$$\widehat{\boldsymbol{Y}}\boldsymbol{1} = \beta\langle\boldsymbol{x},\boldsymbol{1}\rangle\boldsymbol{x} + \widehat{\boldsymbol{W}}\boldsymbol{1}.$$

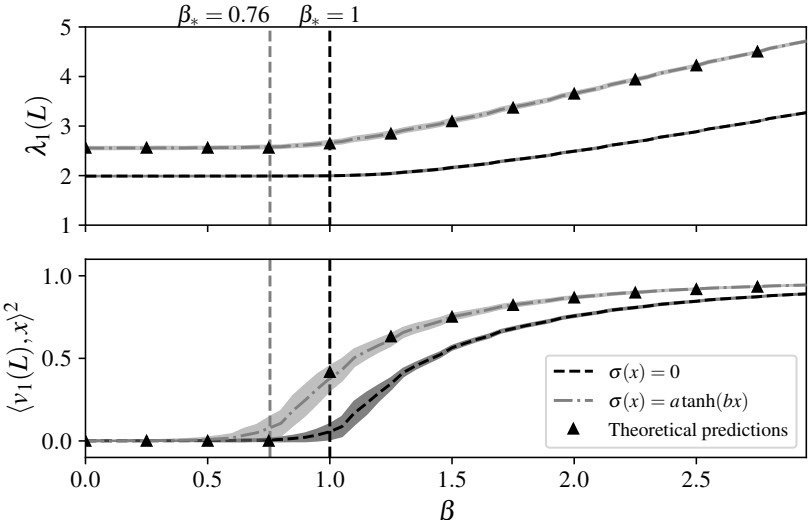

Figure 5: An illustration of the phase transition of top eigenvalue and eigenvector for the Planted Clique model. Each point of the plot shows the average over 500 random inputs of the largest eigenvalue and the squared correlation of the top eigenvector with the signal $x$ for a nonlinear Laplacian $L_\sigma(Y)$ with $Y$ drawn from the Gaussian Planted Submatrix model, for $\sigma = 0$ (dark line) and $\sigma$ the best "S-shaped" nonlinearity found by black-box optimization (light line). Error bars have width of plus/minus one standard deviation over these trials. We also incorporate the theoretical prediction for the Gaussian Planted Submatrix model from Theorem 3.3. The numerical outputs align with the theoretical predictions, supporting Conjecture 3.6.

As in the sparse case, the second term has independent entries approximately distributed as $\mathcal{N}(0,1)$ and in the first term $\langle x, 1 \rangle \approx \sqrt{2/\pi}$.

Already here the first difference with the sparse case appears: in that case the first term above does not affect the weak limit of $\mathrm{ed}(\widehat{Y}1)$ and thus of $\mathrm{esd}(X^{(n)})$, since only $o(n)$ entries of $x$ are non-zero. Here, however, this term does play a role, and instead of $\mathrm{esd}(X^{(n)}) \xrightarrow{(w)} \sigma(\mathcal{N}(0,1))$ as in Lemma 3.7, we should expect the different limit

$$\mathrm{esd}(X^{(n)}) \xrightarrow{(w)} \nu = \nu_{\beta,\sigma} := \mathrm{Law}(\sigma(\beta\sqrt{2/\pi}|g| + h)) \quad \text{for} \quad g, h \overset{\text{i.i.d.}}{\sim} \mathcal{N}(0,1),$$

which notably depends on $\beta$.

Adjusting the rest of Lemma 3.7 accordingly, we expect $\lambda_1(X^{(n)}) \to \theta = \theta_\sigma(\beta)$ where this $\theta$ solves the equation

$$\mathbb{E}_{g,h\sim\mathcal{N}(0,1)} \left[ \frac{g^2}{\theta - \sigma(\beta\sqrt{2/\pi}|g| + h)} \right] = \frac{1}{\beta}$$

if such $\theta > \mathrm{edge}^+(\sigma)$ exists, and $\theta = \mathrm{edge}^+(\sigma)$ otherwise. Similarly, $\beta_* = \beta_*(\sigma)$ would then solve

$$\mathbb{E}_{g,h\sim\mathcal{N}(0,1)} \left[ \frac{1}{(\theta_\sigma(\beta_*) - \sigma(\beta_*\sqrt{2/\pi}|g| + h))^2} \right] = 1,$$

and when $\beta > \beta_*$, then we would predict

$$\lambda_1(L^{(n)}) \to \theta_\sigma(\beta) + \mathbb{E}_{g,h\sim\mathcal{N}(0,1)} \left[ \frac{1}{\theta_\sigma(\beta) - \sigma(\beta\sqrt{2/\pi}|g| + h)} \right] > \mathrm{edge}^+(\mu_{\mathrm{sc}} \boxplus \nu).$$

**Remark D.2.** *In fact, since the limiting esd of $L^{(n)}$ will be $\mu_{\mathrm{sc}} \boxplus \nu_{\beta,\sigma}$ in this case, which depends non-trivially on $\beta$ (we do not give a proof but this much is not difficult to show using basic free probability tools), it is easy to distinguish the case $\beta = 0$ from that of $\beta > 0$, achieving strong*

*detection, by merely considering, e.g., a suitable moment of the esd, which is an expression of the form $\frac{1}{n}\operatorname{Tr}(\widehat{\boldsymbol{Y}}^k)$. It is also easy to achieve weak recovery, since the vector $\boldsymbol{1}/\sqrt{n}$ is macroscopically correlated with any dense non-negative vector $\boldsymbol{x}$. So, the questions of thresholds for strong detection and weak recovery we have been concerned with in this work trivialize for models with dense signals. A more interesting question in this setting would be for what signal strength $\beta$ it becomes possible to estimate $\boldsymbol{x}$ with a correlation superior to $\boldsymbol{1}/\sqrt{n}$. However, per the discussion in Appendix C.5, it seems challenging even in the sparse case to understand what such correlation is achieved by a $\sigma$-Laplacian spectral algorithm for recovery.*

# E    Details of numerical methods

Below we give some details of the more involved numerical computations we have referred to throughout. The code used to generate the results presented in this paper is also available online anonymously at `https://github.com/yuxinma98/NonlinearLaplacian`.

### E.1    Estimation of additive free convolution

We describe how we compute the analytic prediction of the empirical spectral distribution of a $\sigma$-Laplacian, as illustrated in Figure 1. Recall that, by Lemma 3.8, the limiting density of this distribution is of the form $\nu \boxplus \mu_{\mathrm{sc}}$, where $\nu$ is a compactly supported measure with a known density, in our case $\nu = \sigma(\mathcal{N}(0,1))$, and $\mu_{\mathrm{sc}}$ is the semicircle distribution.

The distribution of $\nu \boxplus \mu_{\mathrm{sc}}$ is characterized by the equation satisfied by its Stieltjes transform: we have

$$G_{\nu\boxplus\mu_{\mathrm{sc}}}^{-1}(z) - \frac{1}{z} = R_{\nu\boxplus\mu_{\mathrm{sc}}}(z) = R_\nu(z) + z = G_\nu^{-1}(z) - \frac{1}{z} + z.$$

Rearranging this gives the implicit equation

$$G_{\nu\boxplus\mu_{\mathrm{sc}}}(z) = G_\nu(z - G_{\nu\boxplus\mu_{\mathrm{sc}}}(z)). \tag{12}$$

Numerically, the density of $\nu \boxplus \mu_{\mathrm{sc}}$ can be computed by the following procedure. Fix a grid of complex numbers $z_j = x_j + i\epsilon$ close to the real axis. Call $G_j = G_{\nu\boxplus\mu_{\mathrm{sc}}}(z_j)$. These numbers satisfy the fixed-point equation $G_j = \int_{-\infty}^{\infty} \frac{1}{z_j - G_j - \sigma(g)} \frac{1}{\sqrt{2\pi}} e^{-g^2/2} dg$, hence it can be computed numerically by iterating

$$G_j^{(t+1)} = \int_{-\infty}^{\infty} \frac{1}{z_j - G_j^{(t)} - \phi(g)} \frac{1}{\sqrt{2\pi}} e^{-g^2/2} dg$$

until convergence. To recover the density of $\nu \boxplus \mu_{\mathrm{sc}}$, we can then use the Stieltjes inversion formula: if this density is $\rho$, then

$$\rho(x) = \lim_{\epsilon \to 0} -\frac{1}{\pi} \mathsf{Im}(G_{\nu\boxplus\mu_{\mathrm{sc}}}(x + i\epsilon))$$

In particular, for $\epsilon$ small enough, we should have

$$\rho(x_j) \approx -\frac{1}{\pi} \mathsf{Im}(G_j),$$

giving us a sequence of estimates of the density at the points $x_j$. We compute the integrals above using the standard numerical integration methods built into the `NumPy` library.

### E.2    Computation of $\theta_\sigma(\beta)$ and $\beta_*(\sigma)$

Recall that, per Theorem 3.3, each of the quantities $\theta_\sigma(\beta)$ and $\beta_*(\sigma)$ is determined by finding the root of a suitable function. For example, $\theta_\sigma(\beta)$ is the $\theta$ that solves $F_{\beta,\sigma}(\theta) = 0$, where

$$F_{\beta,\sigma}(\theta) = \mathop{\mathbb{E}}_{\substack{y \sim \eta \\ g \sim \mathcal{N}(0,1)}} \left[ \frac{y^2}{\theta - \sigma(\frac{m_1}{m_2}\beta y + g)} \right] - \frac{m_2}{\beta}.$$

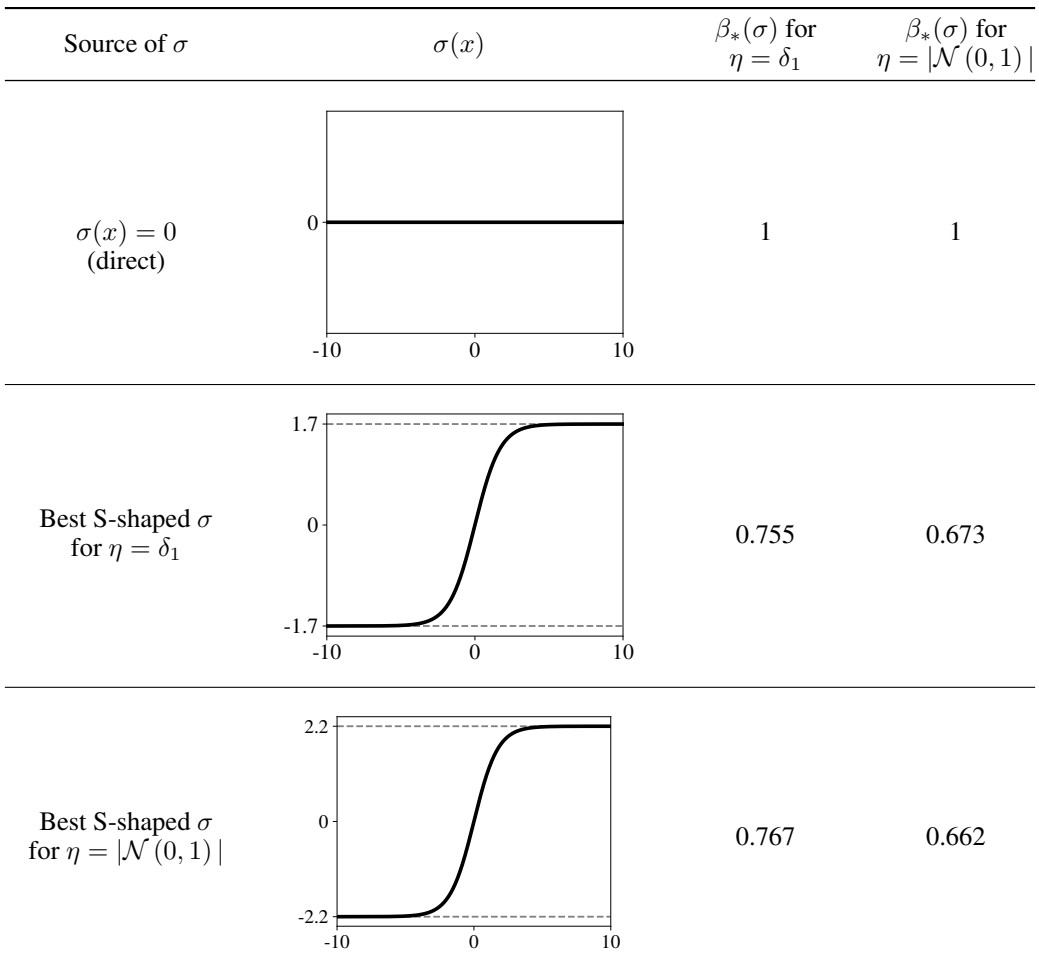

| Source of $\sigma$ | $\sigma(x)$ | $\beta_*(\sigma)$ for $\eta = \delta_1$ | $\beta_*(\sigma)$ for $\eta = \lvert \mathcal{N}(0,1) \rvert$ |
|---|---|---|---|
| $\sigma(x) = 0$ (direct) | | 1 | 1 |
| Best S-shaped $\sigma$ for $\eta = \delta_1$ | | 0.755 | 0.673 |
| Best S-shaped $\sigma$ for $\eta = \lvert \mathcal{N}(0,1) \rvert$ | | 0.767 | 0.662 |

Figure 6: We present the optimal "S-shaped" nonlinearities $\sigma(x)$ found for two signal entry distributions $\eta$, and give the threshold signal strength $\beta_*(\sigma)$ achieved by each nonlinearity for each entry distribution. At the top we also include the zero nonlinearity, leading to a direct spectral algorithm, for reference.

In principle one could differentiate this function (differentiating under the integral in the first term) and use more sophisticated methods like Newton's method for root-finding, but for the sake of simplicity we use the implementation of Brent's method in the `SciPy` library. In any case, the computational bottleneck of this procedure is the repeated evaluation of the integral implicit in expectations of the above kind (which would remain with different integrands upon taking derivatives). We have not attempted to optimize this numerical integration for the specific integrals we encounter, but this would likely be a useful step for a more detailed numerical study of the thresholds $\beta_*(\sigma)$.

### E.3 Neural network training procedure

For the method of optimizing $\sigma$ using an MLP to learn from data, the neural network was implemented with the `PyTorch` framework and trained on a randomly generated dataset of size $n = 5000$, split into training, validation, and test sets in a ratio of $0.6 : 0.1 : 0.3$. Training was performed by minimizing the binary cross-entropy loss using the AdamW optimizer with an initial learning rate of $0.01$ and weight decay of $0.01$. The learning rate was automatically halved if the validation loss did not improve for 10 consecutive epochs. Training was conducted for 350 epochs on a single NVIDIA A5000 GPU. Each such run takes around 10 minutes.

# F  Additional discussion

## F.1  Applicability to real-world data

We remark that nonlinear Laplacians appear to be a reasonable method for searching for dense subgraphs of a given graph, even when the statistical models we have studied do not hold or are not reasonable. Indeed, when applied to graph data, nonlinear Laplacians may be viewed as merely a deformed version of standard spectral approaches to finding dense subgraphs.

We have tested nonlinear Laplacians on the widely studied dataset of the neural network of C. Elegans, a network on 297 vertices proposed as an example of a "small-world" network by [WS98]. To obtain a binary and symmetric adjacency matrix, we have ignored all ancillary data including directedness of interactions in this dataset, and on the resulting dataset we consider the task of finding unusually densely connected subgraphs. We reliably find that a suitable nonlinear Laplacian algorithm finds slightly denser and substantially different subgraphs compared to a naive algorithm (here, to view an algorithm as outputting a subgraph, we compute the top eigenvector of the associated matrix and then take the subset of vertices associated to some number of the eigenvector's largest coordinates). For instance, searching for a dense subgraph of 15 vertices, a naive spectral algorithm finds a subgraph having 65% of all possible edges present, while a nonlinear Laplacian algorithm (for instance, that obtained by optimizing in the "S-shaped" class of nonlinearities based on the $\tanh$ function) finds one having 73% of all possible edges.

## F.2  Transferability of nonlinearities

One advantage of nonlinear Laplacian algorithms is that it is sensible to use the same algorithm—the same choice of $\sigma$—for different values of the signal entry distribution $\eta$. (Indeed, while we have not explored this rigorously here, it also seems like a given $\sigma$ should be a reasonable choice for either sparse or dense signals, the latter as discussed in Appendix D.3.) We give a simple illustration in Figure 6. For the two choices $\eta = \delta_1$ (corresponding to a planted submatrix model) and $\eta = |\mathcal{N}(0,1)|$. For each, we use black-box optimization to optimize $\sigma$ within the simple class of functions $\sigma(x) = a\tanh(bx)$, which we call "S-shaped" choices of $\sigma$. We find that using either $\sigma$ for either choice of $\eta$ is effective, in both cases much more effective than a direct spectral algorithm, and that "fine-tuning" $\sigma$ to the model gives only a modest further improvement. Further, as is also visible in the plots of these $\sigma$, the optimized values of $b$ are quite close (roughly $b \approx 0.584$ for $\eta = \delta_1$ and $b \approx 0.575$ for $\eta = |\mathcal{N}(0,1)|$), so the resulting nonlinearities are close to but not quite rescalings of one another.

## F.3  Generalized directional prior information

Both in the models we have singled out in Definition 1.1 and in our formulation of nonlinear Laplacian spectral algorithms, the direction of the $\mathbf{1}$ vector has played a distinguished role. This is important to the reasoning behind our algorithms, because in order for adding $\boldsymbol{D}$ to $\widehat{\boldsymbol{Y}}$ to improve the performance of a spectral algorithm, we must have that $\boldsymbol{x}^\top \boldsymbol{D} \boldsymbol{x}$ is large. For monotone $\sigma$ the diagonal entries of $\boldsymbol{D}$ will be positively correlated with the entries of $\boldsymbol{x}$, so $\boldsymbol{x}$ must be biased entrywise to be positive, as we have assumed.

On the other hand, if one is instead given prior information that $\boldsymbol{x}$ is correlated with some other vector $\boldsymbol{v} \neq \mathbf{1}$, of course one may just preprocess $\widehat{\boldsymbol{Y}}$ into $\boldsymbol{Q}^\top \widehat{\boldsymbol{Y}} \boldsymbol{Q}$ for some orthogonal transformation $\boldsymbol{Q}$ that maps $\mathbf{1}$ to $\boldsymbol{v}$ (for instance the reflection that swaps the $\mathbf{1}$ and $\boldsymbol{v}$ directions) and then apply a nonlinear Laplacian algorithm to this matrix. Similarly, if one is given the information that $\boldsymbol{x}$ lies in a cone $\mathcal{C} \subset \mathbb{R}^n$, one may try to either find a vector $\boldsymbol{v}$ that lies near the "center" of that cone so that $\boldsymbol{x}$ is correlated with $\boldsymbol{v}$ and then apply the above strategy, or to directly look for $\boldsymbol{Q}$ as above that maps the positive orthant $\{\boldsymbol{x} \in \mathbb{R}^n : \boldsymbol{x} \geq \mathbf{0}\}$ to a cone covering $\mathcal{C}$. If $\mathcal{C}$ is not convex (say being the union of a small number of convex cones), one may also attempt to find or learn from data several different directions to play the role of $\mathbf{1}$, as we discuss further below. We have not explored these ideas systematically, but they seem to be promising directions for future investigation.

### F.4 Generalized architectures and relation to graph neural networks

The original motivation that led us to study $\sigma$-Laplacians was a considerably more general question of how to learn effective spectral algorithms. That is, can one learn a function $M : \mathbb{R}^{n \times n}_{\mathrm{sym}} \to \mathbb{R}^{n \times n}_{\mathrm{sym}}$ such that $\lambda_{\max}(M(\widehat{\boldsymbol{Y}}))$ is a good test statistic for estimation or $\boldsymbol{v}_1(M(\widehat{\boldsymbol{Y}}))$ is a good estimator for recovery in PCA problems? Two structural criteria seem reasonable for such $M$:

1. $M$ should be an equivariant matrix-valued function with respect to the two-sided action of symmetric group on symmetric matrices, so that for any permutation matrix $\boldsymbol{P}$ we have $M(\boldsymbol{P}\widehat{\boldsymbol{Y}}\boldsymbol{P}^\top) = \boldsymbol{P}M(\widehat{\boldsymbol{Y}})\boldsymbol{P}^\top$.

2. $M$ should allow for "dimension generalization," so that we may learn $M$ from datasets of small $n_0 \times n_0$ matrices but apply it to $n \times n$ matrices with $n \gg n_0$.

The work of [MBHSL18] provides building blocks for constructing such $M$ as a neural network. Namely, they classify the equivariant *linear* functions $\mathbb{R}^{n \times n}_{\mathrm{sym}} \to \mathbb{R}^{n \times n}_{\mathrm{sym}}$, which also are given in a basis that is consistent across dimensions, so that a function on one dimensionality of matrices naturally extends to higher dimensions. (See also the work of [LD24] for a general framework for dimension generalization in such settings.) They frame their constructions as giving natural graph neural networks acting on $\boldsymbol{Y}$ an adjacency matrix, but really the key structure involved is equivariance (which in graphs becomes the particularly natural invariance under relabelling of vertices of a graph). Using this, the following is a natural class of functions for our purposes: given $L_1, \ldots, L_k : \mathbb{R}^{n \times n}_{\mathrm{sym}} \to \mathbb{R}^{n \times n}_{\mathrm{sym}}$ such equivariant linear functions and $\sigma_1, \ldots, \sigma_k : \mathbb{R} \to \mathbb{R}$ nonlinearities that extend to apply entrywise to matrices, we may compute

$$M(\widehat{\boldsymbol{Y}}) := \sigma_k(L_k(\cdots \sigma_1(L_1(\widehat{\boldsymbol{Y}})))).$$

When can such a construction be analyzed by tools of random matrix theory of the kind we have used? Unfortunately, it seems that if we want to have that $\|\widehat{\boldsymbol{Y}}\| = O(1)$ (say having a compactly supported continuous bulk of eigenvalues with a finite number of eigenvalues of constant order) implies $\|M(\widehat{\boldsymbol{Y}})\| = O(1)$ (with a similar structure), we must restrict this natural general structure considerably. One issue is that some basis elements of equivariant linear matrix functions roughly preserve the rank of a matrix, for instance the functions $\boldsymbol{Z} \mapsto \boldsymbol{Z}$ or $\boldsymbol{Z} \mapsto \mathrm{diag}(\boldsymbol{Z}\boldsymbol{1})$. But, other functions map high-rank matrices to low-rank ones, such as $\boldsymbol{Z} \mapsto \boldsymbol{1}(\boldsymbol{Z}\boldsymbol{1})^\top + (\boldsymbol{Z}\boldsymbol{1})\boldsymbol{1}^\top$. Thus if we are not very careful to center and rescale the inputs into each $L_i$, we can easily end up with outlier eigenvalues of order $\omega(1)$. This is not obviously an issue algorithmically or computationally (though it seems plausible it might lead to instability in training), but it does lead to random matrices different from the "signal plus noise" structure where both summands are of comparable order.

Similarly, if the $\sigma_i$ are not carefully centered, then the output of a given $\sigma_i$ can have, say, a positive entrywise bias. This corresponds, roughly speaking, to an eigenvector correlated with the $\boldsymbol{1}$ direction with very large eigenvalue relative to the remaining eigenvalues (similar to the Perron-Frobenius eigenvalue of a matrix with positive entries). If we further take such a matrix and apply the above function $\boldsymbol{Z} \mapsto \boldsymbol{1}(\boldsymbol{Z}\boldsymbol{1})^\top + (\boldsymbol{Z}\boldsymbol{1})\boldsymbol{1}^\top$, we will further inflate this eigenvalue, and iterating this procedure can allow the largest eigenvalue of $M(\widehat{\boldsymbol{Y}})$ to quickly diverge with the dimension $n$ and the number of layers $k$.

Our choice of nonlinear Laplacians is in part made to avoid all of these issues by restricting the nonlinear part of our (very simple relative to the above) architecture to yield a *diagonal* matrix. Thus, so long as our $\sigma$ is bounded, the above situation of "blowing up" eigenvalues cannot occur.

It is perhaps natural to try to extend this construction to more complex diagonal matrix functions of $\widehat{\boldsymbol{Y}}$. For instance, one may parametrize an equivariant function $\boldsymbol{d} : \mathbb{R}^{n \times n}_{\mathrm{sym}} \to \mathbb{R}^n$ (that is, one having $\boldsymbol{d}(\boldsymbol{P}\widehat{\boldsymbol{Y}}\boldsymbol{P}^\top) = \boldsymbol{P}\boldsymbol{d}(\widehat{\boldsymbol{Y}})$) in a way similar to the approach of [MBHSL18] to equivariant linear functions together with entrywise nonlinearities and use $M(\widehat{\boldsymbol{Y}}) = \widehat{\boldsymbol{Y}} + \mathrm{diag}(\boldsymbol{d}(\widehat{\boldsymbol{Y}}))$. Provided that the specification of $\boldsymbol{d}$ ends by applying a bounded nonlinearity, we again obtain a reasonable random matrix model for theoretical analysis. Many similar variants are natural; for instance, closer to our choice here, one could also fix or learn several vectors $\boldsymbol{v}_1, \ldots, \boldsymbol{v}_k$ and compute $\boldsymbol{d}$ an equivariant function of the tuple of vectors $(\widehat{\boldsymbol{Y}}\boldsymbol{v}_1, \ldots, \widehat{\boldsymbol{Y}}\boldsymbol{v}_k)$, allowing the "distinguished direction" $\boldsymbol{1}$ in nonlinear Laplacians to instead be learned from data (see the discussion in the previous section

for a setting where this could be useful). We believe this class of examples is an intriguing direction for future work on data-driven design of spectral algorithms.

