# OpenReview forum: "Nonlinear Laplacians: Tunable principal component analysis under directional prior information"
_NeurIPS.cc/2025/Conference — NeurIPS 2025 spotlight_

### Official Review · Reviewer_Jsip · 2025-07-03

**Clarity:** 3
**Significance:** 3
**Originality:** 3
**Rating:** 5
**Confidence:** 3

**Summary:**

This paper introduces the Nonlinear Laplacian, a method within the PCA family of algorithms, for detecting and recovering low-rank signals from noisy data. The technique is specifically designed for scenarios where the signal is known to be highly correlated with the vector of all ones, leveraging this bias as prior information. The authors demonstrate that by applying and tuning the Nonlinear Laplacian, the largest eigenvalue corresponding to the low-rank signal is isolated as a distinct outlier from the other eigenvalues, enabling its detection.

**Questions:**

A experiment with a real-world dataset would strengthen the practical aspect of the work.

Could you provide an experiment to test Conjecture 3.7?

Regarding the generalized directional priors mentioned as future work in Appendix F.2: How would the method's performance be affected when using a more general prior instead of the all-ones vector? Specifically, could you provide a high-level intuition on how changing the direction of the prior would impact the phase transition threshold, $\beta$? Is this direction handled in the previous works introduced in your paper?

**Ethical Concerns:**

["NO or VERY MINOR ethics concerns only"]

**Final Justification:**

An interesting method, the Nonlinear Laplacian, for detecting and recovering low-rank signals from noisy data. The method is well-described and it has a solid theoretical analysis.

**Limitations:**

Yes.

**Paper Formatting Concerns:**

No.

**Quality:**

3

**Strengths And Weaknesses:**

Strengths:
- The paper is well-structured, and the problem is introduced clearly.
- The simple experiment and the analytical results support the claim.

Weaknesses:
- Some of the paper's conjectures could be tested with an experiment (see question #2).
- The comparison with prior work for the $\beta>0$ case is not clear. As noted in the paper's limitations, the AMP algorithm can outperform their method on the chosen evaluation data. It is also unclear why the authors claim AMP is "more complex," as no analytical or wall-clock time comparison between the two methods is provided.
- Lack of experiments to show the applicability of their method.

---

> ### Author Rebuttal · Authors · 2025-07-31
>
> We thank the reviewer for their careful reading of the manuscript.
>
> Since several reviewers made similar comments but it is not possible to include a global reply, we have repeated some responses nearly verbatim. For all citations mention in our replies, if the citation was included in the manuscript then we have referred to it with the same abbreviation as in the bibliography. If not, we have included a citation with an abbreviation in our reply. Below, we address each of the concerns and comments posed.
>
> > The comparison with prior work for the $\beta>0$ case is not clear. As noted in the paper's limitations, the AMP algorithm can outperform their method on the chosen evaluation data. It is also unclear why the authors claim AMP is "more complex," as no analytical or wall-clock time comparison between the two methods is provided.
>
> First, as a point of clarification, after looking more closely into these papers, we have realized that the algorithms of [DM15, HWX17] for the Gaussian Planted Submatrix are not AMP in the usual sense of a "nonlinear power method." Rather, they are "full" message passing similar to belief propagation (BP), where the state of the algorithm is an $n \times n$ matrix of messages between all pairs of indices (though this does not need to be stored explicitly). We will edit our discussion accordingly. For the dense signals of [DMR14, MR15], the algorithms proposed are AMP in the usual sense. However, our analysis does not apply to those models (though we believe our algorithms should still work well), so the former is a better comparison. In both cases, these iterative algorithms do not require a spectral initialization, and we will revise our remarks about this.
>
> When we refer to our algorithms as "simpler" than AMP or BP, we refer to their conceptual simplicity rather than their runtime (though we will comment on computation below). In a nonlinear Laplacian algorithm, examining the nonlinearity $\sigma$ (Figures 2 and 3) makes it clear how the algorithm relatively weighs spectral and degree information. The algorithm's success corresponds to the simple phenomenon of outlier eigenvalues, which is again easy to visualize and understand (Figure 1). In contrast, AMP requires carefully designed iterations, including the non-backtracking structure of BP or the corresponding "Onsager correction" in AMP, whose effect is subtle yet which are important to the state evolution analysis of these algorithms going through. The role of this structure in the effectiveness of AMP or BP is, in our opinion, considerably trickier to understand.
>
> The relative simplicity of nonlinear Laplacians also leads to more concrete advantages. For example, it is easy to argue that small adversarial corruptions do not affect a nonlinear Laplacian very much, by standard spectral perturbation bounds. Similar robustness does not hold for AMP, and to guarantee it requires modifying the algorithm. Both of these claims were, for example, studied recently by [IS24, IS25].
>
> Also, spectral algorithms may be carried out by any method that approximates the top eigenvector of a matrix, a task of broad interest to which many approaches are known. For instance, [BBCIK25] is one recent work that uses sketching to approximate this computation for matrices that are too large to fit in memory. To our knowledge, such large-scale execution of AMP or BP has not yet been achieved.
>
> We are happy to soften our claims that AMP is broadly "more complex" in a revision, and instead to focus on these more specific advantages.
>
> [IS24] Ivkov, M. and Schramm, T. 2024. Semidefinite Programs Simulate Approximate Message Passing Robustly. In STOC 2024 (pp. 348–357).
>
> [IS25] Ivkov, M. and Schramm, T. 2025. Fast, Robust Approximate Message Passing. In STOC 2025 (pp. 2237–2248).
>
> [BBCIK25] Boahen, E., Brugiapaglia, S., Chou, H.H., Iwen, M., Krahmer, F. 2025. Fast One-Pass Sparse Approximation of the Top Eigenvectors of Huge Low-Rank Matrices? Yes, $MAM^{∗}$! arXiv:2507.17036.
>
> > A experiment with a real-world dataset would strengthen the practical aspect of the work.
>
> While a detailed study on realistic network datasets would require more time, we have done some initial experiments on the widely studied dataset of the neural network of C. Elegans, a network on 297 vertices proposed as an example of a "small-world" network by [WS98]. To obtain a binary and symmetric adjacency matrix, we have ignored all ancillary data including directedness of interactions in this dataset, and on the resulting dataset we consider the task of finding unusually densely connected subgraphs. We reliably find that a suitable nonlinear Laplacian algorithm finds slightly denser and substantially different subgraphs compared to a naive algorithm (here, to view an algorithm as outputting a subgraph, we compute the top eigenvector of the associated matrix and then take the subset of nodes associated to some number of the eigenvector's largest coordinates). For instance, searching for a dense subgraph of 15 vertices, a naive spectral algorithm finds a subgraph having 65\% of all possible edges present, while a nonlinear Laplacian algorithm (for instance, that obtained by optimizing in the "S-shaped" class using the $\tanh(\cdot)$ function) finds one having 73\% of all possible edges.
>
> [WS98] Watts, D., Strogatz, S. Collective dynamics of ‘small-world’ networks. Nature 393, 440–442 (1998).
>
> > Could you provide an experiment to test Conjecture 3.7?
>
> We performed the same experiment as in Figure 4, using the same nonlinearity $\sigma$ chosen for the Gaussian Planted Submatrix model. However, in this case, we computed the top eigenvalue and the eigenvector overlap over random inputs drawn from the Planted Clique distribution instead. The final plot is almost identical to Figure 4, showing a phase transition at $\beta_*\approx 0.76$. Thus, the experimental results support the Conjecture.
>
> > Regarding the generalized directional priors mentioned as future work in Appendix F.2: How would the method's performance be affected when using a more general prior instead of the all-ones vector? Specifically, could you provide a high-level intuition on how changing the direction of the prior would impact the phase transition threshold, $\beta$? Is this direction handled in the previous works introduced in your paper?
>
> To clarify, our characterization of the phase transition threshold $\beta_*$ in Theorem 3.3 depends on the Sparse Biased PCA model assumption in Definition 1.1, which specifies the prior distribution of the signal vector $x$. Our argument in Appendix F.2 considers the case where the model assumption is exactly an "orthogonally transformed version" of the Sparse Biased PCA model, i.e. the prior for the spiked signal $x$ is instead $x = \frac{Qy}{\\|y\\|}$, where $Q$ is an orthogonal matrix, and $y \in \mathbb{R}^n$ is defined as in Definition 1.1. Indeed, in this case, vector $x$ is correlated with $v=Q 1$ instead of $1$. Our proposed strategy involves first preprocessing $\hat{Y}$ into $Q^{\top} \hat{Y} Q$ (effectively "undoing" the transformation), and this leads to exactly the same performance. If the prior for the signal $x$ does not follow such a distribution, then our theory does not offer guarantees about the method's performance. Nevertheless, we speculate that the core idea would still apply, although additional studies are required to analyze the corresponding phase transition threshold $\beta_*$.

---

> > ### Comment · Reviewer_Jsip · 2025-08-05
> >
> > Thank you for the clarifications. My positive assessment of the paper remains.

---

### Official Review · Reviewer_o8QY · 2025-07-03

**Clarity:** 4
**Significance:** 3
**Originality:** 3
**Rating:** 5
**Confidence:** 4

**Summary:**

The paper studies the problem of rank-one spiked model when a prior directional information of the signal $x$ is known. The authors propose a family of spectral methods using the nonlinear Laplacian with some nonlinear mapping $\sigma$. The authors derive exact phase transitions and detection thresholds in this scenario, and shows a lower detection threshold compared to the vanilla BBP phase transition.

**Questions:**

1.	It would be very interesting to study the sharp detection threshold optimized over all possible transformation function $\sigma$. Does the authors have an intuition what the optimal function $\sigma$ could be?

**Ethical Concerns:**

["NO or VERY MINOR ethics concerns only"]

**Final Justification:**

I remain my recommendation for acceptance. The paper is techinically solid and theoretically interesting. The paper leaves a few interesting future directions, and I would like to see the work published.

**Limitations:**

See "Strengths and Weakness".

**Paper Formatting Concerns:**

No concerns noted.

**Quality:**

3

**Strengths And Weaknesses:**

Strengths:

1.	The paper is well-written, clearly organized and easy to follow. The results using nonlinear Laplacians are interesting to me.
2.	The theorems are well presented and are sound for the parts that I checked.

Weakness:

1.	The optimal detection threshold is not known and only numerically computed for family of nonlinear functions $\sigma$.

---

> ### Author Rebuttal · Authors · 2025-07-31
>
> We thank the reviewer for their careful reading of the manuscript. Below, we address the question posed:
>
> > It would be very interesting to study the sharp detection threshold optimized over all possible transformation function $\sigma$. Does the authors have an intuition what the optimal function $\sigma$ could be?
>
> Unfortunately, we were unable to identify a tractable method to analytically determine the optimal $\sigma$, and we therefore had to rely on numerical methods. The objective function $\beta_*(\sigma)$ is given by a fixed point equation in terms of a function $\theta_\sigma$, which itself is defined in terms of another "inner" fixed point equation. We do not know how to solve either "layer" of these equations in closed form, even for simple examples.
>
> On the other hand, our numerical results provide hints about the potential form of the optimal $\sigma$. In Figure 2, we presented the $\sigma$ functions obtained through four different numerical optimization methods for the Gaussian Planted Submatrix problem. Notably, both (d) and (e), which optimize over a large degree of freedom, have similar shapes. Additionally, the positive part of (c) has a comparable shape, while the negative part appears to have minimal impact on the value of $\beta_*$. This observation suggests that, in the Gaussian Planted Submatrix model, the optimal function $\sigma$ may resemble this general shape, modulo robustness to various factors. Such robustness is demonstrated in Figure 3 and further discussed in lines 1176–1179, where we show that certain characteristics of the shape of $\sigma$ do not significantly affect $\beta_*$. The robustness to the "height" of $\sigma$ also helps explain the disparity in heights between Figures 2(d) and 2(e).

---

> > ### Comment · Reviewer_o8QY · 2025-08-03
> >
> > Thank you for your responses. I remain thinking the paper is interesting and the question I proposed were largely curiousity of the next step/future directions. It would be also interesting to study optimality under a certain class of "first order spectral algorithm", as is brought up in other reviewers' discussion regarding AMP.

---

### Official Review · Reviewer_BvVy · 2025-07-03

**Clarity:** 4
**Significance:** 3
**Originality:** 3
**Rating:** 5
**Confidence:** 5

**Summary:**

This paper studies a model called "Sparse Biased PCA". Here, one observes $Y = \beta \sqrt{n} xx^\top  + W$ where $x$ is an unknown signal vector and $W$ is i.i.d. Gaussian noise. The author consider a sparse prior for the signal $x$ where approximately $p_n$ fraction of the entries are non-zero where  $\log^6(n)/n \ll p_n \ll 1$. Moreover, the mean of the non-zero coordinates is positive. The authors focus primarily on the detection problem, where the goal is to test if $\beta = 0$ or if $\beta$ takes a given non-zero value.

Intuitively, one expects that the largest eigenvector of $Y$ can help us detect the signal. Since the signal prior has positive mean, the vector $Y 1$ might also be helpful. The authors propose to combine these two sources of information using the non-linear Laplacian matrix $L = Y + \mathrm{diag}(\sigma(Y \cdot 1))$ where $\sigma: \mathbb{R} \rightarrow \mathbb{R}$ is a user-specified function.

The primary contributions of the authors are as follows:

1) The authors characterize asymptotic spectrum of the non-linear Laplacian matrix (as $n$, the dimension tends to infinity). This includes a characterization of the bulk spectrum and the outlier eigenvalues (if any). This provides a consistent test for the problem, provided that $\beta$ is above a critical threshold.

2) A special case of their model is the Gaussian planted submatrix problem. The authors show that their proposed test succeeds when $\beta > 0.79$. On the other hand, any test based on the eigenvalues of $Y$ alone would require $\beta > 1$, and any test that depends only on $Y1$ will always fail. Hence, the non-linear Laplacian, which combines both these sources of information, improves on both of them.

3) The authors also explore some numerical approaches to find a good $\sigma$ function.

**Questions:**

Here are some suggestions and questions:

1) There has been some work in combining linear and spectral estimators for generalized linear models, which seems related. It would be good to cite and discuss the relationship. See: M. Mondelli, C. Thrampoulidis, R. Venkataramanan, "Optimal Combination of Linear and Spectral Estimators for Generalized Linear Models"
There are, of course, important differences: the work by Mondelli et. al. combines a spectral estimator and a linear estimator, whereas this paper incorporates a "linear estimator" into the matrix and computes a spectral estimator on the resulting matrix.

2) It would be good to discuss the bottleneck in proving Theorem 3.3 for the Laplacian matrix rather than the "compressed Laplacian matrix". If there are no significant bottlenecks, it would be good to tie up this loose end.

3) Regarding the optimal $\sigma$: the optimization problem over $\sigma$ has some similarities to problems in prior work. See: Optimal Spectral Initialization for Signal Recovery with Applications to Phase Retrieval by Wangyu Luo, Wael Alghamdi, Yue M. Lu.
Is there a possibility of adapting their arguments to find the optimal $\sigma$ mathematically?

4) Regarding Eigenvector Overlap: I might be mistaken but a simple argument on top of the results of the authors should also give a characterization of the eigenvector overlap. Let $\lambda_{n}(\beta)$ denote the largest eigenvalue of $Y$, viewed as a function of $\beta$. Using the standard argument to differentiate eigenvalues (see Section 4 in https://terrytao.wordpress.com/2010/01/12/254a-notes-3a-eigenvalues-and-sums-of-hermitian-matrices/), one can show that the derivative of $\lambda_n(\beta)$ with respect to $\beta$, will be the squared overlap between the signal and the eigenvector. One can swap the limit and derivative because the largest eigenvalue is a convex function of beta, and conclude that the overlap should converge to the derivative of the limit of the largest eigenvalue.
Another option could be to try the argument given in Proposition 2 of https://arxiv.org/pdf/1702.06435.

5) Is there an absolute value missing on Pg 23, first equation on the page (RHS)?

6) Section C: If I am not misunderstanding, I think some of the arguments here can be simplified. The basic task seems to be to show that the root of $G_n(\lambda) = 1$ converges to the root of $G(\lambda) = 1$. Because $G_n, G$ are strictly decreasing functions, one doesn't need uniform convergence for this. If the function $G = 1$ has a root $\theta$, then for any for any $\epsilon>0$, $G(\theta - \epsilon)$ will be positive and $G(\theta+\epsilon)$ will be negative. For sufficiently large $n$, these signs will be reflected in $G_n$. This means that $G_n$ must cross zero at least once in $(\theta-\epsilon, \theta+\epsilon)$, and this root must be unique because $G_n$ is strictly decreasing.

7) I also think it would be helpful to motivate the model studied in the paper in a better way. Currently, the reader is left wondering where did this specific model came from and why one should study it?

**Ethical Concerns:**

["NO or VERY MINOR ethics concerns only"]

**Final Justification:**

I think this is a good paper that meets the bar of acceptance. I found the overall message of the paper quite appealing: there are regimes where neither the leading eigenvector of $Y$ nor the vector $Y1$ by itself can solve the detection problem, but when combined together in a non-linear Laplacian matrix, they can. This seems quite original.

During the rebuttal, the authors resolved many of my concerns:
-- Added a few missing references.
-- The authors also have a characterization of the eigenvector overlap.

While the paper still has a few loose ends in my opinion, the authors have explained the technical challenges in overcoming them.

**Limitations:**

yes

**Quality:**

3

**Strengths And Weaknesses:**

Strengths:

1) The paper is very well-written and clear. The authors have made a commendable effort to include statements of random matrix theory results and other mathematical preliminaries to make the paper self-contained.

2) I found the overall message of the paper quite appealing: there are regimes where neither the leading eigenvector of $Y$ nor the vector $Y1$ by themself can solve the detection problem, but when combined together in a non-linear Laplacian matrix, they can. This seems quite original.

Weaknesses:

1) The paper seems to have a few "loose ends". I might be mistaken, but it is quite plausible that some of these "loose ends" can be resolved using existing techniques:
-- The authors prove their result for a "theoretically convenient" modification of the Laplacian matrix rather than the Laplacian matrix.
-- A characterization of the eigenvector overlap is missing.

---

> ### Author Rebuttal · Authors · 2025-07-31
>
> We thank the reviewer for their careful reading of the manuscript.
>
> Since several reviewers made similar comments but it is not possible to include a global reply, we have repeated some responses nearly verbatim. For all citations mention in our replies, if the citation was included in the manuscript then we have referred to it with the same abbreviation as in the bibliography. If not, we have included a citation with an abbreviation in our reply. Below, we address each of the concerns and comments posed.
>
> ## Question 1: combining linear and spectral estimators
>
> We will cite this paper in a revision, and are happy to include other work in this vein if the reviewer is aware of any.
>
> Our approach may indeed be viewed as optimizing over ways to combine two algorithmic approaches: if we considered only $\lambda_1(\text{diag}(Y1)) = \max_i (Y1)_i$ then we would in effect look only at (for graph problems) the highest degree of all vertices, while if we considered only $\lambda_1(Y)$ we would look only at a spectral algorithm. However, in our setting, unlike the suggested citation, it probably would not be useful to first compute these two quantities and then try to combine them (or to compute the whole spectrum $\lambda(Y)$ and the whole degree vector $Y1$ and try to combine those); as our Theorem 3.5 suggests, it seems that the information $Y1$ carries about the signal is contained not in the extremal value $\max_i (Y1)_i$ but only in weak fluctuations of certain entries of intermediate size of the vector $Y1$.
>
> ## Question 2: compressed Laplacian
>
> Applying the interlacing inequalities, we have $\lambda_{1}(L) \geq \lambda_{1}(\tilde{L}) \geq \lambda_{2}(L) \geq \lambda_{2}(\tilde{L}) \geq \dots \geq \lambda_{n-1}(\tilde{L}) \geq \lambda_{n}(L),$ where $L$ is the nonlinear Laplacian matrix, and $\tilde{L}$ is the compressed Laplacian matrix. This result indicates that when $\tilde{L}$ has one outlier eigenvalue outside its bulk, $L$ would have $\lambda_3(L), \dots, \lambda_n(L)$ within the same bulk, along with either one or two outlier eigenvalues. However, this does not clarify which signal-to-noise ratio causes $\lambda_1(L)$ to emerge as an outlier eigenvalue, nor does it explain whether $L$ could possibly have an outlier eigenvalue under the null distribution.
>
> We tried some other techniques, but we were unable to precisely characterize the limit of $\lambda_1(L)$, as this depends on how $L$ interacts with the $1$-direction that was projected away during the compression. This creates a bottleneck that we have not yet been able to overcome.
>
> ## Question 3: optimization of $\sigma$
>
> While the ideas in this paper are quite connected to ours and we will certainly mention it as related prior work, we have not found a way to directly adapt the argument there to our setting. The main difference appears to us to be that, ultimately, the measurement of the performance of a given nonlinearity ($\mathcal{T}$ in their notation) in the citation can be written in closed form in terms of some ancillary functions associated to the problem setting. The optimization of the nonlinearity can then be written as the optimization of a closed-form objective over closed-form constraints (as given explicitly in their Section III.B).
>
> If we were able to do the same, then, while it is not a given that the resulting problem could be solved analytically, at least we could proceed with standard calculus of variations tools. Unfortunately, in our case, this measurement of the performance of a nonlinearity ($\sigma$ in our notation) is seemingly much more complicated, given by a fixed point equation in terms of a function $\theta_{\sigma}$, which itself is defined in terms of another "inner" fixed point equation. We do not know how to solve either "layer" of these equations in closed form, even for simple examples, so we do not see a way forward to determine the optimal $\sigma$ by this kind of direct analysis.
>
> ## Question 4: eigenvector overlap
>
> There seem to be a few issues with the first method proposed. First, and most importantly, the following calculation suggests that it actually should not be true that the overlap of the top eigenvector with the signal will be the derivative of the top eigenvalue in this model (unlike the ordinary spiked Wigner model). Recall that the matrix we are interested in is given by $L = L(\beta) = \hat W + \beta xx^{\top} + \mathrm{diag}(\sigma(\hat{W}1 + \beta \langle x, 1 \rangle x))$. We will indeed have $\frac{d}{d\beta} \lambda_1(L(\beta)) = \langle v_1v_1^{\top}, L^{\prime}(\beta)\rangle$, where $v_1$ is the top eigenvector of $L$. However, while the first term of this inner product is $\langle x, v_1 \rangle^2$ as we wish to analyze, there is also a second term, $\sum_{i = 1}^n \sigma^{\prime}((\hat{W}1)\_{i} + \beta \langle x, 1 \rangle x_i) \langle x, 1 \rangle x_i (v\_1)\_i^2$. We claim that, at least in simple cases of our model, this term should be $\Theta(1)$, of the same order as $\langle x, v_1 \rangle^2$. Indeed, say that in our notation $\eta = \delta_1$, so that the non-zero entries of $x$ are all equal and positive and there are roughly $pn$ such entries for $p = p(n)$. Then, since $\sigma^{\prime} > 0$ is of order constant, all terms above are constant, and the sum is $\approx \sqrt{pn} \sum_{i = 1}^n x\_i (v\_1)\_i^2$. Since $v_1$ has constant correlation with $x$, this should be $\gtrsim \sqrt{pn} \sum_{i = 1}^n x\_i^3 = \sqrt{pn} \cdot pn \cdot (\sqrt{pn})^{-3} = \Theta(1)$. Of course, if we could characterize this second term then we could subtract it to again analyze $\langle x, v_1 \rangle^2$, but it seems even more complicated than that quantity to understand precisely.
>
> Second, and maybe less significantly, the reviewer's sketched argument uses that $\lambda_1(L(\beta))$ is a convex function of $\beta$. However, according to calculations similar to the above, it seems that for general nonlinearities $\sigma$ in our setup this will no longer be the case, making the swap of limit and derivative invalid as well.
>
> Regarding the second method proposed, it appears that a key difference in the setting of the mentioned paper is that the underlying signal $\xi$ to be recovered is deterministic. Moreover, due to orthogonal invariance of the distribution of $a_i$'s, $\xi$ can be assumed to lie in the direction of $e_1$ without loss of generality. This assumption seems to be critical for the method in Proposition 2, which relies on the block-partitioned form of the matrix. In our case, it is difficult to characterize the block-partition form of the resulting matrix (in terms of $\beta$ and $\sigma$) after an orthogonal transformation $QLQ^\top$, where $L$ is our nonlinear Laplacian matrix, and $Q$ is an orthogonal transformation that maps $x$ to $e_1$. Further, in our situation, $L$ has the second term depending nonlinearly on $x$, whose product with $Q$ again seems difficult to work with.
>
> ## Question 5: absolute value missing
>
> We thank the reviewer for pointing out this error, and will make this correction in a revision.
>
> ## Question 6: simplification of Section C
>
> We thank the reviewer for the suggestion. Indeed, the arguments in Appendix C.1 and C.2.2 can be significantly simplified with this approach. In particular, this allows us to avoid the technicalities of complex analysis (Rouché's theorem) and some of the concentration inequalities. We will edit and streamline these arguments in the revision.
>
> ## Question 7: motivation
>
> Our original motivation was a considerably more general question of how to learn effective spectral algorithms, i.e., how to learn a transformation $M: \mathbb{R}^{n \times n}\_{sym} \to \mathbb{R}^{n \times n}\_{sym}$ such that $\lambda_{max}(M(Y))$ serves as a good test statistic (or $v_1(M(Y))$ serves as a good estimator) in PCA-type problems. This idea is inspired by several previous works. In [HSSS16], it is shown in an abstract way related to the sum-of-squares hierarchy that spectral algorithms exist for certain testing problems, but no explicit construction of the matrices $M(Y)$ that would be required (possibly larger than $n \times n$ in that setting) is given. In [LKZ15, PWBM18], it is shown that for PCA with non-Gaussian noise, preprocessing $Y$ by applying an entrywise nonlinearity improves the performance of a spectral algorithm.
>
> These works suggest that strong spectral algorithms can be designed for many problems by choosing a suitable $M$. We hoped to explore this idea, but were constrained to study $M$ that would yield $M(Y)$ whose spectrum is tractable to analyze with random matrix theory. We discuss this in Appendix F.3, explaining how we initially considered a general class of functions for $M$, encountered challenges in analyzing or formulating sensible limits for the eigenvalues of $M(Y)$, and finally chose nonlinear Laplacians as a tractable example and first step towards implementing our general proposal.
>
> In our revision, we will provide more context regarding the goal of learning spectral algorithms and our specific reasons for working with nonlinear Laplacians.
>
> [HSSS16] Hopkins, S.B., Schramm, T., Shi, J. and Steurer, D. 2016. Fast spectral algorithms from sum-of-squares proofs: tensor decomposition and planted sparse vectors. In STOC 2016 (pp. 178-191).

---

> ### Comment · Reviewer_BvVy · 2025-08-02
>
> Thank you to the authors for their detailed responses. I continue to have a very positive opinion about the paper.
>
> Regarding the eigenvector overlap, my apologies, I missed the fact that the non-linear diagonal term in the non-linear Laplacian matrix also depends on $\beta$.
>
> I wonder if the following fix would work. Consider a slight generalization of the non-linear Laplacian matrix studied by the authors:
>
> $$L_n(t) = \hat{W} + (\beta+t) xx^{\top} + \mathrm{diag}(\sigma(\hat{W}1 + \beta \langle x, 1 \rangle x))$$
>
> and let $\lambda_{1,n}(t)$  denote the largest eigenvalue of $L(t)$ and let $v_{1,n}(t)$ denote the corresponding eigenvector. Here, $t$ is an additional parameter, and differentiating with respect to $t$ will help in the analysis of the eigenvector overlap.
>
> Notice that the non-linear Laplacian matrix corresponds to $L_n(0)$.
>
> If the authors arguments are robust enough to obtain a characterization of $\lim_{n \rightarrow \infty} \lambda_{1,n}(t)$, I think one would be able to infer the eigenvector overlap from this result (although I might again be missing something):
>
> 1) $\lambda_{1,n}(t)$ is a convex function of $t$ because the variational formula for the largest eigenvalue shows that it is the max of linear functions, which is convex.
>
> 2) The derivative of $\lambda_{1,n}(t)$ with respect to $t$ is  $\langle v_{1,n}(t), x \rangle^2$.
>
> 3) So the asymptotic eigenvector overlap for the non-linear Laplacian matrix $\langle v_{1,n}(0), x \rangle^2$ should be $\lim_{n \rightarrow \infty} \lambda_{1,n}^\prime(0)$

---

> > ### Author Response · Authors · 2025-08-03
> >
> > We sincerely thank the reviewer for the insightful suggestion. We applied the method and found that it works effectively. Additionally, we conducted a numerical experiment and observed that the computed limiting eigenvector overlap aligns closely with the average obtained from the random data. We will incorporate this proof into the next revision of the paper.
> >
> > We were also wondering if you have a reference where some version of this method appears. We were not aware of this method previously and would like to cite it in our paper to acknowledge its origin.

---

> > > ### Comment · Reviewer_BvVy · 2025-08-05
> > >
> > > I am glad the argument worked out.
> > >
> > > Regarding a reference, I am not sure what would be the most appropriate reference for this method. It seems to be frequently used in statistical physics, where one differentiates the log-normalizing constant of a Gibbs measure with respect to suitable parameters to analyze overlaps. See, for example, Lemma 1.1.1 in https://arxiv.org/abs/2204.02909. In the context of the submitted paper, the method discussed above would be the natural "zero-temperature" version of this statistical physics argument.

---

### Official Review · Reviewer_ggYq · 2025-07-09

**Clarity:** 3
**Significance:** 2
**Originality:** 3
**Rating:** 5
**Confidence:** 3

**Summary:**

This paper introduces a new familty of algorithms for detecting and estimating a rank-one matrix from a noisy observation, assuming access to prior information about the hidden signal's direction. Specifically, they consider a non-linear Laplacian constructed using a non-linear functional $\sigma$, and characterize the spectral properties of such non-linear Laplacian matrix. As an application of their main theorem, they show that the proposed algorithms outperform direct spectral methods for the Gaussian Planted Submatrix problem.

**Questions:**

1. I am wondering if this result can be generalized to non-Bernoulli prior, say, a prior distribution that has non-zero expectation?
2. I am not quite sure if the proposed algorithm is much simpler than the AMP algorithm: AMP is an iterative algorithm that involves mainly matrix-vector multiplication and entry-wise transformation. Specifically, spectral initialization is not needed for the sparse biased PCA setting that is considered in the current paper. On the other hand, the proposed algorithm requires taking the top eigenvalue of a matrix.
3. As shown in [EAKJ20], the information-theoretic detection threshold for rank-one spiked model is much smaller than the current theoretical guarantee. I think the authors should discuss this point in the paper.
4. For the motivation of their proposed algorithm, the authors gave the following explanation: "both the spectrum of Y and the vector Y 1 carry information about x". I am wondering if there is a specific reason for constructing a diagonal matrix using Y 1 and using it to assist recovery? In other words, I am curious about how the authors came up with this specific proposal.

**Ethical Concerns:**

["NO or VERY MINOR ethics concerns only"]

**Final Justification:**

The authors have addressed my concerns, hence I raise my rating.

**Limitations:**

yes

**Paper Formatting Concerns:**

No formatting concerns

**Quality:**

2

**Strengths And Weaknesses:**

This is a nice theoretical paper exploring the use of non-linear Laplacian for detecting and estimating a rank-one matrix from noisy observation. The presentation is neat and clear, and the theoretical results are pretty solid. My main concern is that the proposed algorithm does not seem to offer clear advantages over previous algorithms, and I would love to see a more detailed discussion in this regard.

---

> ### Author Rebuttal · Authors · 2025-07-31
>
> We thank the reviewer for their careful reading of the manuscript.
>
> Since several reviewers made similar comments but it is not possible to include a global reply, we have repeated some responses nearly verbatim. For all citations mention in our replies, if the citation was included in the manuscript then we have referred to it with the same abbreviation as in the bibliography. If not, we have included a citation with an abbreviation in our reply. Below, we address each of the concerns and comments posed.
>
> ## Question 1: generalization to other priors
>
> We are not entirely clear on the meaning of the question, but let us cover two possibilities.
> First, if the reviewer is asking whether we can allow for the (entries of the) vector $x$ to have more general distributions than Bernoulli, then our setup already allows for this. This is the role played by the distribution $\eta$ in the definition of Sparse Biased PCA (Definition 1.1), and indeed our assumption is precisely that $\eta$ has positive expectation as the reviewer suggests, so that the resulting $x$ is positively correlated with the all-ones direction. Our main result, Theorem 3.3, applies generally to such models, where Section 3.3 focused on Gaussian Planted Submatrix (the case of Bernoulli $x$) merely as a simple and familiar example.
>
> Alternatively, perhaps the reviewer is referring to the distribution of the background noise (entries of the symmetric matrix $W$). In the Sparse Biased PCA model, $W$ is Gaussian. The orthogonal invariance of the GOE distribution is essential for our compression argument, which, as we discussed in a response to Reviewer BvVy, we do not think can be easily avoided. As a result, generalizing the model to Bernoulli or other noise distributions or "observation channels" would probably require considerably different proof strategies, which we have not developed here. Nonetheless, we would speculate that the result may generalize to a broader class of noisy observations of low-rank signals.
>
> As for the noise having non-zero expectation, say $W_{ij} = W_{ji} \sim \nu$ with $m = \mathbb{E}_{w \sim \nu}[w] \neq 0$, it is convenient to write $W = W' + m n \hat{1}\hat{1}^\top$, where $W'$ is a matrix of zero-mean entries, and $\hat{1}$ is the unit vector in the all-ones direction. This reduces the model to $Y = \beta \sqrt{n} xx^{\top} + mn 11^{\top} + W',$ which can be interpreted as a low-rank (namely, rank 2) perturbation of $W'$. Notably, the non-zero expectation contributes a spiked signal in the direction of the all-ones vector, which will create another and much larger outlier eigenvalue than the signal $xx^{\top}$ creates as $n \to \infty$.
>
> ## Question 2: comparison with AMP
>
> First, as a point of clarification, after looking more closely into these papers, we have realized that the algorithms of [DM15, HWX17] for the Gaussian Planted Submatrix are not AMP in the usual sense of a "nonlinear power method." Rather, they are "full" message passing similar to belief propagation (BP), where the state of the algorithm is an $n \times n$ matrix of messages between all pairs of indices (though this does not need to be stored explicitly). We will edit our discussion accordingly. For the dense signals of [DMR14, MR15], the algorithms proposed are AMP in the usual sense. However, our analysis does not apply to those models (though we believe our algorithms should still work well), so the former is a better comparison. In both cases, these iterative algorithms do not require a spectral initialization, and we will revise our remarks about this.
>
> When we refer to our algorithms as "simpler" than AMP or BP, we refer to their conceptual simplicity rather than their runtime (though we will comment on computation below). In a nonlinear Laplacian algorithm, examining the nonlinearity $\sigma$ (Figures 2 and 3) makes it clear how the algorithm relatively weighs spectral and degree information. The algorithm's success corresponds to the simple phenomenon of outlier eigenvalues, which is again easy to visualize and understand (Figure 1). In contrast, AMP requires carefully designed iterations, including the non-backtracking structure of BP or the corresponding "Onsager correction" in AMP, whose effect is subtle yet which are important to the state evolution analysis of these algorithms going through. The role of this structure in the effectiveness of AMP or BP is, in our opinion, considerably trickier to understand.
>
> The relative simplicity of nonlinear Laplacians also leads to more concrete advantages. For example, it is easy to argue that small adversarial corruptions do not affect a nonlinear Laplacian very much, by standard spectral perturbation bounds. Similar robustness does not hold for AMP, and to guarantee it requires modifying the algorithm. Both of these claims were, for example, studied recently by [IS24, IS25].
>
> Also, spectral algorithms may be carried out by any method that approximates the top eigenvector of a matrix, a task of broad interest to which many approaches are known. For instance, [BBCIK25] is one recent work that uses sketching to approximate this computation for matrices that are too large to fit in memory. To our knowledge, such large-scale execution of AMP or BP has not yet been achieved.
>
> We are happy to soften our claims that AMP is broadly "more complex" in a revision, and instead to focus on these more specific advantages.
>
> [IS24] Ivkov, M. and Schramm, T. 2024. Semidefinite Programs Simulate Approximate Message Passing Robustly. In STOC 2024 (pp. 348–357).
>
> [IS25] Ivkov, M. and Schramm, T. 2025. Fast, Robust Approximate Message Passing. In STOC 2025 (pp. 2237–2248).
>
> [BBCIK25] Boahen, E., Brugiapaglia, S., Chou, H.H., Iwen, M., Krahmer, F. 2025. Fast One-Pass Sparse Approximation of the Top Eigenvectors of Huge Low-Rank Matrices? Yes, $MAM^{∗}$! arXiv:2507.17036.
>
> ## Question 3: information-theoretic detection threshold
>
> We are happy to mention this paper as well as the long related line of work around determining the information-theoretic thresholds of spiked matrix models. In general, as the reviewer is perhaps suggesting, for sparse distributions of the spike such models can have statistical-to-computational gaps, where the information-theoretic threshold for detection (or non-trivial estimation) is much lower than what is conjectured to be the computational threshold.
>
> However, let us emphasize that our setting is different in one important way from that of this paper and similar ones: we assume that the number of non-zero entries of $x$, given by $pn$ for the function $p = p(n)$, is decreasing as a fraction $n$, i.e., $p(n) = o(1)$.
> In this case of vanishingly sparse signals, the difference between statistical and computational thresholds has a different character, where there is a "smoother" tradeoff between the runtime of an algorithm and the strength of signal it can detect (e.g., [DKWB24]).
> For instance, in the problem of detecting a planted clique of size $\beta \sqrt{n}$ in a random graph on $n$ vertices, there exist polynomial-time algorithms that can succeed for any $\beta$, but these are believed to require a longer polynomial runtime as $\beta$ decreases. Indeed, this kind of question is one of the motivations of our work: while enhancing spectral algorithms with extra techniques like applying them to many submatrices is known to achieve such tradeoffs (mentioned in [AKS98] for the case of planted clique), we were wondering how well algorithms can do that are more "purely spectral" and instead just make a small modification to the input matrix.
>
> [DKWB24] Yunzi Ding, Dmitriy Kunisky, Alexander S. Wein, and Afonso S. Bandeira. 2023. Subexponential-Time Algorithms for Sparse PCA. Found. Comput. Math. 24, 3 (Jun 2024), 865–914.
>
> ## Question 4: motivation
>
> Our original motivation was a considerably more general question of how to learn effective spectral algorithms, i.e., how to learn a transformation $M: \mathbb{R}\_{sym}^{n \times n}\to \mathbb{R}\_{sym}^{n \times n}$ such that $\lambda_{max}(M(Y))$ serves as a good test statistic (or $v_1(M(Y))$ serves as a good estimator) in PCA-type problems. This idea is inspired by several previous works. In [HSSS16], it is shown in an abstract way related to the sum-of-squares hierarchy that spectral algorithms exist for certain testing problems, but no explicit construction of the matrices $M(Y)$ that would be required (possibly larger than $n \times n$ in that setting) is given. In [LKZ15, PWBM18], it is shown that for PCA with non-Gaussian noise, preprocessing $Y$ by applying an entrywise nonlinearity improves the performance of a spectral algorithm.
>
> These works suggest that strong spectral algorithms can be designed for many problems by choosing a suitable $M$. We hoped to explore this idea, but were constrained to study $M$ that would yield $M(Y)$ whose spectrum is tractable to analyze with random matrix theory. We discuss this in Appendix F.3, explaining how we initially considered a general class of functions for $M$, encountered challenges in analyzing or formulating sensible limits for the eigenvalues of $M(Y)$, and finally chose nonlinear Laplacians as a tractable example and first step towards implementing our general proposal.
>
> In our revision, we will provide more context regarding the goal of learning spectral algorithms and our specific reasons for working with nonlinear Laplacians.
>
> [HSSS16] Hopkins, S.B., Schramm, T., Shi, J. and Steurer, D. 2016. Fast spectral algorithms from sum-of-squares proofs: tensor decomposition and planted sparse vectors. In STOC 2016 (pp. 178-191).

---

> ### Author Response · Authors · 2025-08-07
>
> We are wondering whether the reviewer has any concerns that we have not addressed in our reply, or any other follow-up questions? We are happy to clarify any other parts of our paper that remain unclear. If our replies and the edits we suggest affect their assessment, we would also appreciate it if the reviewer would reconsider their evaluation of the paper.

---

### Decision · Program_Chairs · 2025-09-17

**Decision:**

Accept (spotlight)

**Comment:**

This paper is concerned with a nonlinear spectral algorithm for estimating a rank-1 signal from noisy observation. The key idea is to exploit directional information (via a nonlinear mapping $\sigma$) that enhances the information from the row sum vector $Y1$. The paper rigorously showed that the nonlinear spectral algorithm outperforms SOTA such as AMP algorithm by enhancin the recoverability guarantee under scenarios such as Gaussian planted submatrix model.

The initial reviews of the paper is quite positive, and the rebuttal has further clarified some minor concerns such as numerical experiments, and unclear parts of the paper. Overall, the paper is above the acceptance threshold.